# Accelerated Evolving Set Processes for Local PageRank Computation

**Binbin Huang** [1]    **Luo Luo** [1,2]    **Yanghua Xiao** [3]    **Deqing Yang** [1,3]    **Baojian Zhou** [1,3*]

[1] School of Data Science, Fudan University,
[2] Shanghai Key Laboratory for Contemporary Applied Mathematics,
[3] Shanghai Key Laboratory of Data Science, School of Computer Science, Fudan University

bbhuang24@m.fudan.edu.cn
luoluo,shawyh,yangdeqing,bjzhou@fudan.edu.cn

## Abstract

This work proposes a novel framework based on *nested evolving set processes* to accelerate Personalized PageRank (PPR) computation. At each stage of the process, we employ a *localized* inexact proximal point iteration to solve a simplified linear system. We show that the time complexity of such localized methods is upper bounded by $\min\{\tilde{\mathcal{O}}(R^2/\epsilon^2), \tilde{\mathcal{O}}(m)\}$ to obtain an $\epsilon$-approximation of the PPR vector, where $m$ denotes the number of edges in the graph and $R$ is a constant defined via nested evolving set processes. Furthermore, the algorithms induced by our framework require solving only $\tilde{\mathcal{O}}(1/\sqrt{\alpha})$ such linear systems, where $\alpha$ is the damping factor. When $1/\epsilon^2 \ll m$, this implies the existence of an algorithm that computes an $\epsilon$-approximation of the PPR vector with an overall time complexity of $\tilde{\mathcal{O}}(R^2/(\sqrt{\alpha}\epsilon^2))$, independent of the underlying graph size. Our result resolves an open conjecture from existing literature [19, 52]. Experimental results on real-world graphs validate the efficiency of our methods, demonstrating significant convergence in the early stages.

## 1 Introduction

We study efficient *local* methods for computing the PPR vector $\boldsymbol{\pi} \in \mathbb{R}^n$, defined by

$$\left(\boldsymbol{I} - (1-\alpha)(\boldsymbol{I} + \boldsymbol{A}\boldsymbol{D}^{-1})/2\right)\boldsymbol{\pi} = \alpha\boldsymbol{e}_s, \tag{1}$$

where $\boldsymbol{e}_s \in \mathbb{R}^n$ is the standard basis vector corresponding to the source node $s \in \mathcal{V}$, and $\alpha \in (0,1)$ is the damping factor. Here, $\boldsymbol{A} \in \mathbb{R}^{n \times n}$ and $\boldsymbol{D} \in \mathbb{R}^{n \times n}$ are the adjacency and degree matrices of an undirected graph $\mathcal{G}(\mathcal{V}, \mathcal{E})$ with $n = |\mathcal{V}|$ nodes and $m = |\mathcal{E}|$ edges, respectively. The vector $\boldsymbol{\pi}$ measures the importance of nodes in $\mathcal{V}$ from the perspective of the source node $s$, which is the steady-state distribution of a lazy random walk on $\mathcal{G}$. Specifically, given a precision parameter $\epsilon$, our goal is to design local algorithms that compute an $\epsilon$-approximation $\hat{\boldsymbol{\pi}}$, i.e., one that satisfies $\|\boldsymbol{D}^{-1}(\hat{\boldsymbol{\pi}} - \boldsymbol{\pi})\|_\infty \leq \epsilon$, while avoiding access to the entire graph.

Andersen et al. [4] proposed the first local method, called the Approximate Personalized PageRank (APPR) algorithm, to approximate $\boldsymbol{\pi}$, achieving a time complexity of $\mathcal{O}(1/(\alpha\epsilon))$. To further characterize the locality of $\boldsymbol{\pi}$, Fountoulakis et al. [20] introduced a variational formulation of Eq. (1) and applied a proximal gradient method to compute local estimates with a comparable time complexity to APPR. Both methods critically rely on the monotonically decreasing $\ell_1$-norm of the residual (or gradient) to ensure the time complexity remains locally bounded.

Note that Eq. (1) can be reformulated as a strongly-convex minimization problem with condition number $1/\alpha$. It is natural to ask whether accelerated local methods can be designed with a time

---

[*]Corresponding author

39th Conference on Neural Information Processing Systems (NeurIPS 2025).

complexity that depends on $1/\sqrt{\alpha}$ [19]. However, the main challenge lies in the fact that accelerated methods [16, 39] typically involve momentum terms, which disrupt the key property, namely the monotonically decreasing $\ell_1$-norm of the residual (or gradient), relied on by existing local algorithms [4, 20]. As a result, standard accelerated methods may access up to $n$ nodes per iteration, leading to known upper bounds of $\tilde{\mathcal{O}}(m/\sqrt{\alpha})$ for solving Eq. (1). To preserve monotonicity, Martínez-Rubio et al. [37] proposed a subspace-pursuit style algorithm that performs accelerated projected gradient descent (APGD) in each iteration; however, the number of APGD calls required can still be as large as $\mathcal{O}(|\mathcal{S}^*|)$, where $\mathcal{S}^*$ is the support of the optimal solution. Recently, Zhou et al. [52] introduced a localized Chebyshev method inspired by the evolving set process [38]. However, the proposed method is heuristic, and its convergence remains unknown, as the accelerated local bounds rely heavily on the assumption that the gradient norm decreases at each iteration.

This work develops a locally Accelerated Evolving Set Process (AESP) framework that provably runs $\tilde{\mathcal{O}}(1/\sqrt{\alpha})$ short evolving set processes instead of a single long one. Our AESP is based on inexact accelerated proximal point iterations to accelerate APPR. Each stage guarantees a monotonic decrease in the $\ell_1$-norm of the gradient by using local methods to solve a regularized PPR linear system with a constant condition number. Hence, it converges faster than APPR in the early stages.

Let $\overline{\mathrm{vol}}(\mathcal{S}_t)$ and $\overline{\gamma}_t$ denote the average volume and the average $\ell_1$-norm of the gradient ratio of the active nodes processed at the $t$-th round. We show that each evolving set process has a time complexity of $\tilde{\mathcal{O}}(\overline{\mathrm{vol}}(\mathcal{S}_t)/\overline{\gamma}_t)$, and that AESP-induced algorithms have a total time complexity of $\tilde{\mathcal{O}}(\overline{\mathrm{vol}}(\mathcal{S}_t)/(\sqrt{\alpha}\overline{\gamma}_t))$, matching the accelerated bound conjectured by Zhou et al. [52]. Additionally, we prove that $\overline{\mathrm{vol}}(\mathcal{S}_t)/\overline{\gamma}_t$ admits an upper bound of $\min\{\mathcal{O}(R^2/\epsilon^2), 2m\}$, where $R$ is a constant defined via nested evolving set processes. As a result, the algorithms induced by AESP achieve a time complexity bound that reflects a trade-off between the dependence on the condition number $1/\alpha$ and the per-round time complexity $\mathcal{O}(R^2/\epsilon^2)$. The AESP framework is also well-suited for solving the variational formulation of Eq. (1), as studied by Fountoulakis et al. [20], with the potential to achieve an accelerated time complexity.

To summarize,

- We propose an Accelerated Evolving Set Process (AESP) framework, which computes an $\epsilon$-approximation for PPR using $\tilde{\mathcal{O}}(1/\sqrt{\alpha})$ short evolving set process. Our framework is built upon the inexact proximal point algorithm, and naturally extends to solving the variational formulation of Eq. (1). Furthermore, the algorithms induced by AESP are parameter-free.

- Our accelerated methods are guaranteed to converge without any additional assumptions. We establish theoretical guarantees for the induced algorithms with the time complexity of $\tilde{\mathcal{O}}(\overline{\mathrm{vol}}(\mathcal{S}_t)/(\sqrt{\alpha}\overline{\gamma}_t))$, which matches the accelerated bound conjectured in the existing literature. This result improves upon existing $\tilde{\mathcal{O}}(\overline{\mathrm{vol}}(\mathcal{S}_t)/(\alpha\overline{\gamma}_t))$ from standard local methods. Furthermore, we show that $\overline{\mathrm{vol}}(\mathcal{S}_t)/\overline{\gamma}_t$ is bounded above by $\min\{\mathcal{O}(R^2/\epsilon^2), 2m\}$, which implies that the overall time complexity $\tilde{\mathcal{O}}(R^2/(\sqrt{\alpha}\epsilon^2))$ is independent of the underlying graph size when $1/\epsilon^2 \ll m$.

- Experimental results on large-scale graphs confirm the efficiency of our method. Unlike standard local methods, AESP-based methods demonstrate a significant speed-up during the early stages. Our code is publicly available for review and will be open-sourced upon publication.[2]

## 2 Preliminaries

**Notations and definitions.** Throughout this paper, we assume that the underlying simple graph $\mathcal{G}(\mathcal{V}, \mathcal{E})$ is undirected and connected, with $n = |\mathcal{V}|$ nodes and $m = |\mathcal{E}|$ edges. The adjacency matrix of $\mathcal{G}$ is denoted by $\boldsymbol{A} = [a_{uv}]$, where $a_{uv} = 1$ if there exists an edge $(u, v) \in \mathcal{E}$, and $a_{uv} = 0$ otherwise. The set of all neighbors of a node $v$ is denoted by $\mathcal{N}(v)$. The degree matrix $\boldsymbol{D}$ is diagonal and has each entry $D_{vv} = d_v = |\mathcal{N}(v)|$. For $\boldsymbol{x} \in \mathbb{R}^n$, the *support* of $\boldsymbol{x}$, denoted by $\mathrm{supp}(\boldsymbol{x})$, is the set of its nonzero indices: $\mathrm{supp}(\boldsymbol{x}) := \{v \in [n] : x_v \neq 0\}$. The *volume* of a node set $\mathcal{S} \subseteq \mathcal{V}$ is defined as the sum of all node degrees in $\mathcal{S}$, i.e., $\mathrm{vol}(\mathcal{S}) := \sum_{v \in \mathcal{S}} d_v$. Note $\mathrm{vol}(\mathcal{V}) = 2m$. For an integer $T$, we denote $[T] := \{1, 2, \ldots, T\}$.

---

[2]For details on the importance of local PPR computation and related work, see Appendix B.

We say that a differentiable function $g : \mathbb{R}^n \to \mathbb{R}$ is $\mu$-*strongly convex* if there exists a constant $\mu > 0$ such that $\forall \boldsymbol{x}, \boldsymbol{y} \in \mathbb{R}^n$, $g(\boldsymbol{y}) \geq g(\boldsymbol{x}) + \langle \nabla g(\boldsymbol{x}), \boldsymbol{y} - \boldsymbol{x} \rangle + \mu \|\boldsymbol{x} - \boldsymbol{y}\|_2^2/2$, where $\nabla g(\boldsymbol{x})$ is the gradient of $g$ at $\boldsymbol{x}$. We say $g : \mathbb{R}^n \to \mathbb{R}$ is $L$-*smooth* if there exists $L > 0$ such that $\forall \boldsymbol{x}, \boldsymbol{y} \in \mathbb{R}^n$, $g(\boldsymbol{y}) \leq g(\boldsymbol{x}) + \langle \nabla g(\boldsymbol{x}), \boldsymbol{y} - \boldsymbol{x} \rangle + L\|\boldsymbol{x} - \boldsymbol{y}\|_2^2/2$. The $u$-th entry of $\nabla g(\boldsymbol{x})$ is denoted as $\nabla_u g(\boldsymbol{x})$. When $g$ is convex and given a smoothing parameter $\eta$, the *proximal mapping* of $g$ at $\boldsymbol{y}$ is given by

$$\mathrm{prox}_{g/\eta}(\boldsymbol{y}) = \arg\min_{\boldsymbol{x} \in \mathbb{R}^n} \left\{ g(\boldsymbol{x}) + \frac{\eta}{2}\|\boldsymbol{x} - \boldsymbol{y}\|_2^2 \right\}, \text{ where } \eta > 0. \tag{2}$$

With a slight abuse of notation, we define the $\boldsymbol{D}^{1/2}$-*scaled gradient* of $g$ at $\boldsymbol{x}$ as $\nabla g^{1/2}(\boldsymbol{x}) := \boldsymbol{D}^{1/2}\nabla g(\boldsymbol{x})$, and the $\boldsymbol{D}^{-1/2}$-*scaled gradient* as $\nabla g^{-1/2}(\boldsymbol{x}) := \boldsymbol{D}^{-1/2}\nabla g(\boldsymbol{x})$.

## 2.1 Problem reformulations and properties

We solve the linear system in Eq. (1) by reformulating it as the following optimization problem

$$\min_{\boldsymbol{x} \in \mathbb{R}^n} \left\{ f(\boldsymbol{x}) \triangleq \frac{1}{2}\boldsymbol{x}^\top \boldsymbol{Q}\boldsymbol{x} - \alpha \boldsymbol{x}^\top \boldsymbol{D}^{-1/2}\boldsymbol{b} \right\}, \tag{P1}$$

where $\boldsymbol{Q} \triangleq \frac{1+\alpha}{2}\boldsymbol{I} - \frac{1-\alpha}{2}\boldsymbol{D}^{-1/2}\boldsymbol{A}\boldsymbol{D}^{-1/2}$, with the eigenvalues satisfying $\lambda(\boldsymbol{Q}) \in [\alpha, 1]$, and $\boldsymbol{b}$ is a sparse vector. The function $f$ is both $\mu$-strongly convex and $L$-smooth, with $\mu = \alpha$ and $L = 1$. The optimal solution of (P1) is denoted by $\boldsymbol{x}_f^* := \alpha \boldsymbol{Q}^{-1}\boldsymbol{D}^{-1/2}\boldsymbol{b}$. When $\boldsymbol{b} = \boldsymbol{e}_s$, it implies $\boldsymbol{\pi} := \boldsymbol{D}^{1/2}\boldsymbol{x}_f^*$. We define the set of $\epsilon$-approximation solutions to (P1) as

$$\mathcal{P}(\epsilon, \alpha, \boldsymbol{b}, \mathcal{G}) \triangleq \left\{ \boldsymbol{x} : \|\boldsymbol{D}^{-1/2}(\boldsymbol{x} - \boldsymbol{x}_f^*)\|_\infty \leq \epsilon \right\}. \tag{3}$$

Based on the above reformulation, we aim to design faster local methods that find $\hat{\boldsymbol{x}} \in \mathcal{P}$. To ensure $\hat{\boldsymbol{x}}$ is sparse, prior works [20, 37] considered the following variational reformulation

$$\min_{\boldsymbol{x} \in \mathbb{R}^n} \left\{ \psi(\boldsymbol{x}) \triangleq f(\boldsymbol{x}) + \hat{\epsilon}\alpha\|\boldsymbol{D}^{1/2}\boldsymbol{x}\|_1 \right\}. \tag{P2}$$

Let $\boldsymbol{x}_\psi^* \triangleq \arg\min_{\boldsymbol{x} \in \mathbb{R}^n} \psi(\boldsymbol{x})$ be the optimal solution of (P2). When $\hat{\epsilon} = \epsilon$, the first-order optimal condition implies that $\boldsymbol{x}_\psi^* \in \mathcal{P}(\epsilon, \alpha, \boldsymbol{e}_s, \mathcal{G})$. The next two lemmas present properties of PPR vectors and the optimal solutions of our reformulated problems.[3]

**Lemma 2.1** (Properties of $\boldsymbol{\pi}$). *Define the PPR matrix* $\boldsymbol{\Pi}_\alpha = \alpha \left(\frac{1+\alpha}{2}\boldsymbol{I} - \frac{1-\alpha}{2}\boldsymbol{A}\boldsymbol{D}^{-1}\right)^{-1}$. *Let the estimate-residual pair* $(\boldsymbol{p}, \boldsymbol{r})$ *for Eq. (1) satisfy* $\boldsymbol{r} = \boldsymbol{e}_s - \boldsymbol{\Pi}_\alpha^{-1}\boldsymbol{p}$. *Then,*

- *The PPR vector is given by* $\boldsymbol{\pi} = \boldsymbol{\Pi}_\alpha \boldsymbol{e}_s$, *which is a probability distribution, i.e.,* $\forall i \in \mathcal{V}$, $\pi_i > 0$ *and* $\|\boldsymbol{\pi}\|_1 = 1$. *For* $\epsilon > 0$, *the stop condition* $\|\boldsymbol{D}^{-1}\boldsymbol{r}\|_\infty < \epsilon$ *ensures* $\|\boldsymbol{D}^{-1}(\boldsymbol{p} - \boldsymbol{\pi})\|_\infty < \epsilon$.

- *The matrix* $\alpha \boldsymbol{Q}^{-1}$ *is similar to the matrix* $\boldsymbol{\Pi}_\alpha$, *i.e.,* $\alpha \boldsymbol{D}^{1/2}\boldsymbol{Q}^{-1}\boldsymbol{D}^{-1/2} = \boldsymbol{\Pi}_\alpha$. *Furthermore, the* $\ell_1$-*norm of* $\boldsymbol{\Pi}_\alpha$ *satisfies* $\|\boldsymbol{\Pi}_\alpha\|_1 = \|\boldsymbol{D}^{-1}\boldsymbol{\Pi}_\alpha \boldsymbol{D}\|_\infty = 1$.

**Lemma 2.2** (Properties of $\boldsymbol{x}_f^*$ and $\boldsymbol{x}_\psi^*$). *Denote the gradient of* $f$ *at* $\boldsymbol{x}$ *as* $\nabla f(\boldsymbol{x}) := \boldsymbol{Q}\boldsymbol{x} - \alpha \boldsymbol{D}^{-1/2}\boldsymbol{b}$ *and optimal solution* $\boldsymbol{x}_f^* = \alpha \boldsymbol{Q}^{-1}\boldsymbol{D}^{-1/2}\boldsymbol{b}$ *satisfying* $\boldsymbol{\pi} := \boldsymbol{D}^{1/2}\boldsymbol{x}_f^*$. *Define* $\boldsymbol{p} = \boldsymbol{D}^{1/2}\boldsymbol{x}$. *Then,*

- *The stop condition* $\|\boldsymbol{D}^{-1/2}\nabla f(\boldsymbol{x})\|_\infty < \alpha\epsilon$ *implies* $\|\boldsymbol{D}^{-1}(\boldsymbol{p} - \boldsymbol{\pi})\|_\infty < \epsilon$.

- *The objective* $f$ *is* $\mu$-*strongly convex and* $L$-*smooth with two constants* $\mu = \alpha$ *and* $L = 1$. *When* $\hat{\epsilon} = \epsilon$, *then* $\boldsymbol{x}_\psi^* \in \mathcal{P}(\epsilon, \alpha, \boldsymbol{e}_s, \mathcal{G})$ *and the solution is sparse, i.e.,* $|\mathrm{supp}(\boldsymbol{x}_\psi^*)| \leq 1/\hat{\epsilon}$.

## 2.2 Inexact accelerated proximal point framework

The inexact accelerated proximal point iteration is a well-known technique to improve the convergence rate of ill-conditioned convex optimization problems. It approximately solves a sequence of well-conditioned subproblems using linearly convergent first-order methods, thereby reducing the overall computational cost (see Chapter 5 in [16]). Catalyst [34] is a representative example of

---

[3]Proofs of lemmas and theorems are postponed to the Appendix A.

such methods. It employs a base algorithm to approximate the proximal operator, corresponding to solving an auxiliary strongly convex optimization problem. Specifically, starting with initial points $\boldsymbol{y}^{(0)} = \boldsymbol{x}^{(0)}$, for $t \geq 1$, Catalyst finds an approximate $\boldsymbol{x}^{(t)} \approx \operatorname{prox}_{f/\eta}(\boldsymbol{y}^{(t-1)})$ for solving (P1), and $\boldsymbol{x}^{(t)} \approx \operatorname{prox}_{\psi/\eta}(\boldsymbol{y}^{(t-1)})$ for solving (P2), where the prox operator is defined in Eq. (2). Given a smoothing parameter $\eta$ and an accuracy $\varphi > 0$, if $\boldsymbol{x}^{(t)}$ is guaranteed in the set of $\varphi$-approximations of the proximal operator $\operatorname{prox}_{f/\eta}(\boldsymbol{y}^{(t-1)})$ denoted by $\mathcal{H}(\varphi) \triangleq \{\boldsymbol{z} \in \mathbb{R}^n : h(\boldsymbol{z}) - h^* \leq \varphi\}$ with $h(\boldsymbol{z}) = f(\boldsymbol{z}) + \frac{\eta}{2}\|\boldsymbol{x} - \boldsymbol{z}\|_2^2$ and $h^*$ is the minimum of $h$. Then, $\boldsymbol{x}^{(t)}$ attains $\mathcal{O}(\varphi)$ precision by updating $\boldsymbol{y}^{(t)} = \boldsymbol{x}^{(t)} + \beta_t(\boldsymbol{x}^{(t)} - \boldsymbol{x}^{(t-1)})$ where $\{\beta_t\}_{t \geq 0}$ are momentum weights. However, directly applying this method still results in the standard accelerated time complexity of $\tilde{\mathcal{O}}(m/\sqrt{\alpha})$. The next section shows how to significantly reduce this bound to a local one using the AESP framework.

## 3 Accelerated Evolving Set Processes

This section presents our main results. We first introduce the nested ESP and propose two local inexact proximal operators. We then establish the accelerated convergence rate of AESP. Finally, we discuss potential improvements to this rate and its connections to related problems.

### 3.1 Nested evolving set process

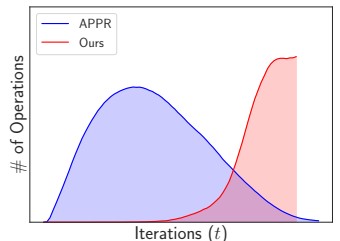

Figure 1: The comparison of volumes of ESP for APPR and Ours.

Our method generates estimates $\{\boldsymbol{x}^{(t)}\}_{t \geq 1}$. At each outer-loop iteration $t$, a local solver $\mathcal{M}$ maintains a sequence of *active sets* $\{\mathcal{S}_t^{(k)}\}_{k \geq 0}$ over the inner-loop iterations $k$. Updates are restricted to nodes within the active set, which is used to refine the approximation $\boldsymbol{z}_t^{(k)}$ in the inner loop. The next set $\mathcal{S}_t^{(k+1)}$ is determined solely by $\mathcal{S}_t^{(k)}$. We refer to this procedure as the *nested evolving set process*, defined as follows.

**Definition 3.1** (Nested evolving set process (ESP)). Given the configuration $\theta \triangleq (\alpha, \boldsymbol{b}, \mathcal{G})$, and a local method $\mathcal{M}$, the nested evolving set process at outer-loop iteration $t$ generates a sequence of $\{\mathcal{S}_t^{(k+1)}, \boldsymbol{z}_t^{(k+1)}\}_{k \geq 0}$ according to the dynamic system $(\mathcal{S}_t^{(k+1)}, \boldsymbol{z}_t^{(k+1)}) = \Phi_{\theta,\mathcal{M}}(\mathcal{S}_t^{(k)}, \boldsymbol{z}_t^{(k)})$, where $\mathcal{S}_t^{(k)} \subseteq \mathcal{V}$ is efficiently maintained using a queue data structure, avoiding accessing the entire graph. We say the process *converges* when $\mathcal{S}_t^{(K_t)} = \emptyset$ for some $K_t$. After $T$ outer-loop iterations, the generated sequences of active sets and estimation pairs are

$$(\mathcal{S}_1^{(0)}, \boldsymbol{z}_1^{(0)}) \to \cdots \to (\mathcal{S}_1^{(K_1)} = \emptyset, \boldsymbol{z}_1^{(K_1)} = \boldsymbol{x}^{(1)}), \ t = 1;$$

$$\vdots \qquad\qquad\qquad \vdots$$

$$(\mathcal{S}_T^{(0)}, \boldsymbol{z}_T^{(0)}) \to \cdots \to (\mathcal{S}_T^{(K_T)} = \emptyset, \boldsymbol{z}_T^{(K_T)} = \boldsymbol{x}^{(T)}), \ t = T.$$

At each outer-loop $t$, we denote the time complexity of the local solver $\mathcal{M}$ by $\mathcal{T}_t^{\mathcal{M}}$, which dominates the total cost. The total time complexity $\mathcal{T}$ of the nested ESP framework is then dominated by

$$\mathcal{T} \triangleq \sum_{t=1}^T \mathcal{T}_t^{\mathcal{M}} := K_t \cdot \overline{\operatorname{vol}}(\mathcal{S}_t), \quad \text{where} \quad \overline{\operatorname{vol}}(\mathcal{S}_t) \triangleq \frac{1}{K_t} \sum_{k=0}^{K_t-1} \operatorname{vol}(\mathcal{S}_t^{(k)}), \tag{4}$$

with $\overline{\operatorname{vol}}(\mathcal{S}_t)$ representing the average volume of the local process at time $t$. Fig. 1 illustrates how the number of operations evolves during the updates in APPR [4] and in our method under this process.

With this nested ESP, accelerated methods can be seamlessly incorporated to improve the efficiency of local PPR computation. Specifically, given the problem configuration $\theta = (\alpha, \boldsymbol{b}, \mathcal{G})$, at each outer-iteration $t$, we propose the following *localized* Catalyst-style updates

$$\text{AESP} \qquad \boldsymbol{x}^{(t)} = \mathcal{M}(\varphi_t, \boldsymbol{y}^{(t-1)}, \eta, \alpha, \boldsymbol{b}, \mathcal{G}), \qquad \boldsymbol{y}^{(t)} = \boldsymbol{x}^{(t)} + \beta_t(\boldsymbol{x}^{(t)} - \boldsymbol{x}^{(t-1)}), \tag{5}$$

where the momentum weight $\beta_t = (\alpha_{t-1}(1 - \alpha_{t-1}))/(\alpha_{t-1}^2 + \alpha_t)$, and $\alpha_t$ is updated in $(0, 1)$ by solving the equation $\alpha_t^2 = (1 - \alpha_t)\alpha_{t-1}^2 + \alpha_0^2 \alpha_t$ with an initial $\alpha_0$ (see the Scheme 2.2.9 in [39]).

For $t \geq 1$, the local operator obtains $\boldsymbol{x}^{(t)} \in \mathcal{H}_t(\varphi_t)$, defined as

$$\boldsymbol{x}^{(t)} \in \mathcal{H}_t(\varphi_t) \triangleq \{\boldsymbol{z} \in \mathbb{R}^n : h_t(\boldsymbol{z}) - h_t^* \leq \varphi_t\}, \tag{C1}$$

where $h_t^*$ is the minimal value of $h_t$, which is the proximal operator objective at $t$-th iteration

$$h_t(\boldsymbol{z}) \triangleq f(\boldsymbol{z}) + \frac{\eta}{2}\|\boldsymbol{z} - \boldsymbol{y}^{(t-1)}\|_2^2. \tag{6}$$

Thus, the minimizer $\boldsymbol{x}_t^* \triangleq \arg\min_{\boldsymbol{z} \in \mathbb{R}^n} h_t(\boldsymbol{z})$ is given by $\boldsymbol{x}_t^* := \mathrm{prox}_{f/\eta}(\boldsymbol{y}^{(t-1)}) = (\boldsymbol{Q} + \eta \boldsymbol{I})^{-1}\boldsymbol{b}^{(t-1)}$ with $\boldsymbol{b}^{(t-1)} = \alpha \boldsymbol{D}^{-1/2}\boldsymbol{b} + \eta \boldsymbol{y}^{(t-1)}$. To characterize the time complexity of the AESP framework, it is convenient to define the following constant

$$R := \max\left\{\|\nabla h_t^{1/2}(\boldsymbol{z}_t^{(0)})\|_1 / \|\nabla h_1^{1/2}(\boldsymbol{z}_1^{(0)})\|_1 : \forall t \in [T]\right\}. \tag{7}$$

The following lemma is key to controlling the time complexity of the local algorithm $\mathcal{M}$.

**Lemma 3.2.** *Let $h_t$ be defined in Eq. (6), and suppose that the initial point $\boldsymbol{z}_t^{(0)}$ of $t$-th process satisfies $\nabla h_t(\boldsymbol{z}_t^{(0)}) \neq \boldsymbol{0}$. If there exists a local algorithm $\mathcal{M}$ such that $\|\nabla h_t^{1/2}(\boldsymbol{z}_t^{(K_t)})\|_1 < \|\nabla h_t^{1/2}(\boldsymbol{z}_t^{(0)})\|_1$, then for a stopping condition $\|\nabla h_t^{-1/2}(\boldsymbol{z}_t^{(k)})\|_\infty < \epsilon_t$ of $\mathcal{M}$ with*

$$\epsilon_t \triangleq \max\left\{\sqrt{\frac{(\mu + \eta)\varphi_t}{m}}, \frac{2(\eta + \alpha)\varphi_t}{\|\nabla h_t^{1/2}(\boldsymbol{z}_t^{(0)})\|_1}\right\}, \text{ where } \varphi_t > 0, \tag{8}$$

*the final solution $\boldsymbol{z}_t^{(K_t)}$ is guaranteed in the ball, i.e., $\boldsymbol{z}_t^{(K_t)} \in \mathcal{H}_t(\varphi_t)$ as defined in* (C1).

Lemma 3.2 provides a way to find $\boldsymbol{x}^{(t)} \in \mathcal{H}_t(\varphi_t)$ under the condition that $\mathcal{M}$ satisfies the monotonicity property, $\|\nabla h_t^{1/2}(\boldsymbol{z}_t^{(K_t)})\|_1 \leq \|\nabla h_t^{1/2}(\boldsymbol{z}_t^{(0)})\|_1$. The next subsection introduces two operators that satisfy this monotonicity property while maintaining local time complexity.

## 3.2 Localized inexact proximal operators

This subsection introduces two localized inexact proximal operators with optimized step sizes, including local gradient descent (LocGD) and an optimized version of APPR (LocAPPR), for computing $\boldsymbol{z}_t^{(K_t)} \in \mathcal{H}_t(\varphi_t)$.[4] Given $\boldsymbol{z}_t^{(0)} \in \mathbb{R}^n$, the first local operator is iteratively defined as

$$\text{LocGD} \qquad \boldsymbol{z}_t^{(k+1)} = \boldsymbol{z}_t^{(k)} - \frac{2\nabla h_t(\boldsymbol{z}_t^{(k)}) \circ \mathbf{1}_{\mathcal{S}_t^k}}{1 + \alpha + 2\eta}, \text{ for } k \geq 0, \tag{9}$$

where $\circ$ means element-wise multiplication. For each $u \in \mathcal{S}_t^k$, then $u$-th entry of $\mathbf{1}_{\mathcal{S}_t^k}$ is 1, otherwise it is 0. The active node set $\mathcal{S}_t^k$ is determined by the following activation condition $\mathcal{S}_t^k = \{u : |\nabla_u h_t^{-1/2}(\boldsymbol{z}_t^{(k)})| \geq \epsilon_t\}$. The stopping criterion for LocGD is when $\mathcal{S}_t^{K_t} = \emptyset$, which is $\|\nabla h_t^{-1/2}(\boldsymbol{z})\|_\infty < \epsilon_t$ as stated in Lemma 3.2. To analyze the convergence and time complexity of LocGD, we characterize the sequences $\{\mathrm{vol}(\mathcal{S}_t^k)\}_{k \geq 0}$, and $\{\|\nabla h_t^{1/2}(\boldsymbol{z}_t^{(k)})\|_1\}_{k \geq 0}$ generated by $\Phi_{\theta, \text{LocGD}}$. To quantify the ratio of progress, we define the average $\ell_1$-norm of the gradient ratio as

$$\overline{\gamma}_t \triangleq \frac{1}{K_t}\sum_{k=0}^{K_t-1}\left\{\gamma_t^{(k)} \triangleq \frac{\|\nabla h_t^{1/2}(\boldsymbol{z}_t^{(k)}) \circ \mathbf{1}_{\mathcal{S}^{(k)}}\|_1}{\|\nabla h_t^{1/2}(\boldsymbol{z}_t^{(k)})\|_1}\right\}. \tag{10}$$

When $\mathcal{S}_t^k = \mathcal{V}$, convergence is straightforward to observe, yielding $\overline{\gamma}_t = 1$ and $\overline{\mathrm{vol}}(\mathcal{S}_t)/\overline{\gamma}_t = 2m$. The quantity $\overline{\mathrm{vol}}(\mathcal{S}_t)/\overline{\gamma}_t$ is a meaningful measure of time complexity as $\overline{\mathrm{vol}}(\mathcal{S}_t)/\overline{\gamma}_t \leq 2m$. The following theorem establishes the local convergence rate and time complexity of LocGD.

**Theorem 3.3 (Convergence of LocGD).** *Let $h_t$ be defined in Eq. (6). LocGD (Algorithm 3) is used to minimize $h_t(\boldsymbol{z})$ and returns $\boldsymbol{z}_t^{(K_t)} = \text{LocGD}(\varphi_t, \boldsymbol{y}^{(t-1)}, \eta, \alpha, \boldsymbol{b}, \mathcal{G}) \in \mathcal{H}_t(\varphi_t)$. Recall the $\boldsymbol{D}^{1/2}$-scaled gradient $\nabla h_t^{1/2}(\boldsymbol{z}_t^{(k)}) := \boldsymbol{D}^{1/2}\nabla h_t(\boldsymbol{z}_t^{(k)})$. For $k \geq 0$, the scaled gradient satisfies*

$$\left\|\nabla h_t^{1/2}(\boldsymbol{z}_t^{(k+1)})\right\|_1 \leq \left(1 - \tau\gamma_t^{(k)}\right)\left\|\nabla h_t^{1/2}(\boldsymbol{z}_t^{(k)})\right\|_1,$$

---

[4]See implementation details of LocGD (Algorithm 3) and LocAPPR (Algorithm 4) in Appendix C.

**Algorithm 1** AESP($\epsilon, \alpha, \boldsymbol{b}, \eta, \mathcal{G}, \mathcal{M}$)

1: $\boldsymbol{y}^{(0)} = \boldsymbol{x}^{(0)} = \boldsymbol{0}, c = 1 - 0.9\sqrt{\mu/(\mu+\eta)}$
2: $T$ is computed in Eq. (12)
3: **for** $t = 1, 2, \ldots, T$ **do**
4: $\quad \varphi_t = (L+\mu)\|\boldsymbol{b}\|_1^2 c^t/18$
5: $\quad \boldsymbol{x}^{(t)} = \mathcal{M}(\varphi_t, \boldsymbol{y}^{(t-1)}, \eta, \alpha, \boldsymbol{b}, \mathcal{G})$
6: $\quad$ // $\mathcal{M}$ in LOCAPPR or LOCGD
7: $\quad$ **if** $\{v : \epsilon\alpha\sqrt{d_v} \le |\nabla_v f(\boldsymbol{x}^{(t)})|\} = \emptyset$ **then**
8: $\quad\quad$ **break**
9: $\quad \boldsymbol{y}^{(t)} = \boldsymbol{x}^{(t)} + \frac{\sqrt{\mu+\eta}-\sqrt{\mu}}{\sqrt{\mu+\eta}+\sqrt{\mu}}\left(\boldsymbol{x}^{(t)} - \boldsymbol{x}^{(t-1)}\right)$
10: **Return** $\hat{\boldsymbol{x}} = \boldsymbol{x}^{(t)}$

**Algorithm 2** AESP-PPR($\epsilon, \alpha, s, \mathcal{G}, \mathcal{M}$)

1: $\boldsymbol{y}^{(0)} = \boldsymbol{x}^{(0)} = \boldsymbol{0}$
2: $T = \lceil \frac{10}{9}\sqrt{\frac{1-\alpha}{\alpha}} \log \frac{400(1-\alpha^2)}{\alpha^2\epsilon^2} \rceil$
3: **for** $t = 1, 2, \ldots, T$ **do**
4: $\quad \varphi_t = \frac{1+\alpha}{18}(1 - \frac{9}{10}\sqrt{\frac{\alpha}{1-\alpha}})^t$
5: $\quad$ // $\mathcal{M}$ is LOCAPPR or LOCGD
6: $\quad \boldsymbol{x}^{(t)} = \mathcal{M}(\varphi_t, \boldsymbol{y}^{(t-1)}, 1-2\alpha, \alpha, \boldsymbol{b}, \mathcal{G})$
7: $\quad$ **if** $\{v : \epsilon\alpha\sqrt{d_v} \le |\nabla_v f(\boldsymbol{x}^{(t)})|\} = \emptyset$ **then**
8: $\quad\quad$ **break**
9: $\quad \boldsymbol{y}^{(t)} = \boldsymbol{x}^{(t)} + \frac{\sqrt{1-\alpha}-\sqrt{\alpha}}{\sqrt{1-\alpha}+\sqrt{\alpha}}(\boldsymbol{x}^{(t)} - \boldsymbol{x}^{(t-1)})$
10: **Return** $\hat{\boldsymbol{\pi}} = \boldsymbol{D}^{1/2}\boldsymbol{x}^{(t)}$

where $\tau := \frac{2(\alpha+\eta)}{1+\alpha+2\eta}$ and $\gamma_t^{(k)}$ is the ratio defined in Eq. (10). Assume $\epsilon_t$ and stop condition are defined in Eq. (8) of Lemma 3.2, then the run time $\mathcal{T}_t^{\text{LOCGD}}$, as defined in Eq. (4), is bounded by

$$\mathcal{T}_t^{\text{LOCGD}} \le \min\left\{\frac{\overline{\text{vol}}(\mathcal{S}_t)}{\tau\overline{\gamma}_t} \log \frac{C_{h_t}^0}{C_{h_t}^{K_t}}, \frac{C_{h_t}^0 - C_{h_t}^{K_t}}{\tau\epsilon_t}\right\},$$

where $C_{h_t}^i = \|\nabla h_t^{1/2}(\boldsymbol{z}_t^{(i)})\|_1$ denote constants. Furthermore, $\overline{\text{vol}}(\mathcal{S}_t)/\overline{\gamma}_t \le \min\{C_{h_t}^0/\epsilon_t, 2m\}$.

Since the Hessian of $h_t$ is $\boldsymbol{Q} + \eta\boldsymbol{I}$ and its eigenvalues $\lambda(\boldsymbol{Q} + \eta\boldsymbol{I}) \in [\eta + \alpha, \eta + 1]$, the condition number of the shifted linear system is $(\eta+1)/(\eta+\alpha)$, which is smaller than $1/\alpha$. Hence, the time complexity per round improves from $\mathcal{O}(1/(\alpha\epsilon_t))$ to $\mathcal{O}(1/(\tau\epsilon_t))$. In our later analysis, we show that for $\alpha < 0.5$ and $\eta = 1 - 2\alpha$, then $\tau = 2/3$, meaning that each local process is independent of $1/\alpha$.

Following the same analysis as LOCGD, we introduce an optimized version of APPR with online updates. For $u_i \in \mathcal{S}_t^k = \{u_1, u_2, \ldots, u_{|\mathcal{S}_t^k|}\}$, the optimized APPR updates are

$$\text{LOCAPPR} \quad \boldsymbol{z}_t^{(k_{i+1})} = \boldsymbol{z}_t^{(k_i)} - \frac{2\nabla h_t(\boldsymbol{z}_t^{(k_i)}) \circ \mathbf{1}_{\{u_i\}}}{1 + \alpha + 2\eta}, \tag{11}$$

where $k_i = k + (i-1)/|\mathcal{S}_t^k|$ for $i = 1, 2, \ldots, |\mathcal{S}_t^k|$. The convergence analysis of LOCAPPR follows a similar approach to that of LOCGD as stated in Theorem A.3 of the Appendix A.

### 3.3 Time complexity analysis and AESP-PPR

This subsection presents the overall time complexity of the AESP framework. First, we analyze the number of outer-loop iterations required to achieve $f(\boldsymbol{x}^{(T)}) - f(\boldsymbol{x}_f^*) \le \mu\epsilon^2/2$, which guarantees $\|\boldsymbol{D}^{-1/2}(\boldsymbol{x}^{(t)} - \boldsymbol{x}_f^*)\|_\infty \le \epsilon$. We derive the iteration complexity of AESP in the following lemma.

**Lemma 3.4** (Outer-loop iteration complexity of AESP). *If each iteration of AESP, presented in Algorithm 1, finds $\boldsymbol{x}^{(t)} := \boldsymbol{z}_t^{(K_t)}$ using $\mathcal{M}$, satisfying $h_t(\boldsymbol{z}_t^{(K_t)}) - h_t^* \le \varphi_t := (L+\mu)\|\boldsymbol{b}\|_1^2(1 - \rho)^t/18$, then the total number of iterations $T$ required to ensure $\hat{\boldsymbol{x}} = \text{AESP}(\epsilon, \alpha, \boldsymbol{b}, \eta, \mathcal{G}, \mathcal{M}) \in \mathcal{P}(\epsilon, \alpha, \boldsymbol{b}, \mathcal{G})$ as defined in Eq. (3), for solving (P1), satisfies the bound*

$$T \le \frac{1}{\rho}\log\left(\frac{4(L+\mu)\|\boldsymbol{b}\|_1^2}{\mu\epsilon^2(\sqrt{q}-\rho)^2}\right), \text{ where } \rho = 0.9\sqrt{q} \text{ and } q = \frac{\mu}{\mu+\eta}. \tag{12}$$

*Furthermore, $\varphi_t$ has a lower bound $\varphi_t \ge \mu\epsilon^2(\sqrt{q}-\rho)^2/72$ for all $t \in [T]$.*

In a practical implementation, our AESP framework is an adaptation of the Catalyst acceleration method applied to local methods, as presented in Algorithm 1. Specifically, Line 7 serves as an early stopping condition since $T$ represents the worst-case number of iterations required. This stopping condition follows directly from Lemma 2.2, i.e., $\{v : \epsilon\alpha\sqrt{d_v} \le |\nabla_v f(\boldsymbol{x}^{(t)})|\} = \emptyset$, which implies $\|\nabla f^{-\frac{1}{2}}(\boldsymbol{x}^{(t)})\|_\infty \le \epsilon\alpha$. Line 9 updates the sequence $\{\beta_t\}_{t\ge 1}$ using $\beta_t = \frac{\sqrt{\mu+\eta}-\sqrt{\mu}}{\sqrt{\mu+\eta}+\sqrt{\mu}}$ as $\alpha_t = \alpha_0 = \sqrt{q}$. The computational cost of verifying this condition is dominated by $\mathcal{T}_t^{\mathcal{M}}$. To minimize $T$,

the goal is to choose a suitable $\eta$ to maximize $1/(\tau(\mu + \eta))$. When $\alpha < 0.5$, we find that setting $\eta = (L - 2\mu)$, though not necessarily optimal, is sufficient for our purposes. Based on this analysis, we now present the total time complexity for solving (P1) using AESP in the following theorem.

**Theorem 3.5** (Time complexity of AESP). *Let the simple graph $\mathcal{G}(\mathcal{V}, \mathcal{E})$ be connected and undirected, and let $f(\boldsymbol{x})$ be defined in (P1). Assume the precision $\epsilon > 0$ satisfies $\{i : |b_i| \geq \epsilon d_i\} \neq \emptyset$ and damping factor $\alpha < 1/2$. Applying $\hat{\boldsymbol{x}} = \mathrm{AESP}(\epsilon, \alpha, \boldsymbol{b}, \eta, \mathcal{G}, \mathcal{M})$ with $\eta = L - 2\mu$ and $\mathcal{M}$ be either $\mathrm{LOCGD}$ or $\mathrm{LOCAPPR}$, then AESP presented in Algorithm 1, finds a solution $\hat{\boldsymbol{x}}$ such that $\|\boldsymbol{D}^{-1/2}(\hat{\boldsymbol{x}} - \boldsymbol{x}_f^*)\|_\infty \leq \epsilon$ with the dominated time complexity $\mathcal{T}$ bounded by*

$$\mathcal{T} \leq \sum_{t=1}^{T} \min \left\{ \frac{\overline{\mathrm{vol}}(\mathcal{S}_t)}{\tau \overline{\gamma}_t} \log \frac{C_{h_t}^0}{C_{h_t}^{K_t}}, \frac{C_{h_t}^0 - C_{h_t}^{K_t}}{\tau \epsilon_t} \right\}, \; with \; \frac{\overline{\mathrm{vol}}(\mathcal{S}_t)}{\overline{\gamma}_t} \leq \min \left\{ \frac{C_{h_t}^0}{\epsilon_t}, 2m \right\},$$

*where $\tau$, $\epsilon_t$, $C_{h_t}^0$ and $C_{h_t}^{K_t}$ are defined in Theorem 3.3. Furthermore, $q = \mu/(L - \mu)$ and the number of outer iterations satisfies*

$$T \leq \frac{10}{9\sqrt{q}} \log \left( \frac{400(L + \mu)\|\boldsymbol{b}\|_1^2}{\mu \epsilon^2 q} \right) = \tilde{\mathcal{O}} \left( \frac{1}{\sqrt{\alpha}} \right).$$

Roughly speaking, Theorem 3.5 indicates that AESP solves Eq. (P1) in a time complexity of

$$\mathcal{T} = \tilde{\mathcal{O}} \left( \frac{\overline{\mathrm{vol}}(\mathcal{S}_t)}{\sqrt{\alpha} \overline{\gamma}_t} \right) = \tilde{\mathcal{O}} \left( \frac{1}{\sqrt{\alpha} \epsilon_T} \right) = \tilde{\mathcal{O}} \left( \frac{1}{\sqrt{\alpha} \epsilon^2} \right),$$

where the last equality follows from $\epsilon_T = \mathcal{O}(\epsilon^2)$. This result is particularly meaningful when $\epsilon \geq 1/\sqrt{m}$. As argued in [19], in many real-world applications, it is typical that $1/\epsilon \ll n$. We now finalize our algorithm and present AESP-PPR for solving Eq. (1) in the following theorem.

**Theorem 3.6** (Time complexity of AESP-PPR). *Let the simple graph $\mathcal{G}(\mathcal{V}, \mathcal{E})$ be connected and undirected, assuming $\alpha < 1/2$. The PPR vector of $s \in \mathcal{V}$ is defined in Eq. (1), and the precision $\epsilon \in (0, 1/d_s)$. Suppose $\hat{\boldsymbol{\pi}} = \mathrm{AESP\text{-}PPR}(\epsilon, \alpha, s, \mathcal{G}, \mathcal{M})$ be returned by Algorithm 2. When $\mathcal{M}$ is either $\mathrm{LOCGD}$ (Algorithm 3) or $\mathrm{LOCAPPR}$ (Algorithm 4), then $\hat{\boldsymbol{\pi}}$ satisfies $\|\boldsymbol{D}^{-1}(\hat{\boldsymbol{\pi}} - \boldsymbol{\pi})\|_\infty \leq \epsilon$ and AESP-PPR has a dominated time complexity bounded by*

$$\mathcal{T} \leq \min \left\{ \tilde{\mathcal{O}} \left( \frac{\overline{\mathrm{vol}}(\mathcal{S}_{T_{\max}})}{\sqrt{\alpha} \overline{\gamma}_{T_{\max}}} \right), \tilde{\mathcal{O}} \left( \frac{\max_t C_{h_t}^0}{\sqrt{\alpha} \epsilon_T} \right) \right\} = \min \left\{ \tilde{\mathcal{O}} \left( \frac{m}{\sqrt{\alpha}} \right), \tilde{\mathcal{O}} \left( \frac{R^2/\epsilon^2}{\sqrt{\alpha}} \right) \right\}, \quad (13)$$

*where $T_{\max} := \arg\max_{t \in [T]} \overline{\mathrm{vol}}(\mathcal{S}_t)/\overline{\gamma}_t$ and $R$ is defined in Eq. (7).*

The time complexity derived in Eq. (13) is significant when $\epsilon \geq 1/\sqrt{m}$. Compared to ASPR [37], which requires $|\operatorname{supp}(\boldsymbol{x}_\psi^*)|$ iterations of APGD, our approach only needs $\mathcal{O}(1/\sqrt{\alpha})$ local evolving set processes. In contrast to $\mathrm{LOCCH}$ [52], which imposes a strong assumption on the $\boldsymbol{D}^{1/2}$-scaled gradient reduction, our method provides a provable stopping criterion and only requires a mild assumption on the bounded level set of the $\boldsymbol{D}^{1/2}$-scaled gradient during AESP-PPR updates.

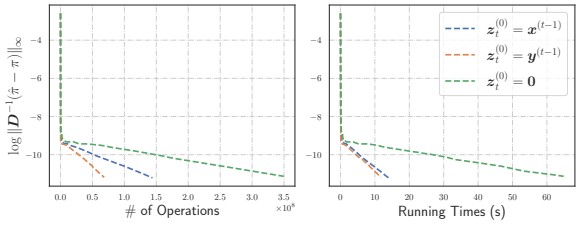

Figure 2: Convergence of $\log \|\boldsymbol{D}^{-1}(\hat{\boldsymbol{\pi}} - \boldsymbol{\pi})\|_\infty$ for AESP-$\mathrm{LOCAPPR}$ with three different initializations for $\boldsymbol{z}_t^{(0)}$ as a function of total operations and running times on the *com-dblp* graph.

**Initialization of $\boldsymbol{z}_t^{(0)}$.** We consider three possible initialization strategies for $\boldsymbol{z}_t^{(0)}$: 1) A cold start with $\boldsymbol{z}_t^{(0)} = \boldsymbol{0}$; 2) Using the previous estimate $\boldsymbol{z}_t^{(0)} = \boldsymbol{x}^{(t-1)}$; and 3) Momentum-based initialization, i.e., $\boldsymbol{z}_t^{(0)} = \boldsymbol{y}^{(t-1)}$. Among these, we find that the momentum-based strategy yields the best overall performance. This choice is well-motivated, since $\nabla h_t^{1/2}(\boldsymbol{y}^{(t-1)}) = -(\alpha \boldsymbol{b} - \boldsymbol{\Pi}_\alpha^{-1} \boldsymbol{D}^{1/2} \boldsymbol{y}^{(t-1)})$, which corresponds to the negative residual of Eq. (1) when treating $\boldsymbol{D}^{1/2} \boldsymbol{y}^{(t-1)}$ as an estimate.

Notably, $\boldsymbol{D}^{1/2} \boldsymbol{y}^{(t-1)} \to \boldsymbol{\pi}$ as $t \to \infty$, justifying this initialization. Fig. 2 empirically supports this analysis, showing that it requires the fewest outer-loop iterations.

**The assumption on the constant $R$.** A limitation of our theoretical analysis is that the constant $R$ is not universally bounded across all configurations $\theta = (\alpha, \boldsymbol{b}, \mathcal{G})$. In particular, we are unable to express $R$ solely in terms of graph size or input parameters. Nevertheless, empirical results (see Fig. 3) consistently show that $R$ remains a small constant and is largely insensitive to the graph size and the condition number, suggesting that this limitation has minimal practical impact. To further upper bound $R$, two possible strategies can be considered: The first is to add a simplex constraint $\Delta := \{\boldsymbol{x} : \|\boldsymbol{D}^{1/2}\boldsymbol{x}\|_1 = $

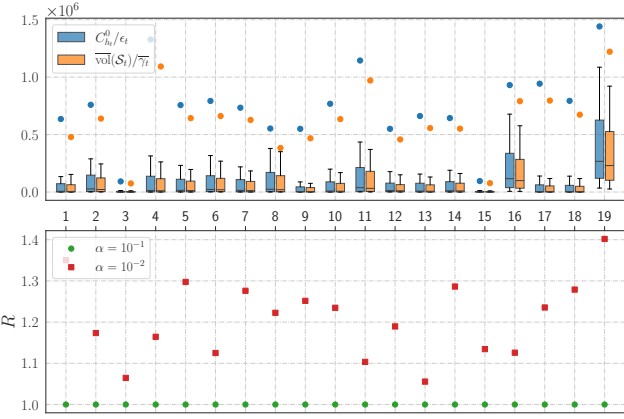

Figure 3: $C_{h_t}^0/\epsilon_t$, $\overline{\mathrm{vol}}(\mathcal{S}_t)/\overline{\gamma_t}$ and $R$ of AESP-LOCAPPR on 19 graphs (in ascending order of $n$) when $\boldsymbol{z}_t^{(0)} = \boldsymbol{y}^{(t-1)}$.

$1, \boldsymbol{x} \in \mathbb{R}_+^n\}$ to (P1) since $\|\boldsymbol{D}^{1/2}\boldsymbol{x}^{(t)}\|_1$ remains bounded, the quantity $\|\nabla h_t^{1/2}(\boldsymbol{z}_t^{(0)})\|_1$ can also be kept bounded. The projection onto $\Delta$ can be solved in $\mathcal{O}(|\operatorname{supp}(\boldsymbol{x}^{(t)})|\log n)$ time [18]. The second strategy is to adopt an adaptive restart scheme [16, 41], which can ensure that $\|\nabla h_t^{1/2}(\boldsymbol{z}_t^{(0)})\|_1 \leq \|\nabla h_1^{1/2}(\boldsymbol{z}_1^{(0)})\|$ throughout the iterations. This may lead to $R \leq 1$ during adaptive updates.

### 3.4 Discussions and related problems

**Adaptive strategy for estimating $\epsilon_t$.** Since $\varphi_T = \mathcal{O}(\epsilon^2)$, our conservative estimation of $e_t$ suggests that $1/\epsilon_T = \mathcal{O}(1/\epsilon^2)$. Hence, the time complexity in Eq. (13) remains unsatisfactory when $\epsilon \in [1/\sqrt{m}, 1/m]$. Naturally, one may ask whether the final bound in our time complexity analysis is optimal. We observed that the bound in Lemma 3.2 provides a pessimistic estimation of the objective error. A more careful error estimate can potentially refine this analysis. Specifically, let $\boldsymbol{x}_t^{(K_t)}$ be the output of either LOCGD or LOCAPPR. Then, by Corollary A.7, we have

$$h_t(\boldsymbol{z}_t^{(K_t)}) - h_t(\boldsymbol{x}_t^*) \leq \frac{\left\|\nabla h_t^{1/2}(\boldsymbol{z}_t^{(0)})\right\|_1^2}{(1-\alpha)} \prod_{k=0}^{K_t-1} \left(1 - 2\gamma_t^{(k)}/3\right)^2. \tag{14}$$

Inspired by Eq. (14), and observing that $\gamma_t^{(k)}$ can be computed in $\mathrm{vol}(\mathcal{S}_t^{(k)})$ time per iteration, one can propose an adaptive adjustment for $\epsilon_t$ as follows: We progressively try different precision levels from $\epsilon_t^1 = \sqrt{(1-\alpha)\varphi_t/2}, \epsilon_t^2 = \sqrt{(1-\alpha)\varphi_t/2^2}, \ldots$, to $\epsilon_t^s = \sqrt{(1-\alpha)\varphi_t/m}$, and at each time, verify whether $\|\nabla h_t(\hat{\boldsymbol{x}})\|_2 \leq \sqrt{2(1-\alpha)\varphi_t}$ is satisfied. This may potentially reduce the runtime, thereby lowering the time complexity per process.

**AESP for the variational form of PPR.** Our AESP framework naturally extends to solving (P2), where each outer iteration solves the following inexact proximal operator: $\boldsymbol{x}^{(t)} \approx \mathrm{prox}_{\psi/\eta}(\boldsymbol{y}^{(t-1)})$. We can employ ISTA or greedy coordinate descent to design local operators within the AESP framework. However, whether these standard methods can be effectively localized remains unclear. A previous study by Fountoulakis et al. [20] suggested that the monotonicity of the $\boldsymbol{D}^{1/2}$-scaled gradient of ISTA depends on the non-negativity of the initial $\boldsymbol{z}_t^{(0)}$, which it may not be true during the updates. It remains an open question whether one can achieve a time complexity of $\tilde{\mathcal{O}}(R^2/(\sqrt{\alpha}\hat{\epsilon}^2))$ without imposing strict non-negativity constraints.

**Application to other related problems.** Our results or framework can also be applied to other related problems. For example, in the thesis of Lofgren [35] (Section 3.3, Corollary 1), the author proposed a bidirectional PPR algorithm for undirected graphs with a relative error guarantee. If our techniques can be incorporated, then their expected runtime could potentially improve from

$$\mathcal{O}\left(\sqrt{m}/(\alpha\epsilon)\right) \xrightarrow{\text{improves to}} \mathcal{O}\left(\sqrt{m}/(\sqrt{\alpha}\epsilon)\right).$$

Additionally, our approach could benefit other problems of single-source PPR estimation, as highlighted in a recent survey by Yang et al. [50], which shows that many PPR-related computation methods have a time complexity proportional to $1/\alpha$.

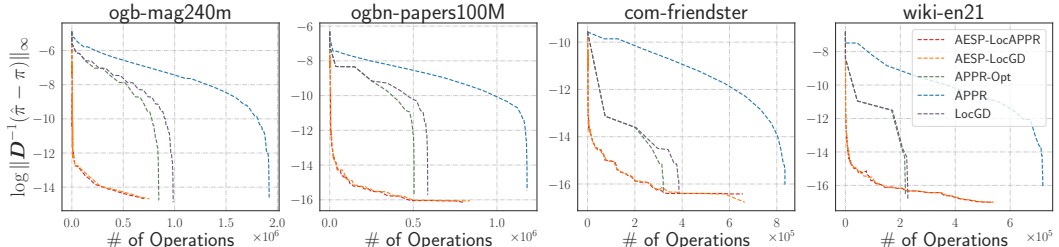

Figure 4: Performance of estimation error reduction, $\log \|\boldsymbol{D}^{-1}(\hat{\boldsymbol{\pi}} - \boldsymbol{\pi})\|_\infty$, as a function of operations $\mathcal{T}$, on the graph *ogb-mag240m*, *ogbn-papers100M*, *com-friendster* and *wiki-en21* with $\alpha = 0.01$ and $\epsilon = 10^{-6}$ where the graph can scale up to $n = 244M$ and $m = 1.728B$.

## 4 Experiments

We conduct experiments on computing PPR for a single source node. We evaluate local methods on real-world graphs to address the following two questions: 1) Does AESP accelerate standard local methods? 2) How do our proposed approaches compare in efficiency with existing local acceleration methods? *Additional experimental results are provided in the Appendix C. Our code is publicly available at https://github.com/Rick7117/aesp-local-pagerank.*

**AESP achieves early-stage acceleration and overall efficiency.** We begin by conducting experiments on four large-scale real-world graphs, with the number of nodes ranging from 6 million to 240 million. We implement two AESP-PPR variants: AESP-LOCGD (where $\mathcal{M}$ = LOCGD) and AESP-LOCAPPR (where $\mathcal{M}$ = LOCAPPR). For comparison, we consider three baselines: 1) APPR [4], 2) APPR-opt (APPR with the optimal step size $2/(1 + \alpha)$), and 3) LOCGD (with the optimal step size $2/(1 + \alpha)$). Fig. 4 presents our experimental results on four real-world graphs. Among all methods, AESP-LOCAPPR is the most efficient due to its online per-coordinate updates. Interestingly, by solving shifted linear systems, AESP-based methods achieve *much faster convergence in the early stages* than the baselines.

**AESP accelerates standard local methods when $\alpha$ is small.** We further validate whether AESP effectively accelerates standard local methods such as LOCGD and APPR. We fix the precision at $\epsilon = 10^{-7}$ and vary $\alpha$ from $10^{-3}$ to $10^{-1}$, selecting 50 source nodes $s$ for each $\alpha$ at random. Compared to non-accelerated methods, both AESP-LOCGD and AESP-LOCAPPR significantly reduce the number of operations and running times required, particularly when $\alpha$ is small.

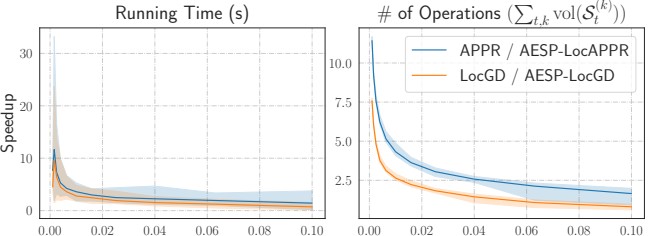

Figure 5: Speedup of AESP-based methods over standard local solvers (LOCAPPR, LOCGD) as a function of $\alpha$, on the *com-dblp* graph with $\epsilon = 0.1/n$ and $\alpha \in (10^{-3}, 10^{-1})$.

## 5 Conclusions and Discussions

In this paper, we propose the Accelerated Evolving Set Process (AESP) framework, which leverages an accelerated inexact proximal operator approach to improve the efficiency of Personalized PageRank (PPR) computation. Our methods provably run in $\tilde{\mathcal{O}}(1/\sqrt{\alpha})$ iterations, each performing a short evolving set process. We establish a time complexity of $\tilde{\mathcal{O}}(\overline{\mathrm{vol}}(\mathcal{S}_t)/(\sqrt{\alpha}\overline{\gamma}_t))$, and under a mild assumption on the bounded ratio of $\ell_1$-norm-scaled gradients, we show that $\mathcal{O}(\overline{\mathrm{vol}}(\mathcal{S}_t)/\overline{\gamma}_t)$ is upper-bounded by $\min\{\mathcal{O}(R^2/\epsilon^2), m\}$. This result demonstrates that our approach is sublinear in time when $\epsilon > 1/\sqrt{m}$, significantly improving over standard methods. Our algorithms not only advance local PPR computation but also offer a general-purpose framework that may benefit a wide range of problems, including positive definite linear systems and related tasks in graph analysis.

Despite these advantages, when $\epsilon < 1/\sqrt{m}$, the local bounds degrade to $\tilde{\mathcal{O}}(m/\sqrt{\alpha})$. A limitation of our theoretical analysis is that the constant $R$ is not universally bounded across all configurations

$\theta = (\alpha, \boldsymbol{b}, \mathcal{G})$. A key open question remains whether the $1/\epsilon^2$ dependence in our complexity bound can be further reduced to match the conjectured $\tilde{\mathcal{O}}(1/(\sqrt{\alpha}\epsilon))$ from existing literature [19], and whether the dependence on the constant $R$ can be entirely eliminated.

## Acknowledgments and Disclosure of Funding

The authors would like to thank the anonymous reviewers for their helpful comments. Luo is supported by the Major Key Project of Pengcheng Laboratory (No. PCL2024A06), National Natural Science Foundation of China (No. 12571557), National Natural Science Foundation of China (No. 62206058), and Shanghai Basic Research Program (23JC1401000). The work of Baojian Zhou is sponsored by the National Natural Science Foundation of China (No. KRH2305047). The work of Deqing Yang is supported by the Chinese NSF Major Research Plan (No.92270121), General Program (No.62572129). The computations in this research were performed using the CFFF platform of Fudan University.

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

# A Missing Proofs

In this section, we first summarize all notations in Table 1 for clarity, followed by the presentation of all missing proofs.

Table 1: Notations

| | Description |
|---|---|
| $\mathcal{G}(\mathcal{V}, \mathcal{E})$ | An undirected and connected simple graph with $n = |\mathcal{V}|$ nodes and $m = |\mathcal{E}|$ edges. |
| $\alpha$ | The damping factor $\alpha$ which lies in the interval $(0, 1)$. |
| $\boldsymbol{A}$ | The adjacency matrix of $\mathcal{G}$. |
| $\boldsymbol{D}$ | The diagonal degree matrix of $\mathcal{G}$. |
| $\boldsymbol{\Pi}_\alpha$ | The PPR matrix defined as $\boldsymbol{\Pi}_\alpha = \alpha \left( \frac{1+\alpha}{2} \boldsymbol{I} - \frac{1-\alpha}{2} \boldsymbol{A}\boldsymbol{D}^{-1} \right)^{-1}$. |
| $\boldsymbol{Q}$ | The symmetric matrix $\boldsymbol{Q} = \frac{1+\alpha}{2} \boldsymbol{I} - \frac{1-\alpha}{2} \boldsymbol{D}^{-1/2}\boldsymbol{A}\boldsymbol{D}^{-1/2}$. |
| $\tilde{\boldsymbol{Q}}$ | The shifted matrix of $\boldsymbol{Q}$, i.e., $\tilde{\boldsymbol{Q}} = \boldsymbol{Q} + \eta\boldsymbol{I}$. |
| $\mathrm{supp}(\boldsymbol{x})$ | The set of nonzero indices of $\boldsymbol{x} \in \mathbb{R}^n$, i.e., $\mathrm{supp}(\boldsymbol{x}) := \{v : x_v \neq 0\}$. |
| $\mathrm{vol}(\mathcal{S})$ | The *volume* of $\mathcal{S} \subseteq \mathcal{V}$ is the sum of all degrees in $\mathcal{S}$, that is, $\mathrm{vol}(\mathcal{S}) = \sum_{v \in \mathcal{S}} d_v$. |
| $\mathcal{S}_t^{(k+1)}$ | The set of active nodes at outer-loop iteration $t$ and inner-loop iteration $k$. |
| $K_t$ | The maximum number of iterations of the inner loop at outer-loop iteration $t$. |
| $\overline{\mathrm{vol}}(\mathcal{S}_t)$ | The average volume of $t$-th local process: $\overline{\mathrm{vol}}(\mathcal{S}_t) \triangleq \frac{1}{K_t} \sum_{k=0}^{K_t-1} \mathrm{vol}(\mathcal{S}_t^{(k)})$. |
| $\gamma_t^{(k)}$ | Active ratio at outer-iteration $t$ and inner-loop iteration $k$ defined in Eq. (10). |
| $\overline{\gamma}_t$ | Average active ratio of $t$-th local process defined in Eq. (10). |
| $\boldsymbol{e}_s$ | The standard basis vector $\boldsymbol{e}_s$ where the $s$-th element is 1, and all other elements are 0. |
| $\boldsymbol{\pi}$ | The PPR vector $\boldsymbol{\pi} = \boldsymbol{\Pi}_\alpha \boldsymbol{e}_s$. |
| $\boldsymbol{p}$ | The $\epsilon$-approximate PPR vector s.t. $\|\boldsymbol{D}^{-1}(\boldsymbol{p} - \boldsymbol{\pi})\|_\infty \leq \epsilon$. |
| $\boldsymbol{b}$ | A sparse vector in Eq. (P1). |
| $\boldsymbol{b}^{(t)}$ | $\boldsymbol{b}^{(t)} = \alpha\boldsymbol{D}^{-1/2}\boldsymbol{b} + \eta\boldsymbol{y}^{(t)}$. |
| $\boldsymbol{r}$ | Residual of $\epsilon$-approximate PPV satisfying $\boldsymbol{r} = \boldsymbol{e}_s - \boldsymbol{\Pi}_\alpha^{-1}\boldsymbol{p}$. |
| $\nabla f^{1/2}(\boldsymbol{x})$ | $\boldsymbol{D}^{1/2}$-scaled gradient of $f$ at $\boldsymbol{x}$, $\nabla f^{1/2}(\boldsymbol{x}) = \boldsymbol{D}^{1/2}\nabla f(\boldsymbol{x})$. |
| $\nabla f^{-1/2}(\boldsymbol{x})$ | $\boldsymbol{D}^{-1/2}$-scaled gradient of $f$ at $\boldsymbol{x}$, $\nabla f^{-1/2}(\boldsymbol{x}) = \boldsymbol{D}^{-1/2}\nabla f(\boldsymbol{x})$. |
| $f$ | Quadratic function defined in Eq. (P1). |
| $h_t$ | Proximal operator objective at $t$-th iteration defined in Eq. (6). |
| $\mu, L$ | Two constants with $\frac{\mu}{2}\|\boldsymbol{x}-\boldsymbol{y}\|_2^2 \leq f(\boldsymbol{y}) - f(\boldsymbol{x}) - \langle \nabla f(\boldsymbol{x}), \boldsymbol{y}-\boldsymbol{x}\rangle \leq \frac{L}{2}\|\boldsymbol{x}-\boldsymbol{y}\|_2^2$. |
| $\eta$ | Smoothing parameter for Eq. (2). |
| $q$ | $q = \mu/(\mu + \eta)$. |
| $\rho$ | $\rho = 0.9\sqrt{q}$ |
| $\varphi_t$ | Inner-loop stop criteria of $h_t(\hat{\boldsymbol{x}}) - h_t^* \leq \varphi_t$. |
| $\epsilon_t$ | Inner-loop stop criteria of $\|\boldsymbol{D}^{-1/2}\nabla h_t(\hat{\boldsymbol{z}})\|_\infty \leq \epsilon_t$. |
| $\mathrm{prox}_{g/\eta}(\boldsymbol{y})$ | The proximal point of $\boldsymbol{y}$, i.e., $\mathrm{prox}_{g/\eta}(\boldsymbol{y}) = \arg\min_{\boldsymbol{x}\in\mathbb{R}^n}\left\{ g(\boldsymbol{x}) + \frac{\eta}{2}\|\boldsymbol{x}-\boldsymbol{y}\|_2^2 \right\}$. |
| $C_{h_t}^i$ | $C_{h_t}^i = \|\boldsymbol{D}^{1/2}\nabla h_t(\boldsymbol{z}_t^{(i)})\|_1$. |
| $\mathcal{T}_t^{\mathcal{M}}$ | Time complexity of $\mathcal{M}$, which is characterized by $\mathcal{T}_t^{\mathcal{M}} = K_t \cdot \overline{\mathrm{vol}}(\mathcal{S}_t)$. |
| $\mathcal{T}$ | Total time complexity. |

## A.1 Proofs of Lemmas 2.1 and 2.2

**Lemma 2.1** (Properties of $\boldsymbol{\pi}$). *Define the PPR matrix $\boldsymbol{\Pi}_\alpha = \alpha \left( \frac{1+\alpha}{2}\boldsymbol{I} - \frac{1-\alpha}{2}\boldsymbol{A}\boldsymbol{D}^{-1} \right)^{-1}$. Let the estimate-residual pair $(\boldsymbol{p}, \boldsymbol{r})$ for Eq. (1) satisfy $\boldsymbol{r} = \boldsymbol{e}_s - \boldsymbol{\Pi}_\alpha^{-1}\boldsymbol{p}$. Then,*

- *The PPR vector is given by $\boldsymbol{\pi} = \boldsymbol{\Pi}_\alpha \boldsymbol{e}_s$, which is a probability distribution, i.e., $\forall i \in \mathcal{V}$, $\pi_i > 0$ and $\|\boldsymbol{\pi}\|_1 = 1$. For $\epsilon > 0$, the stop condition $\|\boldsymbol{D}^{-1}\boldsymbol{r}\|_\infty < \epsilon$ ensures $\|\boldsymbol{D}^{-1}(\boldsymbol{p} - \boldsymbol{\pi})\|_\infty < \epsilon$.*

- *The matrix $\alpha Q^{-1}$ is similar to the matrix $\Pi_\alpha$, i.e., $\alpha D^{1/2} Q^{-1} D^{-1/2} = \Pi_\alpha$. Furthermore, the $\ell_1$-norm of $\Pi_\alpha$ satisfies $\|\Pi_\alpha\|_1 = \|D^{-1}\Pi_\alpha D\|_\infty = 1$.*

*Proof.* The PPR vector is a probability distribution and is given by $\pi = \Pi_\alpha e_s$, which follows directly from the definition in Eq. (1). To verify the stopping condition, note that the residual satisfies $\Pi_\alpha r = \pi - p$, that is,

$$D^{-1}\Pi_\alpha D D^{-1} r = D^{-1}(\pi - p).$$

To meet the stop condition $\|D^{-1}(p - \pi)\|_\infty \le \epsilon$, we note

$$
\begin{aligned}
\|D^{-1}(p - \pi)\|_\infty &= \|D^{-1}\Pi_\alpha D \cdot D^{-1} r\|_\infty \\
&\le \|D^{-1}\Pi_\alpha D\|_\infty \cdot \|D^{-1} r\|_\infty \\
&= \|D^{-1} r\|_\infty \le \epsilon,
\end{aligned}
$$

where the second inequality is due to $\|D^{-1}\Pi_\alpha D\|_\infty = \|\Pi_\alpha\|_1 = 1$. To verify the second item, note that

$$
\begin{aligned}
\|D^{-1}\Pi_\alpha D\|_\infty &= \left\| \alpha \left( \frac{1+\alpha}{2} I - \frac{1-\alpha}{2} D^{-1} A \right)^{-1} \right\|_\infty \\
&= \left\| \alpha \left( \left( \frac{1+\alpha}{2} I - \frac{1-\alpha}{2} D^{-1} A \right)^{-1} \right)^\top \right\|_1 = \|\Pi_\alpha\|_1 = 1.
\end{aligned}
$$

$\square$

**Lemma 2.2** (Properties of $x_f^*$ and $x_\psi^*$). *Denote the gradient of $f$ at $x$ as $\nabla f(x) := Qx - \alpha D^{-1/2} b$ and optimal solution $x_f^* = \alpha Q^{-1} D^{-1/2} b$ satisfying $\pi := D^{1/2} x_f^*$. Define $p = D^{1/2} x$. Then,*

- *The stop condition $\|D^{-1/2}\nabla f(x)\|_\infty < \alpha\epsilon$ implies $\|D^{-1}(p - \pi)\|_\infty < \epsilon$.*

- *The objective $f$ is $\mu$-strongly convex and $L$-smooth with two constants $\mu = \alpha$ and $L = 1$. When $\hat{\epsilon} = \epsilon$, then $x_\psi^* \in \mathcal{P}(\epsilon, \alpha, e_s, \mathcal{G})$ and the solution is sparse, i.e., $|\operatorname{supp}(x_\psi^*)| \le 1/\hat{\epsilon}$.*

*Proof.* For item 1, note $x_f^* = \alpha Q^{-1} D^{-1/2} b = D^{-1/2}\Pi_\alpha D^{1/2} D^{-1/2} e_s = D^{-1/2}\pi$. Hence, $\pi = D^{1/2} x_f^*$. With $\nabla f(x) = Qx - \alpha D^{-1/2} b$, we have

$$
\begin{aligned}
\|D^{-1}(p - \pi)\|_\infty &= \|D^{-1/2}(x - x_f^*)\|_\infty \\
&= \|D^{-1/2}(Q^{-1}\nabla f(x))\|_\infty \\
&= \|D^{-1/2} Q^{-1} D^{1/2} D^{-1/2} \nabla f(x)\|_\infty \\
&= \frac{1}{\alpha} \|D^{-1}\Pi_\alpha D D^{-1/2}\nabla f(x)\|_\infty \\
&\le \frac{1}{\alpha} \|D^{-1}\Pi_\alpha D\|_\infty \cdot \|D^{-1/2}\nabla f(x)\|_\infty \\
&= \frac{1}{\alpha} \|D^{-1/2}\nabla f(x)\|_\infty < \epsilon.
\end{aligned}
$$

For item 2, the Hessian of $f$ is $H_f = Q$ and the eigenvalues of $\lambda(AD^{-1})$ satisfy $\lambda(AD^{-1}) \in (-1, 1]$, so $\lambda(Q) \in [\alpha, 1]$. When $\hat{\epsilon} = \epsilon$, then $x_\psi^* \in \mathcal{P}(\epsilon, \alpha, e_s, \mathcal{G})$, which follows from the result of Fountoulakis et al. [20]. $\square$

The following lemma provides useful results that will be useful in our later proofs.

**Lemma A.1** (Gradient Descent for $(\mu, L)$-convex $f$ [17, 48]). *Let $f$ be $\mu$-strongly convex and $L$-smooth. Consider the gradient descent update*

$$x^{(t+1)} = x^{(t)} - \frac{2}{\mu + L}\nabla f(x^{(t)})$$

*Then, the following properties hold*

- *Bounds on initial function error, $\frac{\mu}{2}\|\boldsymbol{x}^{(0)} - \boldsymbol{x}^*\|_2^2 \le f(\boldsymbol{x}^{(0)}) - f(\boldsymbol{x}^*) \le \frac{L}{2}\|\boldsymbol{x}^{(0)} - \boldsymbol{x}^*\|_2^2$.*

- *Gradient norm bounds on initial gradient,*

$$\frac{1}{2L}\|\nabla f(\boldsymbol{x}^{(0)}) - \nabla f(\boldsymbol{x}^*)\|_2^2 \le f(\boldsymbol{x}^{(0)}) - f(\boldsymbol{x}^*) \le \frac{1}{2\mu}\|\nabla f(\boldsymbol{x}^{(0)}) - \nabla f(\boldsymbol{x}^*)\|_2^2.$$

- *Per-iteration reduction in function error, $f(\boldsymbol{x}^{(t+1)}) - f^* \le \left(\frac{L-\mu}{L+\mu}\right)^2 (f(\boldsymbol{x}^{(t)}) - f^*)$.*

- *Per-iteration reduction in estimation error, $\|\boldsymbol{x}^{(t+1)} - \boldsymbol{x}^*\|_2^2 \le \left(\frac{L-\mu}{L+\mu}\right)^2 \|\boldsymbol{x}^{(t)} - \boldsymbol{x}^*\|_2^2$.*

## A.2 The proof of Lemma 3.2

**Lemma 3.2.** *Let $h_t$ be defined in Eq. (6), and suppose that the initial point $\boldsymbol{z}_t^{(0)}$ of $t$-th process satisfies $\nabla h_t(\boldsymbol{z}_t^{(0)}) \neq \boldsymbol{0}$. If there exists a local algorithm $\mathcal{M}$ such that $\|\nabla h_t^{1/2}(\boldsymbol{z}_t^{(K_t)})\|_1 < \|\nabla h_t^{1/2}(\boldsymbol{z}_t^{(0)})\|_1$, then for a stopping condition $\|\nabla h_t^{-1/2}(\boldsymbol{z}_t^{(k)})\|_\infty < \epsilon_t$ of $\mathcal{M}$ with*

$$\epsilon_t \triangleq \max\left\{ \sqrt{\frac{(\mu+\eta)\varphi_t}{m}}, \frac{2(\eta+\alpha)\varphi_t}{\|\nabla h_t^{1/2}(\boldsymbol{z}_t^{(0)})\|_1} \right\}, \text{ where } \varphi_t > 0,$$

*the final solution $\boldsymbol{z}_t^{(K_t)}$ is guaranteed in the ball, i.e., $\boldsymbol{z}_t^{(K_t)} \in \mathcal{H}_t(\varphi_t)$ as defined in* (C1).

*Proof.* The proximal objective $h_t$ defined in Eq. (6) at $t$-th iteration can be expanded as

$$h_t(\boldsymbol{z}) = \frac{1}{2}\boldsymbol{z}^\top \left( \frac{1+\alpha+2\eta}{2}\boldsymbol{I} - \frac{1-\alpha}{2}\boldsymbol{D}^{-1/2}\boldsymbol{A}\boldsymbol{D}^{-1/2} \right) \boldsymbol{z} - \boldsymbol{z}^\top \left( \alpha\boldsymbol{D}^{-1/2}\boldsymbol{b} + \eta\boldsymbol{y}^{(t-1)} \right) + \frac{\eta}{2}\|\boldsymbol{y}^{(t-1)}\|_2^2. \tag{15}$$

Let $\tilde{\boldsymbol{Q}} = \frac{1+\alpha+2\eta}{2}\boldsymbol{I} - \frac{1-\alpha}{2}\boldsymbol{D}^{-1/2}\boldsymbol{A}\boldsymbol{D}^{-1/2}$ and $\boldsymbol{b}^{(t-1)} = \alpha\boldsymbol{D}^{-1/2}\boldsymbol{b} + \eta\boldsymbol{y}^{(t-1)}$, then

$$\tilde{\boldsymbol{Q}}\boldsymbol{z} = \boldsymbol{b}^{(t-1)}, \quad \tilde{\boldsymbol{Q}} \triangleq \frac{1+\alpha+2\eta}{2}\boldsymbol{I} - \frac{1-\alpha}{2}\boldsymbol{D}^{-1/2}\boldsymbol{A}\boldsymbol{D}^{-1/2}, \quad \boldsymbol{b}^{(t-1)} = \alpha\boldsymbol{D}^{-1/2}\boldsymbol{b} + \eta\boldsymbol{y}^{(t-1)}. \tag{16}$$

We know the optimal solution $\boldsymbol{x}_t^* = \tilde{\boldsymbol{Q}}^{-1}\boldsymbol{b}^{(t-1)}$. Note that the objective error can be rewritten as

$$\begin{aligned}
h_t(\boldsymbol{z}) - h_t(\boldsymbol{x}_t^*) &= \frac{1}{2}\boldsymbol{z}^\top \tilde{\boldsymbol{Q}}\boldsymbol{z} - \boldsymbol{z}^\top \boldsymbol{b}^{(t-1)} - \left( \frac{1}{2}\boldsymbol{x}_t^{*\top}\tilde{\boldsymbol{Q}}\boldsymbol{x}_t^* - \boldsymbol{x}_t^{*\top}\boldsymbol{b}^{(t-1)} \right) \\
&= \frac{1}{2}\boldsymbol{z}^\top \tilde{\boldsymbol{Q}}\boldsymbol{z} - \boldsymbol{z}^\top \boldsymbol{b}^{(t-1)} - \left( \frac{1}{2}\boldsymbol{x}_t^{*\top}\tilde{\boldsymbol{Q}}\boldsymbol{x}_t^* - \boldsymbol{x}_t^{*\top}\tilde{\boldsymbol{Q}}\boldsymbol{x}_t^* \right) \\
&= \frac{1}{2}\boldsymbol{z}^\top \tilde{\boldsymbol{Q}}\boldsymbol{z} - \boldsymbol{z}^\top \boldsymbol{b}^{(t-1)} + \frac{1}{2}\boldsymbol{x}_t^{*\top}\tilde{\boldsymbol{Q}}\boldsymbol{x}_t^* \\
&= \frac{1}{2}\boldsymbol{z}^\top \tilde{\boldsymbol{Q}}\boldsymbol{z} - \frac{1}{2}\boldsymbol{z}^\top \tilde{\boldsymbol{Q}}\boldsymbol{x}_t^* - \frac{1}{2}\boldsymbol{x}_t^{*\top}\tilde{\boldsymbol{Q}}\boldsymbol{z} + \frac{1}{2}\boldsymbol{x}_t^{*\top}\tilde{\boldsymbol{Q}}\boldsymbol{x}_t^* \\
&= \frac{1}{2}(\boldsymbol{z} - \boldsymbol{x}_t^*)^\top \tilde{\boldsymbol{Q}}(\boldsymbol{z} - \boldsymbol{x}_t^*).
\end{aligned}$$

Since $\boldsymbol{z}_t^{(K_t)} - \boldsymbol{x}_t^* = \tilde{\boldsymbol{Q}}^{-1}\nabla h_t(\boldsymbol{z}_t^{(K_t)})$, we show $h_t(\boldsymbol{z}_t^{(K_t)}) - h_t^*$ can be rewritten in terms of $\nabla h_t(\boldsymbol{z}_t^{(K_t)})$

$$\begin{aligned}
h_t(\boldsymbol{z}_t^{(K_t)}) - h_t(\boldsymbol{x}_t^*) &= \frac{1}{2}(\boldsymbol{z}_t^{(K_t)} - \boldsymbol{x}_t^*)^\top \tilde{\boldsymbol{Q}}(\boldsymbol{z}_t^{(K_t)} - \boldsymbol{x}_t^*) \\
&= \frac{1}{2}\nabla h_t(\boldsymbol{z}_t^{(K_t)})^\top \tilde{\boldsymbol{Q}}^{-1}\tilde{\boldsymbol{Q}}\tilde{\boldsymbol{Q}}^{-1}\nabla h_t(\boldsymbol{z}_t^{(K_t)}) \\
&= \frac{1}{2}\nabla h_t(\boldsymbol{z}_t^{(K_t)})^\top \tilde{\boldsymbol{Q}}^{-1}\nabla h_t(\boldsymbol{z}_t^{(K_t)}) \\
&= \frac{1}{2(\eta+\alpha)}\nabla h_t(\boldsymbol{z}_t^{(K_t)})^\top \boldsymbol{D}^{-1/2}\boldsymbol{\Pi}_{\frac{\eta+\alpha}{1+\eta}}\boldsymbol{D}^{1/2}\nabla h_t(\boldsymbol{z}_t^{(K_t)})
\end{aligned}$$

$$= \frac{1}{2(\eta + \alpha)} (\boldsymbol{D}^{-1/2} \nabla h_t(\boldsymbol{z}_t^{(K_t)}))^\top \boldsymbol{\Pi}_{\frac{\eta+\alpha}{1+\eta}} \boldsymbol{D}^{1/2} \nabla h_t(\boldsymbol{z}_t^{(K_t)}),$$

where the fourth equality is due to the identity $\boldsymbol{D}^{-1/2} \boldsymbol{\Pi}_{\frac{\eta+\alpha}{1+\eta}} \boldsymbol{D}^{1/2} = (\eta + \alpha) \tilde{\boldsymbol{Q}}^{-1}$. By Hölder's inequality, we have

$$
\begin{aligned}
h_t(\boldsymbol{z}_t^{(K_t)}) - h_t(\boldsymbol{x}_t^*) &\leq \frac{1}{2(\eta + \alpha)} \| \boldsymbol{D}^{-1/2} \nabla h_t(\boldsymbol{z}_t^{(K_t)}) \|_\infty \cdot \| \boldsymbol{\Pi}_{\frac{\eta+\alpha}{1+\eta}} \boldsymbol{D}^{1/2} \nabla h_t(\boldsymbol{z}_t^{(K_t)}) \|_1 \\
&\leq \frac{1}{2(\eta + \alpha)} \| \boldsymbol{D}^{-1/2} \nabla h_t(\boldsymbol{z}_t^{(K_t)}) \|_\infty \cdot \| \boldsymbol{D}^{1/2} \nabla h_t(\boldsymbol{z}_t^{(K_t)}) \|_1 \\
&\leq \frac{1}{2(\eta + \alpha)} \| \boldsymbol{D}^{-1/2} \nabla h_t(\boldsymbol{z}_t^{(K_t)}) \|_\infty \cdot \| \boldsymbol{D}^{1/2} \nabla h_t(\boldsymbol{z}_t^{(0)}) \|_1 \leq \varphi_t,
\end{aligned}
$$

where the last inequality gives the second part of $\epsilon_t$. On the other hand, we know

$$
\begin{aligned}
h_t(\boldsymbol{z}_t^{(K_t)}) - h_t(\boldsymbol{x}_t^*) &\leq \frac{1}{2(\eta + \alpha)} \| \boldsymbol{D}^{-1/2} \nabla h_t(\boldsymbol{z}_t^{(K_t)}) \|_\infty \cdot \| \boldsymbol{D}^{1/2} \nabla h_t(\boldsymbol{z}_t^{(K_t)}) \|_1 \\
&\leq \frac{1}{2(\eta + \alpha)} \| \boldsymbol{D}^{-1/2} \nabla h_t(\boldsymbol{z}_t^{(K_t)}) \|_\infty \cdot \| \boldsymbol{D} \boldsymbol{D}^{-1/2} \nabla h_t(\boldsymbol{z}_t^{(K_t)}) \|_1 \\
&\leq \frac{1}{2(\eta + \alpha)} \operatorname{vol}(\operatorname{supp}(\nabla h_t(\boldsymbol{z}_t^{(K_t)}))) \| \boldsymbol{D}^{-1/2} \nabla h_t(\boldsymbol{z}_t^{(K_t)}) \|_\infty^2 \\
&\leq \frac{m}{\eta + \alpha} \| \boldsymbol{D}^{-1/2} \nabla h_t(\boldsymbol{z}_t^{(K_t)}) \|_\infty^2 \leq \varphi_t,
\end{aligned}
$$

which gives the first part of $\epsilon_t$. $\qquad \square$

*Remark* A.2. Choosing a starting point $\boldsymbol{z}_t^{(0)}$ is straightforward. For example, if $\boldsymbol{z}_t^{(0)} = \boldsymbol{0}$, then $\nabla h_t(\boldsymbol{z}_t^{(0)}) = \boldsymbol{b}^{(t-1)}$. Consequently, the following stopping condition is sufficient:

$$\| \boldsymbol{D}^{-1/2} \nabla h_t(\boldsymbol{z}_t^{(K_t)}) \|_\infty \leq \epsilon_t := \max \left\{ \sqrt{\frac{(\mu + \eta)\varphi_t}{m}}, \frac{2(\eta + \alpha)\varphi_t}{\| \boldsymbol{D}^{1/2} \boldsymbol{b}^{(t-1)} \|_1} \right\}.$$

### A.3 Proofs of Theorems 3.3 and A.3

At each iteration $t$ of the AESP framework in Algorithm 1, we solve the inexact proximal operator $\boldsymbol{x}^{(t)} \approx \operatorname{prox}_{f/\eta}(\boldsymbol{y}^{(t-1)})$, as defined in (2), where $f$ is given in (P1). The following theorem establishes the convergence properties of LOCGD.

**Theorem 3.3** (**Convergence of LOCGD**)**.** *Let $h_t$ be defined in Eq. (6). LOCGD (Algorithm 3) is used to minimize $h_t(\boldsymbol{z})$ and returns $\boldsymbol{z}_t^{(K_t)} = \operatorname{LOCGD}(\varphi_t, \boldsymbol{y}^{(t-1)}, \eta, \alpha, \boldsymbol{b}, \mathcal{G}) \in \mathcal{H}_t(\varphi_t)$. Recall the $\boldsymbol{D}^{1/2}$-scaled gradient $\nabla h_t^{1/2}(\boldsymbol{z}_t^{(k)}) := \boldsymbol{D}^{1/2} \nabla h_t(\boldsymbol{z}_t^{(k)})$. For $k \geq 0$, the scaled gradient satisfies*

$$\big\| \nabla h_t^{1/2}(\boldsymbol{z}_t^{(k+1)}) \big\|_1 \leq \big(1 - \tau \gamma_t^{(k)}\big) \big\| \nabla h_t^{1/2}(\boldsymbol{z}_t^{(k)}) \big\|_1,$$

*where $\tau := \frac{2(\alpha + \eta)}{1 + \alpha + 2\eta}$ and $\gamma_t^{(k)}$ is the ratio defined in Eq. (10). Assume $\epsilon_t$ and stop condition are defined in Eq. (8) of Lemma 3.2, then the run time $\mathcal{T}_t^{\text{LOCGD}}$, as defined in Eq. (4), is bounded by*

$$\mathcal{T}_t^{\text{LOCGD}} \leq \min \left\{ \frac{\overline{\operatorname{vol}}(\mathcal{S}_t)}{\tau \overline{\gamma}_t} \log \frac{C_{h_t}^0}{C_{h_t}^{K_t}}, \frac{C_{h_t}^0 - C_{h_t}^{K_t}}{\tau \epsilon_t} \right\},$$

*where $C_{h_t}^i = \| \nabla h_t^{1/2}(\boldsymbol{z}_t^{(i)}) \|_1$ denote constants. Furthermore, $\overline{\operatorname{vol}}(\mathcal{S}_t) / \overline{\gamma}_t \leq \min \{ C_{h_t}^0 / \epsilon_t, 2m \}$.*

*Proof.* Recall that $h_t(\boldsymbol{z})$ is defined in Eq. (15). Minimizing $h_t(\boldsymbol{z})$ is equivalent to solving the linear system $\tilde{\boldsymbol{Q}} \boldsymbol{z} = \boldsymbol{b}^{(t-1)}$, as defined in Eq. (16). Using the iteration of *local gradient descent* in Eq. (9), and noting that $h_t(\boldsymbol{z})$ is $(\mu + \eta)$-strongly convex and $(L + \eta)$-smooth, the optimal step size is given

by $2/(\mu + L + 2\eta)$, as established in Lemma A.1. The updated gradient at iteration $(t + 1)$ is then computed as follows.

$$
\begin{aligned}
\nabla h_t(\boldsymbol{z}_t^{(k+1)}) &= \tilde{\boldsymbol{Q}} \boldsymbol{z}_t^{(k+1)} - \boldsymbol{b}^{(t-1)} \\
&= \tilde{\boldsymbol{Q}} \left( \boldsymbol{z}_t^{(k)} - \frac{2}{1 + \alpha + 2\eta} \nabla h_t(\boldsymbol{z}_t^{(k)}) \circ \mathbf{1}_{\mathcal{S}_t^{(k)}} \right) - \boldsymbol{b}^{(t-1)} \\
&= \nabla h_t(\boldsymbol{z}_t^{(k)}) - \frac{2}{1 + \alpha + 2\eta} \tilde{\boldsymbol{Q}} \nabla h_t(\boldsymbol{z}_t^{(k)}) \circ \mathbf{1}_{\mathcal{S}_t^{(k)}}.
\end{aligned}
$$

For simplicity, recall that we denote the normalized gradient $\nabla h_t^{1/2}(\boldsymbol{z}_t^{(k)}) = \boldsymbol{D}^{1/2} \nabla h_t(\boldsymbol{z}_t^{(k)})$, then we continue to have

$$
\begin{aligned}
\left\| \nabla h_t^{1/2}(\boldsymbol{z}_t^{(k+1)}) \right\|_1 &= \left\| \nabla h_t^{1/2}(\boldsymbol{z}_t^{(k)}) - \left( \boldsymbol{I} - \frac{1 - \alpha}{1 + \alpha + 2\eta} \boldsymbol{A}\boldsymbol{D}^{-1} \right) \nabla h_t^{1/2}(\boldsymbol{z}_t^{(k)}) \circ \mathbf{1}_{\mathcal{S}_t^{(k)}} \right\|_1 \\
&\leq \| \nabla h_t^{1/2}(\boldsymbol{z}_t^{(k)}) - \nabla h_t^{1/2}(\boldsymbol{z}_t^{(k)}) \circ \mathbf{1}_{\mathcal{S}_t^{(k)}} \|_1 + \left\| \frac{1 - \alpha}{1 + \alpha + 2\eta} \boldsymbol{A}\boldsymbol{D}^{-1} \nabla h_t^{1/2}(\boldsymbol{z}_t^{(k)}) \circ \mathbf{1}_{\mathcal{S}_t^{(k)}} \right\|_1 \\
&\leq \| \nabla h_t^{1/2}(\boldsymbol{z}_t^{(k)}) \|_1 - \| \nabla h_t^{1/2}(\boldsymbol{z}_t^{(k)}) \circ \mathbf{1}_{\mathcal{S}_t^{(k)}} \|_1 + \frac{1 - \alpha}{1 + \alpha + 2\eta} \| \nabla h_t^{1/2}(\boldsymbol{z}_t^{(k)}) \circ \mathbf{1}_{\mathcal{S}_t^{(k)}} \|_1 \\
&= \| \nabla h_t^{1/2}(\boldsymbol{z}_t^{(k)}) \|_1 - \frac{2\alpha + 2\eta}{1 + \alpha + 2\eta} \| \nabla h_t^{1/2}(\boldsymbol{z}_t^{(k)}) \circ \mathbf{1}_{\mathcal{S}_t^{(k)}} \|_1. \quad (17)
\end{aligned}
$$

Therefore, associated with the gradient reduction ratio defined in Eq. (10), we obtain

$$
\| \nabla h_t^{1/2}(\boldsymbol{z}_t^{(k+1)}) \|_1 \leq \left( 1 - \frac{2(\alpha + \eta)}{1 + \alpha + 2\eta} \gamma_t^{(k)} \right) \| \nabla h_t^{1/2}(\boldsymbol{z}_t^{(k)}) \|_1.
$$

For any $K \geq 1$, the above per-iteration reduction gives us

$$
\| \boldsymbol{D}^{1/2} \nabla h_t(\boldsymbol{z}_t^{(K)}) \|_1 \leq \prod_{k=0}^{K-1} \left( 1 - \frac{2(\alpha + \eta)}{1 + \alpha + 2\eta} \gamma_t^{(k)} \right) \| \boldsymbol{D}^{1/2} \nabla h_t(\boldsymbol{z}_t^{(0)}) \|_1.
$$

Specifically, let $K_t$ be the total number of iterations of LOCGD called from Local-Catalyst at $t$-th iteration using the precision $\epsilon_t$. Denote that $\bar{\gamma}_t = \frac{1}{K_t} \sum_{k=0}^{K_t - 1} \gamma_t^{(k)}$, we can obtain an upper bound of $K_t$ as the following

$$
\log \frac{\| \boldsymbol{D}^{1/2} \nabla h_t(\boldsymbol{z}_t^{(K_t)}) \|_1}{\| \boldsymbol{D}^{1/2} \nabla h_t(\boldsymbol{z}_t^{(0)}) \|_1} \leq \sum_{k=0}^{K_t - 1} \log \left( 1 - \frac{2(\alpha + \eta)}{1 + \alpha + 2\eta} \gamma_t^{(k)} \right) \leq - \sum_{k=0}^{K_t - 1} \frac{2(\alpha + \eta)}{1 + \alpha + 2\eta} \gamma_t^{(k)}.
$$

The above inequality implies $K_t \leq \frac{1 + \alpha + 2\eta}{2(\alpha + \eta)\bar{\gamma}_t} \log \frac{\| \boldsymbol{D}^{1/2} \nabla h_t(\boldsymbol{z}_t^{(0)}) \|_1}{\| \boldsymbol{D}^{1/2} \nabla h_t(\boldsymbol{z}_t^{(K_t)}) \|_1}$. Therefore, we have the time complexity as

$$
\sum_{k=0}^{K_t - 1} \text{vol}(\mathcal{S}_t^{(k)}) = K_t \cdot \overline{\text{vol}}(\mathcal{S}_t) \leq \frac{(1 + \alpha + 2\eta)\overline{\text{vol}}(\mathcal{S}_t)}{2(\alpha + \eta)\bar{\gamma}_t} \log \frac{\| \boldsymbol{D}^{1/2} \nabla h_t(\boldsymbol{z}_t^{(0)}) \|_1}{\| \boldsymbol{D}^{1/2} \nabla h_t(\boldsymbol{z}_t^{(K_t)}) \|_1}.
$$

On the other hand, from inequality of inequality (17), we know that

$$
\begin{aligned}
\frac{2\alpha + 2\eta}{1 + \alpha + 2\eta} \cdot \epsilon_t \cdot \text{vol}(\mathcal{S}_t^{(k)}) &\leq \frac{2\alpha + 2\eta}{1 + \alpha + 2\eta} \| \nabla h_t^{1/2}(\boldsymbol{z}_t^{(k)}) \circ \mathbf{1}_{\mathcal{S}_t^{(k)}} \|_1 \\
&\leq \| \nabla h_t^{1/2}(\boldsymbol{z}_t^{(k)}) \|_1 - \| \nabla h_t^{1/2}(\boldsymbol{z}_t^{(k+1)}) \|_1.
\end{aligned}
$$

The total runtime can also be bounded as

$$
\begin{aligned}
\mathcal{T}_t^{\text{LOCGD}} = \sum_{k=0}^{K_t - 1} \text{vol}(\mathcal{S}_t^{(k)}) &\leq \frac{1 + \alpha + 2\eta}{2(\alpha + \eta)\epsilon_t} \sum_{k=0}^{K_t - 1} \left( \| \nabla h_t^{1/2}(\boldsymbol{z}_t^{(k)}) \|_1 - \| \nabla h_t^{1/2}(\boldsymbol{z}_t^{(k+1)}) \|_1 \right) \\
&= \frac{1 + \alpha + 2\eta}{2(\alpha + \eta)\epsilon_t} \left( \| \nabla h_t^{1/2}(\boldsymbol{z}_t^{(0)}) \|_1 - \| \nabla h_t^{1/2}(\boldsymbol{z}_t^{(K_t)}) \|_1 \right).
\end{aligned}
$$

Combining the two bounds above, we establish the first part of the theorem. To verify that the ratio serves as a lower bound for $\|\nabla h_t^{1/2}(z_t^{(0)})\|_1/\epsilon_t$, note that for any $u_i \in \mathcal{S}_t^{(k)}$, we have $\nabla h_t(z_t^{(k)})u_i > \epsilon_t\sqrt{d_{u_i}}$. This further leads to

$$\epsilon_t \operatorname{vol}(\mathcal{S}_t^{(k)}) = \epsilon_t \sum_{i=1}^{|\mathcal{S}_t^{(k)}|} d_{u_i} < \sum_{i=1}^{|\mathcal{S}_t^{(k)}|} \left|\sqrt{d_{u_i}}\nabla_{u_i} h_t(z_t^{(k)})\right| = \gamma_k \|\boldsymbol{D}^{1/2}\nabla h_t(z_t^{(k)})\|_1.$$

Since $\|\boldsymbol{D}^{1/2}\nabla h_t(z_t^{(0)})\|_1 \geq \|\boldsymbol{D}^{1/2}\nabla h_t(z_t^{(1)})\|_1 \geq \cdots \geq \|\boldsymbol{D}^{1/2}\nabla h_t(z_t^{(K_t)})\|_1$, this leads to

$$\epsilon_t \frac{1}{K_t} \sum_{k=0}^{K_t-1} \operatorname{vol}(\mathcal{S}_t^{(k)}) \leq \frac{1}{K_t} \sum_{k=0}^{K_t-1} \gamma_t^{(k)} \|\boldsymbol{D}^{1/2}\nabla h_t(z_t^{(k)})\|_1 \leq \frac{1}{K_t} \sum_{k=0}^{K_t-1} \gamma_t^{(k)} \|\boldsymbol{D}^{1/2}\nabla h_t(z_t^{(0)})\|_1$$

$$\Rightarrow \quad \frac{\overline{\operatorname{vol}}(\mathcal{S}_t)}{\overline{\gamma}_t} < \frac{\|\boldsymbol{D}^{1/2}\nabla h_t(z_t^{(0)})\|_1}{\epsilon_t}.$$

On the other hand, $\frac{\overline{\operatorname{vol}}(\mathcal{S}_t)}{\overline{\gamma}_t} \leq 2m$. To see this, by Eq. (10), note for each iteration $k$,

$$\gamma_t^{(k)} \triangleq \frac{\|\nabla h_t^{1/2}(z_t^{(k)}) \circ \mathbf{1}_{\mathcal{S}_t^{(k)}}\|_1}{\|\nabla h_t^{1/2}(z_t^{(k)})\|_1}$$

For $u \in \mathcal{S}_t^{(k)}$, we have $|\nabla_u h_t^{1/2}(z_t^{(k)})| \geq \epsilon_t d_u$ and for $v \in \mathcal{V}\backslash\mathcal{S}_t^{(k)}$, we have $|\nabla_v h_t^{1/2}(z_t^{(k)})| \geq \epsilon_t d_v$, which means

$$\frac{|\nabla_u h_t^{1/2}(z_t^{(k)})|}{d_u} \geq \epsilon_t \geq \frac{|\nabla_v h_t^{1/2}(z_t^{(k)})|}{d_v}$$

$$\Rightarrow \quad \frac{\|\nabla h_t^{1/2}(z_t^{(k)}) \circ \mathbf{1}_{\mathcal{S}_t^{(k)}}\|_1}{\operatorname{vol}(\mathcal{S}_t^{(k)})} \geq \epsilon_t \geq \frac{\|\nabla h_t^{1/2}(z_t^{(k)}) \circ \mathbf{1}_{\mathcal{V}\backslash\mathcal{S}_t^{(k)}}\|_1}{\operatorname{vol}(\mathcal{V}\backslash \operatorname{vol}(\mathcal{S}_t^{(k)}))}.$$

Given $a, b, c, d > 0$ and $\frac{a}{b} > \frac{c}{d}$, we have $\frac{a}{b} > \frac{a+c}{b+d}$. Then,

$$\frac{\|\nabla h_t^{1/2}(z_t^{(k)}) \circ \mathbf{1}_{\mathcal{S}_t^{(k)}}\|_1}{\operatorname{vol}(\mathcal{S}_t^{(k)})} \geq \frac{\|\nabla h_t^{1/2}(z_t^{(k)}) \circ \mathbf{1}_{\mathcal{V}\backslash\mathcal{S}_t^{(k)}}\|_1 + \|\nabla h_t^{1/2}(z_t^{(k)}) \circ \mathbf{1}_{\mathcal{S}_t^{(k)}}\|_1}{\operatorname{vol}(\mathcal{V}\backslash \operatorname{vol}(\mathcal{S}_t^{(k)})) + \operatorname{vol}(\mathcal{S}_t^{(k)})}$$

$$= \frac{\|\nabla h_t^{1/2}(z_t^{(k)})\|_1}{\operatorname{vol}(\mathcal{V})} = \frac{\|\nabla h_t^{1/2}(z_t^{(k)})\|_1}{2m}$$

$$\Rightarrow \quad \frac{\overline{\operatorname{vol}}(\mathcal{S}_t)}{\overline{\gamma}_t} \leq 2m.$$

Hence, we prove two upper bounds of $\frac{\overline{\operatorname{vol}}(\mathcal{S}_t)}{\overline{\gamma}_t}$.  $\qquad\qquad\square$

The following theorem establishes the convergence and time complexity of LOCAPPR. We first define a similar active node ratio for LOCAPPR as the following

$$\overline{\gamma}_t \triangleq \frac{1}{K_t} \sum_{k=0}^{K_t-1} \sum_{i=1}^{|\mathcal{S}_t^{(k)}|} \left\{\gamma_t^{(k_i)} \triangleq \frac{\|\nabla h_t^{1/2}(z_t^{(k_i)}) \circ \mathbf{1}_{\{u_i\}}\|_1}{\|\nabla h_t^{1/2}(z_t^{(k_i)})\|_1}\right\}, \tag{18}$$

where $k_i = k + (i-1)/|\mathcal{S}_t^{(k)}|$ for $i = 1, 2, \ldots, |\mathcal{S}_t^{(k)}|$.

**Theorem A.3** (The convergence and time complexity of LOCAPPR). *Let $h_t$ be defined in Eq. (6). The LOCAPPR algorithm, implemented as in Algorithm 4, is used to minimize $h_t(z)$. Recall the $\boldsymbol{D}^{1/2}$-scaled gradient $\nabla h_t^{1/2}(z_t^{(k)}) := \boldsymbol{D}^{1/2}\nabla h_t(z_t^{(k)})$. For $k \geq 0$, the scaled gradient satisfies the following reduction property:*

$$\left\|\nabla h_t^{1/2}(z_t^{(k+1)})\right\|_1 \leq \left(1 - \tau \sum_{i=1}^{|\mathcal{S}_t^{(k)}|} \gamma_t^{(k_i)}\right) \left\|\nabla h_t^{1/2}(z_t^{(k)})\right\|_1,$$

*where the constant* $\tau := \frac{2(\alpha+\eta)}{1+\alpha+2\eta}$ *and* $\gamma_t^{(k_i)}$ *is the active node ratio defined in Eq. (18). Assume the precision* $\epsilon_t$ *and stop condition is defined in (8) of Lemma 3.2 , then the returned satisfies*

$$z_t^{(K_t)} = \text{LOCAPPR}(\varphi_t, \eta, y^{(t-1)}, \alpha, b, \mathcal{G}) \in \mathcal{H}_t(\varphi_t).$$

*The time complexity* $\mathcal{T}_t^{\text{LOCAPPR}}$*, as defined in Eq. (4), is bounded by*

$$\mathcal{T}_t^{\text{LOCAPPR}} \le \min\left\{ \frac{\overline{\text{vol}}(\mathcal{S}_t)}{\tau \overline{\gamma}_t} \log \frac{C_{h_t}^0}{C_{h_t}^{K_t}}, \frac{C_{h_t}^0 - C_{h_t}^{K_t}}{\tau \epsilon_t} \right\},$$

*where* $C_{h_t}^i = \|\nabla h_t^{1/2}(z_t^{(i)})\|_1$ *denote constants at iteration t. Furthermore,* $\overline{\text{vol}}(\mathcal{S}_t)/\overline{\gamma}_t$ *has the following upper bound*

$$\frac{\overline{\text{vol}}(\mathcal{S}_t)}{\overline{\gamma}_t} \le \min\left\{ \frac{C_{h_t}^0}{\epsilon_t}, 2m \right\}.$$

*Proof.* Recall that $u_i \in \mathcal{S}_t^{(k)} = \{u_1, \ldots, u_{|\mathcal{S}_t|}\}$ and $k_i = k + (i-1)/|\mathcal{S}_t^{(k)}|$ for $i = 1, 2, \ldots, |\mathcal{S}_t^{(k)}|$. The LOCSOR algorithm in Algorithm 4 updates as follows: $z_t^{(k_{i+1})} = z_t^{(k_i)} - 2\nabla h_t(z_t^{(k_i)}) \circ \mathbf{1}_{\{u_i\}}/(1 + \alpha + 2\eta)$. Then, the gradient is updated as

$$\nabla h_t(z_t^{(k_{i+1})}) = \tilde{Q}z_t^{(k_{i+1})} - b^{(t-1)} = \tilde{Q}\left( z_t^{(k_i)} - \frac{2\nabla h_t(z_t^{(k_i)}) \circ \mathbf{1}_{\{u_i\}}}{1 + \alpha + 2\eta} \right) - b^{(t-1)}$$

$$= \nabla h_t(z_t^{(k_i)}) - \frac{2\tilde{Q}\nabla h_t(z_t^{(k_i)}) \circ \mathbf{1}_{\{u_i\}}}{1 + \alpha + 2\eta}.$$

Then, for all $i = 1, 2, \ldots, |\mathcal{S}_t^{(k)}|$, following similar steps as in the proof of Theorem 3.3, we have

$$\left\| \nabla h_t^{1/2}(z_t^{(k_{i+1})}) \right\|_1 = \left\| \nabla h_t^{1/2}(z_t^{(k_i)}) - \left( I - \frac{1-\alpha}{1+\alpha+2\eta} AD^{-1} \right) \nabla h_t^{1/2}(z_t^{(k_i)}) \circ \mathbf{1}_{\{u_i\}} \right\|_1$$

$$\le \|\nabla h_t^{1/2}(z_t^{(k_i)}) - \nabla h_t^{1/2}(z_t^{(k_i)}) \circ \mathbf{1}_{\{u_i\}}\|_1 + \left\| \frac{1-\alpha}{1+\alpha+2\eta} AD^{-1}\nabla h_t^{1/2}(z_t^{(k_i)}) \circ \mathbf{1}_{\{u_i\}} \right\|_1$$

$$\le \|\nabla h_t^{1/2}(z_t^{(k_i)})\|_1 - \|\nabla h_t^{1/2}(z_t^{(k_i)}) \circ \mathbf{1}_{\{u_i\}}\|_1 + \frac{1-\alpha}{1+\alpha+2\eta}\|\nabla h_t^{1/2}(z_t^{(k_i)}) \circ \mathbf{1}_{\{u_i\}}\|_1$$

$$= \|\nabla h_t^{1/2}(z_t^{(k_i)})\|_1 - \frac{2\alpha+2\eta}{1+\alpha+2\eta}\|\nabla h_t^{1/2}(z_t^{(k_i)}) \circ \mathbf{1}_{\{u_i\}}\|_1. \tag{19}$$

Recall the definition of gradient reduction ratio for active nodes is in 18. Summing over the above equations over $u_i$, we have

$$\left\| \nabla h_t^{1/2}(z_t^{(k_{i+1})}) \right\|_1 \le \|\nabla h_t^{1/2}(z_t^{(k_i)})\|_1 - \frac{2\alpha+2\eta}{1+\alpha+2\eta}\|\nabla h_t^{1/2}(z_t^{(k_i)}) \circ \mathbf{1}_{\{u_i\}}\|_1$$

$$= \left( 1 - \frac{2(\alpha+\eta)\gamma_t^{(k_i)}}{1+\alpha+2\eta} \right)\|\nabla h_t^{1/2}(z_t^{(k_i)})\|_1.$$

For any $k \ge 1$, the above per-iteration reduction gives us

$$\|\nabla h_t^{1/2}(z_t^{(K_t)})\|_1 \le \prod_{k=0}^{K_t} \prod_{i=1}^{|\mathcal{S}_t^{(k)}|} \left( 1 - \frac{2(\alpha+\eta)}{1+\alpha+2\eta}\gamma_t^{(k_i)} \right)\|\nabla h_t^{1/2}(z_t^{(0)})\|_1.$$

Denote that $\overline{\gamma}_t = \frac{1}{K_t}\sum_{k=0}^{K_t-1}\sum_{i=1}^{|\mathcal{S}_t^{(k)}|}\gamma_t^{(k_i)}$, we can obtain an upper bound of $K_t$ as the following

$$\log \frac{\|\nabla h_t^{1/2}(z_t^{(K_t)})\|_1}{\|\nabla h_t^{1/2}(z_t^{(0)})\|_1} \le \sum_{k=0}^{K_t-1}\sum_{i=1}^{|\mathcal{S}_t^{(k)}|} \log\left( 1 - \frac{2(\alpha+\eta)}{1+\alpha+2\eta}\gamma_t^{(k_i)} \right) \le -\sum_{k=0}^{K_t-1}\sum_{i=1}^{|\mathcal{S}_t^{(k)}|} \frac{2(\alpha+\eta)\gamma_t^{(k_i)}}{1+\alpha+2\eta}.$$

The above inequality implies $K_t \leq \frac{1+\alpha+2\eta}{2(\alpha+\eta)\overline{\gamma}_t} \log \frac{\|\nabla h_t^{1/2}(\mathbf{z}_t^{(0)})\|_1}{\|\nabla h_t^{1/2}(\mathbf{z}_t^{(K_t)})\|_1}$. Therefore, we have the time complexity as

$$\sum_{k=0}^{K_t-1} \mathrm{vol}(\mathcal{S}_t^{(k)}) = K_t \cdot \overline{\mathrm{vol}}(\mathcal{S}_t) \leq \frac{(1+\alpha+2\eta)\overline{\mathrm{vol}}(\mathcal{S}_t)}{2(\alpha+\eta)\overline{\gamma}_t} \log \frac{\|\nabla h_t^{1/2}(\mathbf{z}_t^{(0)})\|_1}{\|\nabla h_t^{1/2}(\mathbf{z}_t^{(K_t)})\|_1}.$$

To check the ratio is a upper bound of $\|\nabla h_t(\mathbf{z}_t^{(0)})\|_1/\epsilon_t$, note that $\nabla h_t(\mathbf{z}_t^{(k)})_{u_i} > \epsilon_t \sqrt{d_{u_i}}$ for $u_i \in \mathcal{S}_t^{(k)}$,

$$\epsilon_t \, \mathrm{vol}(\mathcal{S}_t^{(k)}) = \epsilon_t \sum_{i=1}^{|\mathcal{S}_t^{(k)}|} d_{u_i} < \sum_{i=1}^{|\mathcal{S}_t^{(k)}|} \left|\sqrt{d_{u_i}} \nabla_{u_i} h_t(\mathbf{z}_t^{(k)})\right| = \|\nabla h_t^{1/2}(\mathbf{z}_t^{(k)})\|_1.$$

Since $\|\nabla h_t^{1/2}(\mathbf{z}_t^{(0)})\|_1 \geq \|\nabla h_t^{1/2}(\mathbf{z}_t^{(1)})\|_1 \geq \cdots \geq \|\nabla h_t^{1/2}(\mathbf{z}_t^{(K_t)})\|_1$, this leads to

$$\epsilon_t \frac{1}{K_t} \sum_{k=0}^{K_t-1} \mathrm{vol}(\mathcal{S}_t^{(k)}) \leq \frac{1}{K_t} \sum_{k=0}^{K_t-1} \gamma_t^{(k_i)} \|\nabla h_t^{1/2}(\mathbf{z}_t^{(k)})\|_1 \leq \frac{1}{K_t} \sum_{k=0}^{K_t-1} \gamma_t^{(k_i)} \|\nabla h_t^{1/2}(\mathbf{z}_t^{(0)})\|_1,$$

where the above implies that $\frac{\overline{\mathrm{vol}}(\mathcal{S}_t)}{\overline{\gamma}_t} < \frac{|\nabla h_t^{1/2}(\mathbf{z}_t^{(0)})|_1}{\epsilon_t}$. The other upper bound follows a similar argument as in Theorem 3.3. Moreover, from the inequality in (17), we obtain that

$$\frac{2\alpha+2\eta}{1+\alpha+2\eta} \cdot \epsilon_t \cdot \mathrm{vol}(\mathcal{S}_t^{(k)}) \leq \frac{2\alpha+2\eta}{1+\alpha+2\eta} \sum_{i=1}^{|\mathcal{S}_t^{(k)}|} \|\nabla h_t^{1/2}(\mathbf{z}_t^{(k_i)}) \circ \mathbf{1}_{\{u_i\}}\|_1$$
$$\leq \|\nabla h_t^{1/2}(\mathbf{z}_t^{(k)})\|_1 - \|\nabla h_t^{1/2}(\mathbf{z}_t^{(k+1)})\|_1.$$

The total runtime can also be bounded as

$$\mathcal{T}_t^{\mathrm{LOCAPPR}} = \sum_{k=0}^{K_t-1} \mathrm{vol}(\mathcal{S}_t^{(k)}) \leq \frac{1+\alpha+2\eta}{2(\alpha+\eta)\epsilon_t} \sum_{k=0}^{K_t-1} \left(\|\nabla h_t^{1/2}(\mathbf{z}_t^{(k)})\|_1 - \|\nabla h_t^{1/2}(\mathbf{z}_t^{(k+1)})\|_1\right)$$
$$= \frac{1+\alpha+2\eta}{2(\alpha+\eta)\epsilon_t} \left(\|\nabla h_t^{1/2}(\mathbf{z}_t^{(0)})\|_1 - \|\nabla h_t^{1/2}(\mathbf{z}_t^{(K_t)})\|_1\right).$$

Combining the two bounds above, we prove the theorem. $\qquad\square$

### A.4 Proof of Lemma 3.4

Before we prove Lemma 3.4, we introduce the following two lemmas from Lin et al. [34] are presented for completeness. Lemma A.5 characterizes the error accumulation of $\{\varphi_t\}_{t\geq0}$ in Catalyst. For completeness, we include the full proof following the argument of Lin et al. [34].

**Lemma A.4** (Inequality of non-negative sequences). *Consider a increasing sequence $\{S_t\}_{t\geq0}$ and two non-negative sequences $\{a_t\}_{t\geq0}$ and $\{u_t\}_{t\geq0}$ such that for all $t$, $u_t^2 \leq S_t + \sum_{i=1}^t a_i u_i$. Then,*

$$S_t + \sum_{i=1}^t a_i u_i \leq \left(\sqrt{S_t} + \sum_{i=1}^t a_i\right)^2.$$

**Lemma A.5** (Convergence of Catalyst, Theorem 3 in Lin et al. [34]). *Consider the sequences $\{\mathbf{x}^{(t)}\}_{t\geq0}$ and $\{\mathbf{y}^{(t)}\}_{t\geq0}$ produced by Algorithm 1 for solving (P1), assuming that $\mathbf{x}^{(t)} \in \mathcal{H}(\varphi_t)$ defined in (C1) for all $t \geq 1$, Then,*

$$f(\mathbf{x}^{(t)}) - f^* \leq A_{t-1} \left(\sqrt{(1-\alpha_0)\left(f\left(\mathbf{x}^{(0)}\right) - f^*\right) + \frac{\gamma_0}{2}\left\|\mathbf{x}^* - \mathbf{x}^{(0)}\right\|^2} + 3\sum_{j=1}^t \sqrt{\frac{\varphi_j}{A_{j-1}}}\right)^2,$$
(20)

*where $\alpha_0 = \sqrt{q}$, $\gamma_0 = (\eta+\mu)\alpha_0(\alpha_0 - q)$ and $A_t = \prod_{j=1}^t(1-\alpha_j)$ with $A_0 = 1$ and $q = \mu/(\mu+\eta)$.*

*Proof.* Let us define the function $h_t(\boldsymbol{x}) = f(\boldsymbol{x}) + \frac{\eta}{2}\|\boldsymbol{x} - \boldsymbol{y}^{(t-1)}\|_2^2$. We show there exists an approximate sufficient descent condition for $h_t$. Since the solution of the proximal operator defined in (2) is $p(\boldsymbol{y}^{(t-1)})$, the unique minimizer of $h_t$, i.e., $p(\boldsymbol{y}^{(t-1)}) = \mathrm{prox}_{h_t/\eta}(\boldsymbol{y}^{(t-1)})$. The strong convexity of $h_t$ yields: for any $t \geq 1$, for all $\boldsymbol{x} \in \mathbb{R}^n$ and any $\theta_t > 0$,

$$h_t(\boldsymbol{x}) \overset{①}{\geq} h_t^* + \frac{\eta + \mu}{2}\|\boldsymbol{x} - p(\boldsymbol{y}^{(t-1)})\|_2^2$$

$$\overset{②}{\geq} h_t^* + \frac{\eta + \mu}{2}(1 - \theta_t)\|\boldsymbol{x} - \boldsymbol{x}^{(t)}\|_2^2 + \frac{\eta + \mu}{2}(1 - 1/\theta_t)\|\boldsymbol{x}^{(t)} - p(\boldsymbol{y}^{(t-1)})\|_2^2$$

$$\overset{③}{\geq} h_t(\boldsymbol{x}^{(t)}) - \varphi_t + \frac{\eta + \mu}{2}(1 - \theta_t)\|\boldsymbol{x} - \boldsymbol{x}^{(t)}\|_2^2 + \frac{\eta + \mu}{2}(1 - 1/\theta_t)\|\boldsymbol{x}^{(t)} - p(\boldsymbol{y}^{(t-1)})\|_2^2,$$

where ① uses the $(\mu + \eta)$-strong convexity of $h_t$. For ②, note for all $\boldsymbol{x}, \boldsymbol{y}, \boldsymbol{z} \in \mathbb{R}^n$ and $\theta > 0$,

$$\|\boldsymbol{x} - \boldsymbol{y}\|_2^2 \geq (1 - \theta)\|\boldsymbol{x} - \boldsymbol{z}\|_2^2 + (1 - 1/\theta)\|\boldsymbol{z} - \boldsymbol{y}\|_2^2$$

To see this, $\|\boldsymbol{x} - \boldsymbol{y}\|_2^2 = \|\boldsymbol{x} - \boldsymbol{z} + \boldsymbol{z} - \boldsymbol{y}\|_2^2 = \|\boldsymbol{x} - \boldsymbol{z}\|_2^2 + \|\boldsymbol{z} - \boldsymbol{y}\|_2^2 + 2\langle \boldsymbol{x} - \boldsymbol{z}, \boldsymbol{z} - \boldsymbol{y}\rangle$

$$2\langle \boldsymbol{x} - \boldsymbol{z}, \boldsymbol{z} - \boldsymbol{y}\rangle = \|\sqrt{\theta}(\boldsymbol{x} - \boldsymbol{z}) + (\boldsymbol{z} - \boldsymbol{y})/\sqrt{\theta}\|_2^2 - \theta\|\boldsymbol{x} - \boldsymbol{z}\|_2^2 - \|\boldsymbol{z} - \boldsymbol{y}\|_2^2/\theta$$

$$\geq -\theta\|\boldsymbol{x} - \boldsymbol{z}\|_2^2 - \|\boldsymbol{z} - \boldsymbol{y}\|_2^2/\theta.$$

The ③ uses $h_t(\boldsymbol{x}^{(t)}) - h_t^* \leq \varphi_t$. Moreover, when $\theta_t \geq 1$, the last term is positive and we have

$$h_t(\boldsymbol{x}) \geq h_t(\boldsymbol{x}^{(t)}) - \varphi_t + \frac{\eta + \mu}{2}(1 - \theta_t)\|\boldsymbol{x} - \boldsymbol{x}^{(t)}\|_2^2.$$

If instead $\theta_t \leq 1$, the coefficient $\frac{1}{\theta_t} - 1 \geq 0$ and we have

$$-\frac{\eta + \mu}{2}(1/\theta_t - 1)\|\boldsymbol{x}^{(t)} - p(\boldsymbol{y}^{(t-1)})\|_2^2 \geq -(1/\theta_t - 1)\left(h_t(\boldsymbol{x}^{(t)}) - h_t^*\right) \geq -(1/\theta_t - 1)\varphi_t$$

In this case, we have

$$h_t(\boldsymbol{x}) \geq h_t(\boldsymbol{x}^{(t)}) - \frac{\varphi_t}{\theta_t} + \frac{\eta + \mu}{2}(1 - \theta_t)\|\boldsymbol{x} - \boldsymbol{x}^{(t)}\|_2^2.$$

As a result, we have for all values of $\theta_t > 0$,

$$h_t(\boldsymbol{x}) \geq h_t(\boldsymbol{x}^{(t)}) + \frac{\eta + \mu}{2}(1 - \theta_t)\|\boldsymbol{x} - \boldsymbol{x}^{(t)}\|_2^2 - \frac{\varphi_t}{\min\{1, \theta_t\}}.$$

After expanding the expression of $h_t$, we then obtain the approximate descent condition.

$$f(\boldsymbol{x}^{(t)}) + \frac{\eta}{2}\|\boldsymbol{x}^{(t)} - \boldsymbol{y}^{(t-1)}\|_2^2 + \frac{\eta + \mu}{2}(1 - \theta_t)\|\boldsymbol{x} - \boldsymbol{x}^{(t)}\|_2^2 \leq f(\boldsymbol{x}) + \frac{\eta}{2}\|\boldsymbol{x} - \boldsymbol{y}^{(t-1)}\|_2^2 + \frac{\varphi_t}{\min\{1, \theta_t\}}. \quad (21)$$

Let us introduce a sequence $(S_t)_{t \geq 0}$ that will act as a Lyapunov function, with

$$S_t = (1 - \alpha_t)(f(\boldsymbol{x}^{(t)}) - f^*) + \alpha_t \frac{\eta \tau_t}{2}\|\boldsymbol{x}^* - \boldsymbol{v}^{(t)}\|_2^2,$$

where $\boldsymbol{x}^*$ is a minimizer of $f$, $\{\boldsymbol{v}^{(t)}\}_{t \geq 0}$ is a sequence defined by $\boldsymbol{v}^{(0)} = \boldsymbol{x}^{(0)}$ and

$$\boldsymbol{v}^{(t)} = \boldsymbol{x}^{(t)} + \frac{1 - \alpha_{t-1}}{\alpha_{t-1}}(\boldsymbol{x}^{(t)} - \boldsymbol{x}^{(t-1)}) \quad \text{for } t \geq 1$$

and $\{\tau_t\}_{t \geq 0}$ is an auxiliary quantity defined by $\tau_t = \frac{\alpha_t - q}{1 - q}$. The way we introduce these variables allows us to write the following relationship,

$$\boldsymbol{y}^{(t)} = \tau_t \boldsymbol{v}^{(t)} + (1 - \tau_t)\boldsymbol{x}^{(t)}, \quad \text{for all } t \geq 0,$$

which follows from a simple calculation. Then by setting $\boldsymbol{z}^{(t)} = \alpha_{t-1}\boldsymbol{x}^* + (1 - \alpha_{t-1})\boldsymbol{x}^{(t-1)}$, the following relations hold for all $t \geq 1$ by $\mu$-strongly convexity of $f$.

$$f(\boldsymbol{z}^{(t)}) \leq \alpha_{t-1}f^* + (1 - \alpha_{t-1})f(\boldsymbol{x}^{(t-1)}) - \frac{\mu \alpha_{t-1}(1 - \alpha_{t-1})}{2}\|\boldsymbol{x}^* - \boldsymbol{x}^{(t-1)}\|_2^2$$

$$\boldsymbol{z}^{(t)} - \boldsymbol{x}^{(t)} = \alpha_{t-1}(\boldsymbol{x}^* - \boldsymbol{v}^{(t)})$$

and also the following one (by expanding out $z^{(t)} = \alpha_{t-1}x^* + (1 - \alpha_{t-1})x^{(t-1)}$).

$$\|z^{(t)} - y^{(t-1)}\|_2^2 = \|(\alpha_{t-1} - \tau_{t-1})(x^* - x^{(t-1)}) + \tau_{t-1}(x^* - v^{(t-1)})\|_2^2$$

$$= \alpha_{t-1}^2 \|(1 - \tau_{t-1}/\alpha_{t-1})(x^* - x^{(t-1)}) + \frac{\tau_{t-1}}{\alpha_{t-1}}(x^* - v^{(t-1)})\|_2^2$$

$$\leq \alpha_{t-1}^2(1 - \tau_{t-1}/\alpha_{t-1})\|x^* - x^{(t-1)}\|_2^2 + \alpha_{t-1}^2 \frac{\tau_{t-1}}{\alpha_{t-1}}\|x^* - v^{(t-1)}\|_2^2$$

$$= \alpha_{t-1}(\alpha_{t-1} - \tau_{t-1})\|x^* - x^{(t-1)}\|_2^2 + \alpha_{t-1}\tau_{t-1}\|x^* - v^{(t-1)}\|_2^2,$$

where we used the convexity of the norm and the fact that $\tau_t \leq \alpha_t$. Using the previous relations in Eq. (21) with $x = z^{(t)} = \alpha_{t-1}x^* + (1 - \alpha_{t-1})x^{(t-1)}$, gives for all $t \geq 1$,

$$f(x^{(t)}) + \frac{\eta}{2}\|x^{(t)} - y^{(t-1)}\|_2^2 + \frac{\eta + \mu}{2}(1 - \theta_t)\alpha_{t-1}^2\|x^* - v^{(t)}\|_2^2$$

$$\leq \alpha_{t-1}f^* + (1 - \alpha_{t-1})f(x^{(t-1)}) - \frac{\mu}{2}\alpha_{t-1}(1 - \alpha_{t-1})\|x^* - x^{(t-1)}\|_2^2$$

$$+ \frac{\eta\alpha_{t-1}(\alpha_{t-1} - \tau_{t-1})}{2}\|x^* - x^{(t-1)}\|_2^2 + \frac{\eta\alpha_{t-1}\tau_{t-1}}{2}\|x^* - v^{(t-1)}\|_2^2 + \frac{\varphi_t}{\min\{1, \theta_t\}}$$

Remark that for all $t \geq 1$, $\alpha_{t-1} - \tau_{t-1} = \alpha_{t-1} - \frac{\alpha_{t-1} - q}{1-q} = \frac{q(1 - \alpha_{t-1})}{1-q} = \frac{\mu}{\eta}(1 - \alpha_{t-1})$, and the quadratic terms $x^* - x^{(t-1)}$ cancel each other. Then, after noticing that for all $t \geq 1$,

$$\tau_t\alpha_t = \frac{\alpha_t^2 - q\alpha_t}{1 - q} = \frac{(\eta + \mu)(1 - \alpha_t)\alpha_{t-1}^2}{\eta} \quad (\text{ by the updates of } \alpha_t),$$

which gives $f(x^{(t)}) - f^* + \frac{\eta + \mu}{2}\alpha_{t-1}^2\|x^* - v^{(t)}\|_2^2 = \frac{S_t}{1 - \alpha_t}$. We are left, for all $t \geq 1$, with

$$\frac{1}{1 - \alpha_t}S_t \leq S_{t-1} + \frac{\varphi_t}{\min\{1, \theta_t\}} - \frac{\eta}{2}\|x^{(t)} - y^{(t-1)}\|_2^2 + \frac{(\eta + \mu)\alpha_{t-1}^2\theta_t}{2}\|x^* - v^{(t)}\|_2^2 \quad (22)$$

Using the fact that $\frac{1}{\min\{1, \theta_t\}} \leq 1 + \frac{1}{\theta_t}$, we immediately derive from equation (22) that

$$\frac{S_t}{1 - \alpha_t} \leq S_{t-1} + \varphi_t + \frac{\varphi_t}{\theta_t} - \frac{\eta}{2}\|x^{(t)} - y^{(t-1)}\|_2^2 + \frac{(\eta + \mu)\alpha_{t-1}^2\theta_t}{2}\|x^* - v^{(t)}\|_2^2.$$

We obtain the following by minimizing the right-hand side of the above w.r.t $\theta_t$.

$$\frac{S_t}{1 - \alpha_t} \leq S_{t-1} + \varphi_t + \sqrt{2\varphi_t(\mu + \eta)}\alpha_{t-1}\|x^* - v^{(t)}\|,$$

and after unrolling the recursion,

$$\frac{S_t}{A_t} \leq S_0 + \sum_{j=1}^{t}\frac{\varphi_j}{A_{j-1}} + \sum_{j=1}^{t}\frac{\sqrt{2\varphi_j(\mu + \eta)}\alpha_{j-1}\|x^* - v^{(j)}\|}{A_{j-1}}$$

We may define $u_j = \sqrt{(\mu + \eta)}\alpha_{j-1}\|x^* - v^{(j)}\|/\sqrt{2A_{j-1}}$ and $a_j = 2\sqrt{\varphi_j}/\sqrt{A_{j-1}}$. Note $S_t/A_t = S_t/(A_{t-1}(1 - \alpha_t))$ where $S_t/(1 - \alpha_t) = f(x^{(t)}) - f^* + \frac{\eta + \mu}{2}\alpha_{t-1}^2\|x^* - v^{(t)}\|_2^2$. Therefore, we have $u_t^2 \leq \frac{S_t}{A_t}$.

$$u_t^2 \leq S_0 + \sum_{j=1}^{t}\frac{\varphi_j}{A_{j-1}} + \sum_{j=1}^{t}a_ju_j \quad \text{for all} \quad t \geq 1.$$

This allows us to apply Lemma A.4 (Let $S_t' = S_0 + \sum_{j=1}^{t}\frac{\varphi_j}{A_{j-1}}$, then we have $u_t^2 \leq S_t' + \sum_{j=1}^{t}a_ju_j \leq (\sqrt{S_t'} + \sum_{j=1}^{t}a_j)^2$ meanwhile $S_t' + \sum_{j=1}^{t}a_ju_j \geq S_t/A_t$ ), which yields

$$\frac{S_t}{A_t} \leq \left(\sqrt{S_0 + \sum_{j=1}^{t}\frac{\varphi_j}{A_{j-1}}} + 2\sum_{j=1}^{t}\sqrt{\frac{\varphi_j}{A_{j-1}}}\right)^2 \leq \left(\sqrt{S_0} + 3\sum_{j=1}^{t}\sqrt{\frac{\varphi_j}{A_{j-1}}}\right)^2,$$

which provides us the desired result given that $f(x^{(t)}) - f^* \leq \frac{S_t}{1 - \alpha_t}$ and that $v^{(0)} = x^{(0)}$. $\quad\square$

We now simplify the above theorem for our quadratic problem (P1). In particular, we prove a variant of Proposition 5 from Lin et al. [34], replacing $f(\boldsymbol{x}^{(0)}) - f(\boldsymbol{x}^*)$ with its upper bound $L\|\boldsymbol{b}\|_1^2/2$.

**Corollary A.6.** *Let $\{\boldsymbol{x}^{(t)}\}_{t\geq 0}$ and $\{\boldsymbol{y}^{(t)}\}_{t\geq 0}$ be generated by Algorithm 1, assuming that $\boldsymbol{x}^{(t)} \in \mathcal{H}(\varphi_t)$ for all $t \geq 1$ where local methods find $\boldsymbol{z}_t^{(K_t)}$ such that*

$$h_t(\boldsymbol{z}_t^{(K_t)}) - h_t^* \leq \varphi_t := \frac{(L+\mu)\|\boldsymbol{b}\|_1^2(1-\rho)^t}{18}.$$

*Let the objective $f$ be defined in (P1) and assume $\boldsymbol{x}^{(0)} = \boldsymbol{0}$. where $\gamma_0 = \mu(1-\sqrt{q})$ and $A_t = (1-\sqrt{q})^t$ with $A_0 = 1$ and $q = \mu/(\mu+\eta)$ and $\alpha_0 = \sqrt{q}$. , then the final solution $\boldsymbol{x}^{(t)}$ is guaranteed*

$$f(\boldsymbol{x}^{(t)}) - f^* \leq \frac{2(L+\mu)\|\boldsymbol{b}\|_1^2}{(\sqrt{q}-\rho)^2}(1-\rho)^{t+1}. \tag{23}$$

*Proof.* We first show the following inequality

$$\|\boldsymbol{D}^{-1/2}(\boldsymbol{x}^{(t)} - \boldsymbol{x}_f^*)\|_\infty \leq \sqrt{\frac{2(1-\sqrt{q})^{t-1}}{\mu}\left(\sqrt{(1-\sqrt{q})\left(\frac{L+\mu}{2}\right)\|\boldsymbol{b}\|_1^2} + 3\sum_{j=1}^t \sqrt{\frac{\varphi_j}{A_{j-1}}}\right)}, \tag{24}$$

By the definition of $f$ in (P1), we know $\boldsymbol{x}_f^* = \alpha\boldsymbol{Q}^{-1}\boldsymbol{D}^{-1/2}\boldsymbol{b}$. Since $\boldsymbol{x}^{(0)} = \boldsymbol{0}$ and $f$ is $L$-smooth, it implies

$$f(\boldsymbol{x}^{(0)}) - f(\boldsymbol{x}_f^*) \leq \frac{L}{2}\|\boldsymbol{x}^{(0)} - \boldsymbol{x}_f^*\|_2^2 = \frac{L}{2}\|\alpha\boldsymbol{Q}^{-1}\boldsymbol{D}^{-1/2}\boldsymbol{b}\|_2^2 = \frac{L}{2}\|\boldsymbol{D}^{-1/2}\boldsymbol{\Pi}_\alpha\boldsymbol{b}\|_2^2 \leq \frac{L}{2}\|\boldsymbol{b}\|_1^2,$$

where the second equality due to the identity $\boldsymbol{\Pi}_\alpha = \alpha\boldsymbol{D}^{1/2}\boldsymbol{Q}^{-1}\boldsymbol{D}^{-1/2}$ and the last inequality is from the fact that $\|\boldsymbol{\Pi}_\alpha\|_1 = 1$ and $\|\boldsymbol{x}\|_2 \leq \|\boldsymbol{x}\|_1$. When $\alpha_0 = \sqrt{q}, q = \frac{\mu}{\mu+\eta}$, then $\gamma_0 = (\eta+\mu)\alpha_0(\alpha_0 - q) = \mu(1-\sqrt{q})$, then it indicates

$$\sqrt{(1-\alpha_0)\left(f\left(\boldsymbol{x}^{(0)}\right) - f^*\right) + \frac{\gamma_0}{2}\|\boldsymbol{x}_f^* - \boldsymbol{x}^{(0)}\|_2^2} \leq \sqrt{(1-\sqrt{q})\left(\frac{L+\mu}{2}\right)\|\boldsymbol{x}^{(0)} - \boldsymbol{x}_f^*\|_2^2}$$

$$\leq \sqrt{(1-\sqrt{q})\left(\frac{L+\mu}{2}\right)\|\boldsymbol{b}\|_1^2}.$$

Since $A_{t-1} = (1-\sqrt{q})^{t-1}$, one can simplify Eq. (20) of Lemma A.5 as the following

$$f(\boldsymbol{x}^{(t)}) - f(\boldsymbol{x}_f^*) \leq (1-\sqrt{q})^{t-1}\left(\sqrt{(1-\sqrt{q})\left(\frac{L+\mu}{2}\right)\|\boldsymbol{b}\|_1^2} + 3\sum_{j=1}^t \sqrt{\frac{\varphi_j}{A_{j-1}}}\right)^2.$$

Let $\varphi_t = (L+\mu)\|\boldsymbol{b}\|_1^2(1-\rho)^t/18$, then

$$3\sqrt{\frac{\varphi_j}{A_{j-1}}} = \sqrt{\frac{9\varphi_j}{(1-\sqrt{q})^{j-1}}} = \sqrt{\frac{\frac{L+\mu}{2}\|\boldsymbol{b}\|_1^2(1-\rho)^j}{(1-\sqrt{q})^{j-1}}} = \sqrt{\frac{(1-\sqrt{q})\frac{L+\mu}{2}\|\boldsymbol{b}\|_1^2(1-\rho)^j}{(1-\sqrt{q})^j}}.$$

Follow the same steps as shown in Proposition 5 of Lin et al. [34], the right-hand side of Eq. (24) is

$$\sqrt{(1-\sqrt{q})\left(\frac{L+\mu}{2}\right)\|\boldsymbol{b}\|_1^2} + 3\sum_{j=1}^t \sqrt{\frac{\varphi_j}{A_{j-1}}} = \sqrt{(1-\sqrt{q})\left(\frac{L+\mu}{2}\right)\|\boldsymbol{b}\|_1^2}\left(1 + \sum_{j=1}^t \left(\sqrt{\frac{1-\rho}{1-\sqrt{q}}}\right)^j\right)$$

$$\leq \sqrt{(1-\sqrt{q})\left(\frac{L+\mu}{2}\right)\|\boldsymbol{b}\|_1^2}\frac{\zeta^{t+1}}{\zeta - 1}, \text{ where } \zeta = \sqrt{\frac{1-\rho}{1-\sqrt{q}}}.$$

This leads to

$$f(\boldsymbol{x}^{(t)}) - f^* \leq (1-\sqrt{q})^{t-1}\left(\sqrt{(1-\sqrt{q})\left(\frac{L+\mu}{2}\right)\|\boldsymbol{b}\|_1^2}\frac{\zeta^{t+1}}{\zeta - 1}\right)^2$$

$$\leq (1 - \sqrt{q})^t \left( \frac{L + \mu}{2} \right) \|\boldsymbol{b}\|_1^2 \left( \frac{\zeta^{t+1}}{\zeta - 1} \right)^2$$

$$= \frac{L + \mu}{2} \|\boldsymbol{b}\|_1^2 \left( \frac{\zeta}{\zeta - 1} \right)^2 ((1 - \sqrt{q})\zeta^2)^t$$

$$= \frac{L + \mu}{2} \|\boldsymbol{b}\|_1^2 \left( \frac{\sqrt{1 - \rho}}{\sqrt{1 - \rho} - \sqrt{1 - \sqrt{q}}} \right)^2 (1 - \rho)^t$$

$$= \frac{L + \mu}{2} \|\boldsymbol{b}\|_1^2 \left( \frac{1}{\sqrt{1 - \rho} - \sqrt{1 - \sqrt{q}}} \right)^2 (1 - \rho)^{t+1}$$

$$\leq \frac{L + \mu}{2} \|\boldsymbol{b}\|_1^2 \frac{4}{(\sqrt{q} - \rho)^2} (1 - \rho)^{t+1} = \frac{2(L + \mu)\|\boldsymbol{b}\|_1^2}{(\sqrt{q} - \rho)^2} (1 - \rho)^{t+1},$$

where the last inequality uses the fact that $\sqrt{1 - \rho} - \sqrt{1 - \sqrt{q}} \geq \frac{\sqrt{q} - \rho}{2}$. Since $f$ is $\mu$-strongly convex, then

$$\|\boldsymbol{D}^{-1/2}(\boldsymbol{x}^{(t)} - \boldsymbol{x}_f^*)\|_\infty \leq \|\boldsymbol{x}^{(t)} - \boldsymbol{x}_f^*\|_2 \leq \sqrt{\frac{2}{\mu}(f(\boldsymbol{x}^{(t)}) - f(\boldsymbol{x}_f^*))},$$

which leads us to have the upper bound in Eq. (24). Then, we have the following inequality

$$\|\boldsymbol{D}^{-1/2}(\boldsymbol{x}^{(t)} - \boldsymbol{x}_f^*)\|_\infty \leq \frac{2\sqrt{(L + \mu)}\|\boldsymbol{b}\|_1}{\sqrt{\mu}(\sqrt{q} - \rho)} (1 - \rho)^{\frac{t+1}{2}}$$

$\square$

The above theorem implies that if $h_t(\boldsymbol{x}^{(t)}) - h_t^* \leq \varphi_t := \frac{(L+\mu)\|\boldsymbol{b}\|_1^2(1-\rho)^t}{18}$, then the function value error satisfies $f(\boldsymbol{x}^{(t)}) - f^* := \frac{2(L+\mu)\|\boldsymbol{b}\|_1^2}{(\sqrt{q}-\rho)^2}(1 - \rho)^{t+1} = \frac{36\varphi_t(1-\rho)}{(\sqrt{q}-\rho)^2}$. Based on Corollary A.6, we establish the total iteration complexity for AESP as presented in the following lemma.

**Lemma 3.4** (Outer-loop iteration complexity of AESP). *If each iteration of AESP, presented in Algorithm 1, finds $\boldsymbol{x}^{(t)} := \boldsymbol{z}_t^{(K_t)}$ using $\mathcal{M}$, satisfying $h_t(\boldsymbol{z}_t^{(K_t)}) - h_t^* \leq \varphi_t := (L + \mu)\|\boldsymbol{b}\|_1^2(1 - \rho)^t/18$, then the total number of iterations $T$ required to ensure $\hat{\boldsymbol{x}} = \text{AESP}(\epsilon, \alpha, \boldsymbol{b}, \eta, \mathcal{G}, \mathcal{M}) \in \mathcal{P}(\epsilon, \alpha, \boldsymbol{b}, \mathcal{G})$ as defined in Eq. (3), for solving (P1), satisfies the bound*

$$T \leq \frac{1}{\rho} \log \left( \frac{4(L + \mu)\|\boldsymbol{b}\|_1^2}{\mu \epsilon^2 (\sqrt{q} - \rho)^2} \right), \text{ where } \rho = 0.9\sqrt{q} \text{ and } q = \frac{\mu}{\mu + \eta}.$$

*Furthermore, $\varphi_t$ has a lower bound $\varphi_t \geq \mu \epsilon^2 (\sqrt{q} - \rho)^2/72$ for all $t \in [T]$.*

*Proof.* As $f$ is $\mu$-strongly convex, then $\frac{\mu}{2}\|\boldsymbol{D}^{-1/2}(\boldsymbol{x}^{(t)} - \boldsymbol{x}^*)\|_\infty^2 \leq \frac{\mu}{2}\|\boldsymbol{x}^{(t)} - \boldsymbol{x}^*\|_2^2 \leq f(\boldsymbol{x}^{(t)}) - f^*$. It is enough to find a minimal integer $T$ such that $f(\boldsymbol{x}^{(T)}) - f^* \leq \frac{\mu \epsilon^2}{2}$. That is, $f(\boldsymbol{x}^{(T)}) - f^* \leq \frac{2(L+\mu)\|\boldsymbol{b}\|_1^2}{(\sqrt{q}-\rho)^2}(1 - \rho)^{T+1} \leq \mu \epsilon^2/2$. We have

$$\frac{2(L + \mu)(1 - \rho)^{T+1}\|\boldsymbol{b}\|_1^2}{(\sqrt{q} - \rho)^2} \leq \frac{\mu \epsilon^2}{2}$$

$$\Rightarrow \quad (1 - \rho)^{T+1} \leq \frac{\mu \epsilon^2 (\sqrt{q} - \rho)^2}{4(L + \mu)\|\boldsymbol{b}\|_1^2} \quad \Rightarrow \quad T \leq \frac{1}{\rho} \log \left( \frac{4(L + \mu)\|\boldsymbol{b}\|_1^2}{\mu \epsilon^2 (\sqrt{q} - \rho)^2} \right).$$

The minimal $T$ satisfies the above inequality means $(1 - \rho)^T$ has the following lower bound

$$(1 - \rho)^T \geq \frac{\mu \epsilon^2 (\sqrt{q} - \rho)^2}{4(L + \mu)\|\boldsymbol{b}\|_1^2}.$$

Applying Corollary A.5, we know that $h_t(\boldsymbol{z}_t^{(K_t)}) - h_t^* \leq \varphi_t := \frac{1}{18}(L + \mu)\|\boldsymbol{b}\|_1^2(1 - \rho)^t$, then $\varphi_T$ is guaranteed lower bounded as

$$72\varphi_T := 4(L + \mu)\|\boldsymbol{b}\|_1^2(1 - \rho)^T \geq \mu \epsilon^2 (\sqrt{q} - \rho)^2 \quad \Rightarrow \quad \varphi_T \geq \frac{\mu \epsilon^2 (\sqrt{q} - \rho)^2}{72}.$$

$\square$

## A.5  Proof of Theorem 3.5

**Theorem 3.5** (Time complexity of AESP). *Let the simple graph $\mathcal{G}(\mathcal{V}, \mathcal{E})$ be connected and undirected, and let $f(\boldsymbol{x})$ be defined in (P1). Assume the precision $\epsilon > 0$ satisfies $\{i : |b_i| \geq \epsilon d_i\} \neq \emptyset$ and damping factor $\alpha < 1/2$. Applying $\hat{\boldsymbol{x}} = \mathrm{AESP}(\epsilon, \alpha, \boldsymbol{b}, \eta, \mathcal{G}, \mathcal{M})$ with $\eta = L - 2\mu$ and $\mathcal{M}$ be either LocGD or LocAPPR, then AESP presented in Algorithm 1, finds a solution $\hat{\boldsymbol{x}}$ such that $\|\boldsymbol{D}^{-1/2}(\hat{\boldsymbol{x}} - \boldsymbol{x}_f^*)\|_\infty \leq \epsilon$ with the dominated time complexity $\mathcal{T}$ bounded by*

$$\mathcal{T} \leq \sum_{t=1}^{T} \min \left\{ \frac{\overline{\mathrm{vol}}(\mathcal{S}_t)}{\tau \overline{\gamma}_t} \log \frac{C_{h_t}^0}{C_{h_t}^{K_t}}, \frac{C_{h_t}^0 - C_{h_t}^{K_t}}{\tau \epsilon_t} \right\}, \ with \ \frac{\overline{\mathrm{vol}}(\mathcal{S}_t)}{\overline{\gamma}_t} \leq \min \left\{ \frac{C_{h_t}^0}{\epsilon_t}, 2m \right\},$$

*where $\tau$, $\epsilon_t$, $C_{h_t}^0$ and $C_{h_t}^{K_t}$ are defined in Theorem 3.3. Furthermore, $q = \mu/(L - \mu)$ and the number of outer iterations satisfies*

$$T \leq \frac{10}{9\sqrt{q}} \log \left( \frac{400(L + \mu)\|\boldsymbol{b}\|_1^2}{\mu \epsilon^2 q} \right) = \tilde{\mathcal{O}} \left( \frac{1}{\sqrt{\alpha}} \right).$$

*Proof.* By the definition of total time complexity $\mathcal{T}$ in Eq. (4), we have $\mathcal{T} = \sum_{t=1}^{T} \mathcal{T}_t^{\mathrm{LocGD}}$ or $\mathcal{T} = \sum_{t=1}^{T} \mathcal{T}_t^{\mathrm{LocAPPR}}$. Hence, the overall time complexity directly follows from Theorem 3.3 and Theorem A.3. The upper bound on the total iteration complexity $T$ is from Lemma 3.4. When $\alpha = \mu < 0.5$, we have $q = \sqrt{\alpha/(1-\alpha)}$, leading to an iteration complexity of $T = \tilde{\mathcal{O}}(1/\sqrt{\alpha})$. $\square$

## A.6  Proof of Theorem 3.6

The following theorem establishes the time complexity of AESP-PPR (Algorithm 2).

**Theorem 3.6** (Time complexity of AESP-PPR). *Let the simple graph $\mathcal{G}(\mathcal{V}, \mathcal{E})$ be connected and undirected, assuming $\alpha < 1/2$. The PPR vector of $s \in \mathcal{V}$ is defined in Eq. (1), and the precision $\epsilon \in (0, 1/d_s)$. Suppose $\hat{\boldsymbol{\pi}} = \mathrm{AESP\text{-}PPR}(\epsilon, \alpha, s, \mathcal{G}, \mathcal{M})$ be returned by Algorithm 2. When $\mathcal{M}$ is either LocGD (Algorithm 3) or LocAPPR (Algorithm 4), then $\hat{\boldsymbol{\pi}}$ satisfies $\|\boldsymbol{D}^{-1}(\hat{\boldsymbol{\pi}} - \boldsymbol{\pi})\|_\infty \leq \epsilon$ and AESP-PPR has a dominated time complexity bounded by*

$$\mathcal{T} \leq \min \left\{ \tilde{\mathcal{O}} \left( \frac{\overline{\mathrm{vol}}(\mathcal{S}_{T_{\max}})}{\sqrt{\alpha} \overline{\gamma}_{T_{\max}}} \right), \tilde{\mathcal{O}} \left( \frac{\max_t C_{h_t}^0}{\sqrt{\alpha} \epsilon_T} \right) \right\} = \min \left\{ \tilde{\mathcal{O}} \left( \frac{m}{\sqrt{\alpha}} \right), \tilde{\mathcal{O}} \left( \frac{R^2/\epsilon^2}{\sqrt{\alpha}} \right) \right\}, \quad (25)$$

*where $T_{\max} := \arg \max_{t \in [T]} \overline{\mathrm{vol}}(\mathcal{S}_t)/\overline{\gamma}_t$ and $R$ is defined in Eq. (7).*

*Proof.* We first analyze the total iteration complexity $T$ required in Algorithm 2. For the PPR problem defined in (1), when $\alpha < 0.5$, $\boldsymbol{b} = \boldsymbol{e}_s$, $\mu = \alpha$, $L = 1$, $\eta = 1 - 2\alpha$, $q = \sqrt{\alpha/(1-\alpha)}$, and $\rho = 0.9\sqrt{q}$, we seek to determine $t$ such that $(1 - \rho)^{t+1} \leq \alpha^2 \epsilon^2/(400(1 - \alpha^2))$. Since $\rho = 0.9\sqrt{q}$, the total iteration complexity simplifies to

$$T \leq \frac{1}{\rho} \log \left( \frac{4(L + \mu)\|\boldsymbol{b}\|_1^2}{\mu \epsilon^2 (\sqrt{q} - \rho)^2} \right) \leq \left\lceil \frac{1}{\rho} \log \left( \frac{400(1 - \alpha^2)}{\alpha^2 \epsilon^2} \right) \right\rceil = \left\lceil \frac{10}{9} \sqrt{\frac{1 - \alpha}{\alpha}} \log \left( \frac{400(1 - \alpha^2)}{\alpha^2 \epsilon^2} \right) \right\rceil.$$

For $t \in [T]$, $\varphi_t := \frac{L+\mu}{18} \|\boldsymbol{b}\|_1^2 (1 - \rho)^t$ and we know that $\frac{2(L+\mu)\|\boldsymbol{b}\|_1^2}{(\sqrt{q}-\rho)^2}(1 - \rho)^{t+1} \leq \frac{\mu \epsilon^2}{2}$, then $\varphi_t = \frac{L+\mu}{18} \|\boldsymbol{b}\|_1^2 (1 - \rho)^t \leq (\sqrt{q} - \rho)^2 \mu \epsilon^2/(72(1 - \rho))$. As $\epsilon_t = \max\{\sqrt{\frac{(\eta+\alpha)\varphi_t}{m}}, \frac{2(\eta+\alpha)\varphi_t}{\|\nabla h_t^{1/2}(\boldsymbol{z}_t^{(0)})\|_1}\}$ from Lemma 3.2, then for $t \in [T]$, we have

$$\begin{aligned} \epsilon_t &\geq \frac{2(\eta + \alpha)\varphi_t}{\|\nabla h_t^{1/2}(\boldsymbol{z}_t^{(0)})\|_1} \geq \frac{2(\eta + \alpha)\varphi_T}{\|\nabla h_t^{1/2}(\boldsymbol{z}_t^{(0)})\|_1} \\ &= \frac{(\eta + \alpha)(\sqrt{q} - \rho)^2 \mu \epsilon^2}{36(1 - \rho)\|\nabla h_t^{1/2}(\boldsymbol{z}_t^{(0)})\|_1} = \frac{\alpha^2 \epsilon^2}{3600(1 - 0.9\sqrt{q})\|\nabla h_t^{1/2}(\boldsymbol{z}_t^{(0)})\|_1}, \end{aligned}$$

where the first equality follows from Lemma 3.4, which states that $\varphi_T \geq \frac{\mu \epsilon^2 (\sqrt{q} - \rho)^2}{72}$. By the bounded level set assumption for the scaled gradient, we have $\|\nabla h_t^{1/2}(\boldsymbol{z}_t^{(0)})\|_1 \leq$

$R\|\nabla h_1^{1/2}(\boldsymbol{z}_1^{(0)})\|_1 = R\|\nabla h_1^{1/2}(\boldsymbol{0})\|_1 = R\|\alpha \boldsymbol{e}_s\|_1 = \alpha R$ where we assume that $\boldsymbol{z}_1^{(0)} = \boldsymbol{0}$. This leads to

$$\frac{\max_t C_{h_t}^0}{\epsilon_T} \leq \frac{3600(1 - 0.9\sqrt{q})\|\nabla h_t^{1/2}(\boldsymbol{z}_t^{(0)})\|_1 \cdot \max_t C_{h_t}^0}{\alpha^2 \epsilon^2}$$

$$\leq \frac{3600(1 - 0.9\sqrt{q})R^2\alpha^2}{\alpha^2 \epsilon^2} = \mathcal{O}\left(\frac{R^2}{\epsilon^2}\right)$$

Note that $T_{\max} := \arg\max_{t \in [T]} \overline{\text{vol}}(\mathcal{S}_t)/\overline{\gamma}_t$ and that $\frac{\overline{\text{vol}}(\mathcal{S}_t)}{\overline{\gamma}_t} \leq \min\left\{\frac{C_{T_{\max}}}{\epsilon_{T_{\max}}}, 2m\right\}$. By combining the two bounds above, we complete the proof of the theorem. $\qquad\square$

## A.7 Proof of Corollary A.7

**Corollary A.7.** *Let $\boldsymbol{x}_t^{(K_t)}$ be the output of either* LOCGD *or* LOCAPPR, *as defined in Algorithm 3 and Algorithm 4, respectively. Define the objective error as $e_t(\boldsymbol{z}_t^{(K_t)}) := h_t(\boldsymbol{z}_t^{(K_t)}) - h_t(\boldsymbol{x}_t^*)$. Then, the following bound holds*

$$e_t(\boldsymbol{z}_t^{(K_t)}) \leq \frac{1}{(1-\alpha)} \cdot \prod_{k=0}^{K_t-1}\left(1 - \frac{2\gamma_t^{(k)}}{3}\right)^2 \|\nabla h_t^{1/2}(\boldsymbol{z}_t^{(0)})\|_1^2.$$

*Proof.* Recall $\tilde{\boldsymbol{Q}} = \frac{1+\alpha+2\eta}{2}\boldsymbol{I} - \frac{1-\alpha}{2}\boldsymbol{D}^{-1/2}\boldsymbol{A}\boldsymbol{D}^{-1/2}$, and the target linear system to solve is $\tilde{\boldsymbol{Q}}\boldsymbol{z} = \boldsymbol{b}^{(t-1)}$. Note that

$$\boldsymbol{\Pi}_{\frac{\eta+\alpha}{1+\eta}} := \frac{\eta+\alpha}{1+\eta}\left(\frac{1 + \frac{\eta+\alpha}{1+\eta}}{2}\boldsymbol{I} - \frac{1 - \frac{\eta+\alpha}{1+\eta}}{2}\boldsymbol{A}\boldsymbol{D}^{-1}\right)^{-1}$$

$$= \frac{\eta+\alpha}{1+\eta}\left(\frac{1+\alpha+2\eta}{2(1+\eta)}\boldsymbol{I} - \frac{1-\alpha}{2(1+\eta)}\boldsymbol{A}\boldsymbol{D}^{-1}\right)^{-1}$$

$$= (\eta+\alpha)\left(\frac{1+\alpha+2\eta}{2}\boldsymbol{I} - \frac{1-\alpha}{2}\boldsymbol{A}\boldsymbol{D}^{-1}\right)^{-1}.$$

Hence, $\boldsymbol{D}^{-1/2}\boldsymbol{\Pi}_{\frac{\eta+\alpha}{1+\eta}}\boldsymbol{D}^{1/2} = (\eta+\alpha)\tilde{\boldsymbol{Q}}^{-1}$. Recall $\boldsymbol{x}_t^* = \tilde{\boldsymbol{Q}}^{-1}\boldsymbol{b}^{(t-1)}$. Let $(\hat{\boldsymbol{z}}, \hat{\boldsymbol{r}})$ be estimate and residual pair, then $\hat{\boldsymbol{z}} - \boldsymbol{x}_t^* = -\tilde{\boldsymbol{Q}}^{-1}\hat{\boldsymbol{r}} = \tilde{\boldsymbol{Q}}^{-1}\nabla h_t(\hat{\boldsymbol{z}})$. Since $h_t$ is $L+\eta$-strongly smooth, then for any $\boldsymbol{z} \in \mathbb{R}^n$, it implies $h_t(\boldsymbol{z}) - h_t(\boldsymbol{x}_t^*) \leq \frac{\eta+L}{2}\|\boldsymbol{z} - \boldsymbol{x}_t^*\|_2^2$. Let $\boldsymbol{z}_t^{(K_t)}$ be the estimate returned by APPR or LOCGD, then we have

$$h_t(\boldsymbol{z}_t^{(K_t)}) - h_t(\boldsymbol{x}_t^*) \leq \frac{\eta+L}{2}\|\boldsymbol{z}_t^{(K_t)} - \boldsymbol{x}_t^*\|_2^2$$

$$= \frac{\eta+L}{2}\|\tilde{\boldsymbol{Q}}^{-1}\nabla h_t(\boldsymbol{z}_t^{(K_t)})\|_1^2$$

$$= \frac{(\eta+L)}{2(\eta+\alpha)^2}\|\boldsymbol{D}^{-1/2}\boldsymbol{\Pi}_{\frac{\eta+\alpha}{1+\eta}}\boldsymbol{D}^{1/2}\nabla h_t(\boldsymbol{z}_t^{(K_t)})\|_1^2$$

$$\leq \frac{(\eta+L)}{2(\eta+\alpha)^2}\|\boldsymbol{D}^{1/2}\nabla h_t(\boldsymbol{z}_t^{(K_t)})\|_1^2,$$

where the last inequality is from the fact that for any $\nu > 0$, $\|\boldsymbol{D}^{-1/2}\boldsymbol{\Pi}_\nu\|_1 \leq 1$. From previous analysis we know that $\|\boldsymbol{D}^{1/2}\nabla h_t(\boldsymbol{z}_t^{(K_t)})\|_1 \leq \prod_{k=0}^{K_t-1}(1 - \frac{2(\alpha+\eta)}{1+\alpha+2\eta}\gamma_t^{(k)})\|\boldsymbol{D}^{1/2}\nabla h_t(\boldsymbol{z}_t^{(0)})\|_1$. We have a final upper-bound

$$h_t(\boldsymbol{z}_t^{(K_t)}) - h_t(\boldsymbol{x}_t^*) \leq \frac{(\eta+L)}{2(\eta+\alpha)^2} \cdot \prod_{k=0}^{K_t-1}\left(1 - \frac{2(\alpha+\eta)}{1+\alpha+2\eta}\gamma_t^{(k)}\right)^2 \|\nabla h_t^{1/2}(\boldsymbol{z}_t^{(0)})\|_1^2.$$

We derive the bound under the condition that $\eta = 1 - 2\alpha$. $\qquad\square$

# B   Related Work

**Personalized PageRank.** Personalized PageRank (PPR), initially introduced as a variant of Googles PageRank [12], was further studied in [25, 27]. A key property of PPR is that its important entries are concentrated near the source node, allowing for effective retrieval of relevant information even at a lower precision $\epsilon$. These important entries follow the power-law distribution [7, 23]. Computing $\epsilon$-approximate PPR vectors is fundamental for analyzing large-scale graph-structured data, with applications in local clustering [4, 5, 36, 44], modeling diffusion processes [7, 14, 21], and training node embeddings or graph neural networks [10, 15, 22, 29]. Further discussions on PPR-related problems can be found in [11, 26, 28, 49, 50].

There exist well-established iterative methods for computing PPR, particularly those based on solving linear systems [24, 43, 51]. Among these, the Conjugate Gradient Method (CGM) and the Chebyshev method [16] are commonly employed for solving the symmetrized form of Eq. (1). These approaches typically achieve a time complexity of $\tilde{\mathcal{O}}(m/\sqrt{\alpha})$, where $m$ is the number of edges. Further improvements have been made through symmetric diagonally dominant solvers [31, 45] and Anikin et al. [6] proposed an algorithm for the PageRank problem with a runtime complexity dependent on $|\mathcal{V}|$. However, we focus on local methods that avoid accessing the entire graph.

**Local algorithms and accelerations.** Unlike standard solvers, local solvers [3, 4, 9, 30, 42] exploit that the big entries of $\boldsymbol{\pi}$ are concentrated in a small part of the graph. Specifically, Andersen et al. [4] proposed the Approximate Personalized PageRank (APPR) algorithm, achieving a time complexity of $\mathcal{O}(1/(\alpha\epsilon))$. To further characterize the locality of $\boldsymbol{\pi}$, Fountoulakis et al. [20] introduced a variational formulation of (1) and applied a proximal gradient method to compute local estimates with time complexity of $\tilde{\mathcal{O}}(1/((\alpha + \mu^2)\epsilon))$, where $\mu > 0$ is a local conductance constant associated with $\mathcal{G}$. Both methods critically depend on the monotonic reduction of the residual or gradient to ensure convergence. The equivalence between APPR and other methods such as Gauss-Seidel and coordinate descent has been considered [32, 40, 46, 47] but does not focus on local analysis.

The question of whether an algorithm with time complexity $\tilde{\mathcal{O}}(1/(\sqrt{\alpha}\epsilon))$ can be achieved using methods such as FISTA [8], linear coupling [2], or other methods [1, 13] was raised in Fountoulakis and Yang [19]. However, the difficulty is that algorithms such as FISTA violate the monotonicity property where the volume accessed of per-iteration cannot be bounded properly. The work of Zhou et al. [52] proposes a locally evolving set process for localizing standard iterative methods for solving large-scale linear systems. However, their accelerated convergence rate framework strongly assumes that the residual has a geometric reduction rate, which could not be true in real-world settings. The work of Martínez-Rubio et al. [37] employs a nested APGD method, achieving a time complexity of $\tilde{\mathcal{O}}(|\mathcal{S}^*|\tilde{\mathrm{vol}}(\mathcal{S}^*)/\sqrt{\alpha} + |\mathcal{S}^*|\,\mathrm{vol}(\mathcal{S}^*))$ where $|\mathcal{S}^*| = |\mathrm{supp}(\boldsymbol{x}_\psi^*)|$ (with $\boldsymbol{x}_\psi^*$ being the optimal solution of Eq. (P2)) and $\tilde{\mathrm{vol}}(\mathcal{S}^*)$ denoting the number of edges in the induced subgraph from $\mathcal{S}^*$. The factor $|\mathcal{S}^*|$ appears in the bound due to the worst-case number of calls required for applying APGD. In contrast, our proposed framework introduces a novel local strategy that provably runs in $1/\sqrt{\alpha}$ outer-loop iterations. Furthermore, we incorporate the Catalyst framework [33, 34], which ensures that each iteration maintains locality, allowing the overall time complexity to be locally bounded.

# C   Implementation Details and More Experimental Results

Algorithm 3 and Algorithm 4 present LOCGD and LOCAPPR respectively. They iteratively update the active node set in a queue data structure $\mathcal{Q}$, ensuring a localized and efficient computation of the PPR estimate.

## C.1   Datasets and Preprocessing

In our main experiments, we evaluate the proposed method on a medium-scale graph *com-dblp* and four large-scale graphs *ogb-mag240m*, *ogbn-papers100M*, *com-friendster*, and *wiki-en21*. To further investigate the effectiveness and acceleration performance of our approach on different sizes of graphs, we conducted additional experiments on more graphs. we treat all 19 graphs as undirected with unit weights. We remove self-loops and keep the largest connected component when the graph is disconnected. Table 2 presents the key statistics of these datasets, including the number of nodes

**Algorithm 3** $\text{LocGD}(\varphi_t, \boldsymbol{y}^{(t-1)}, \eta, \alpha, \boldsymbol{b}, \mathcal{G})$

1: Initialize: $\boldsymbol{z} \leftarrow \boldsymbol{y}^{(t-1)}$
2: **if** $\|\nabla h_t^{1/2}(\boldsymbol{z}_t^{(0)})\|_1 = 0$ **then**
3:      **Return** $\boldsymbol{z}$
4: $\epsilon_t = \max\left\{\sqrt{\frac{(\mu+\eta)\varphi_t}{m}}, \frac{2(\eta+\alpha)\varphi_t}{\|\nabla h_t^{1/2}(\boldsymbol{z}_t^{(0)})\|_1}\right\}$
5: $\mathcal{Q} \leftarrow \{u : \epsilon_t\sqrt{d_u} \le |\nabla_u h_t(\boldsymbol{z})|\}$
6: $k = 0$
7: **while** $\mathcal{Q} \ne \emptyset$ **do**
8:      $\mathcal{S}_t^{(k)} = []$
9:      **while** $\mathcal{Q} \ne \emptyset$ **do**
10:         $u \leftarrow \mathcal{Q}.\text{dequeue}()$
11:         $\mathcal{S}_t^{(k)}.\text{append}\,((u, \nabla_u h_t(\boldsymbol{z})))$
12:         $\boldsymbol{z}_u \leftarrow \boldsymbol{z}_u - \frac{2}{1+\alpha+2\eta}\nabla_u h_t(\boldsymbol{z})$
13:         $\nabla h_t(\boldsymbol{z})_u \leftarrow 0$
14:      **for** $(u, \nabla_u h_t(\boldsymbol{z})) \in \mathcal{S}_t$ **do**
15:         **for** $v \in \mathcal{N}(u)$ **do**
16:             $\nabla_v h_t(\boldsymbol{z}) \leftarrow \nabla_v h_t(\boldsymbol{z}) + \frac{1-\alpha}{1+\alpha+2\eta}\frac{\nabla_u h_t(\boldsymbol{z})}{\sqrt{d_u d_v}}$
17:             **if** $|\nabla_v h_t(\boldsymbol{z})| \ge \epsilon_t\sqrt{d_v}$ and $v \notin \mathcal{Q}$ **then**
18:                $\mathcal{Q}.\text{enqueue}(v)$
19:      $k \leftarrow k+1$
20: **Return** $\boldsymbol{x}^{(t)} \leftarrow \boldsymbol{z}$.

**Algorithm 4** $\text{LocAPPR}(\varphi_t, \boldsymbol{y}^{(t-1)}, \eta, \alpha, \boldsymbol{b}, \mathcal{G})$

1: Initialize: $\boldsymbol{z} \leftarrow \boldsymbol{y}^{(t-1)}$
2: **if** $\|\nabla h_t^{1/2}(\boldsymbol{z}_t^{(0)})\|_1 = 0$ **then**
3:      **Return** $\boldsymbol{z}$
4: $\epsilon_t \leftarrow \max\left\{\sqrt{\frac{(\mu+\eta)\varphi_t}{m}}, \frac{2(\eta+\alpha)\varphi_t}{\|\nabla h_t^{1/2}(\boldsymbol{z}_t^{(0)})\|_1}\right\}$
5: $\mathcal{Q} \leftarrow \{u : \epsilon_t\sqrt{d_u} \le |\nabla_u h_t(\boldsymbol{z})|\}$
6: **while** $\mathcal{Q} \ne \emptyset$ **do**
7:      $u \leftarrow \mathcal{Q}.\text{dequeue}()$
8:      **if** $|\nabla_u h_t(\boldsymbol{z})| < \epsilon_t\sqrt{d_u}$ **then**
9:         **continue**
10:      $\delta \leftarrow \nabla_u h_t(\boldsymbol{z})$
11:      $z_u \leftarrow z_u - \frac{2\delta}{1+\alpha+2\eta}$
12:      **for** $v \in \mathcal{N}(u)$ **do**
13:         $\nabla_v h_t(\boldsymbol{z}) \leftarrow \nabla_v h_t(\boldsymbol{z}) + \frac{1-\alpha}{1+\alpha+2\eta} \cdot \frac{\delta}{\sqrt{d_u}\sqrt{d_v}}$
14:         **if** $|\nabla_v h_t(\boldsymbol{z})| \ge \epsilon_t\sqrt{d_v}$ **and** $v \notin \mathcal{Q}$ **then**
15:             $\mathcal{Q}.\text{enqueue}(v)$
16:      **if** $|\nabla_u h_t(\boldsymbol{z})| \ge \epsilon_t\sqrt{d_u}$ **and** $u \notin \mathcal{Q}$ **then**
17:         $\mathcal{Q}.\text{enqueue}(u)$
18: **Return** $\boldsymbol{x}^{(t)} \leftarrow \boldsymbol{z}$

$(n)$ and edges $(m)$. The largest graph in our extended experiments contains up to 200 million nodes and 1 billion edges, as shown in Table 2.

## C.2   Problem Settings and Baseline Methods

For solving Equation (P1) and (P2) on 19 graphs, we randomly select 5 source nodes $s$ from each graph. The damping factor $\alpha$ and convergence threshold $\epsilon$ were fixed at $\alpha = 0.1$ and $\epsilon = 1 \times 10^{-6}$ throughout all experiments unless otherwise specified. To compare with AESP-LOCAPPR and AESP-LOCGD, we primarily consider LOCGD, APPR, LOCCH, FISTA, and ASPR methods as baselines. All our methods are implemented in Python with numba acceleration tools. Both ASPR and FISTA use a precision of $\tilde{\epsilon} = 0.1$ and the parameter $\hat{\epsilon} = \epsilon/(1 + \tilde{\epsilon})$ as suggested in [20].

## C.3   Additional experimental results

**Comparison of baseline methods.** Fig. 6 compares the convergence behaviors of AESP-LOCAPPR and ASPR for the *com-dblp* graph, with parameters $\alpha = 0.01$ and $\epsilon = 0.1/n$. As evidenced by the early-stage iterations in the subplots, AESP-LOCAPPR achieves significantly faster convergence compared to ASPR. Although ASPR guarantees monotonic decrease in the $\ell_1$-norm of gradients, this property comes at the expense of requiring increasingly iterative points, which consequently reduces computational efficiency.

Fig. 7 demonstrates the superior convergence behavior of AESP-LOCAPPR, AESP-LOCGD compared to baseline methods (LOCCH, and FISTA) on the *com-dblp* graph, with parameters $\alpha = 0.01$ and $\epsilon = 0.1/n$. AESP-LOCAPPR and AESP-LOCGD has rapid error reduction within the first $1 \times 10^7$ operations.

Fig. 8 presents results on the estimation error reduction for 4 datasets: *wiki-talk*, *ogbn-arxiv*, *com-youtube*, and *com-dblp*. The acceleration effect of the AESP method is particularly evident in the initial stages.

**Full results of 19 graphs.** Fig. 9 demonstrates the performance comparison of our proposed algorithm against baseline methods (APPR, APPR Opt, and LocGD) across 19 graphs of varying scales,

Table 2: Dataset Statistics

| Notations | Dataset Name | $n$ | $m$ |
|---|---|---|---|
| $\mathcal{G}_1$ | *as-skitter* | 1694616 | 11094209 |
| $\mathcal{G}_2$ | *cit-patent* | 3764117 | 16511740 |
| $\mathcal{G}_3$ | *com-dblp* | 317080 | 1049866 |
| $\mathcal{G}_4$ | *com-friendster* | 65608366 | 1806067135 |
| $\mathcal{G}_5$ | *com-lj* | 3997962 | 34681189 |
| $\mathcal{G}_6$ | *com-orkut* | 3072441 | 117185083 |
| $\mathcal{G}_7$ | *com-youtube* | 1134890 | 2987624 |
| $\mathcal{G}_8$ | *ogb-mag240m* | 244160499 | 1728364232 |
| $\mathcal{G}_9$ | *ogbl-ppa* | 576039 | 21231776 |
| $\mathcal{G}_{10}$ | *ogbn-arxiv* | 169343 | 1157799 |
| $\mathcal{G}_{11}$ | *ogbn-mag* | 1939743 | 21091072 |
| $\mathcal{G}_{12}$ | *ogbn-papers100M* | 111059433 | 1615685450 |
| $\mathcal{G}_{13}$ | *ogbn-products* | 2385902 | 61806303 |
| $\mathcal{G}_{14}$ | *ogbn-proteins* | 132534 | 39561252 |
| $\mathcal{G}_{15}$ | *soc-lj1* | 4843953 | 42845684 |
| $\mathcal{G}_{16}$ | *soc-pokec* | 1632803 | 22301964 |
| $\mathcal{G}_{17}$ | *sx-stackoverflow* | 2584164 | 28183518 |
| $\mathcal{G}_{18}$ | *wiki-en21* | 6216199 | 160823797 |
| $\mathcal{G}_{19}$ | *wiki-talk* | 2388953 | 4656682 |

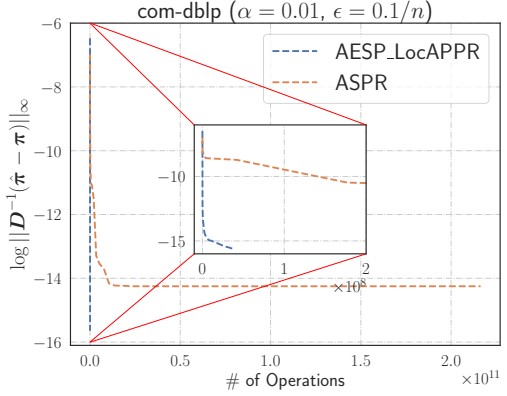

Figure 6: Comparison of of AESP-LOCAPPR versus ASPR, $\log \|\boldsymbol{D}^{-1}(\hat{\boldsymbol{\pi}} - \boldsymbol{\pi})\|_\infty$ over the operations on the *com-dblp* graph with $\alpha = 0.01$ and $\epsilon = 0.1/n$ (Insets Show Early-stage Iteration Details)

Figure 7: Comparison of $\log \|\boldsymbol{D}^{-1}(\hat{\boldsymbol{\pi}} - \boldsymbol{\pi})\|_\infty$ over the operations on the *com-dblp* graph with $\alpha = 0.01$ and $\epsilon = 0.1/n$, illustrating the performance of AESP-LOCAPPR, AESP-LOCGD, LOCCH, and FISTA.

while Table 3 and 4 present the corresponding operation counts and running times across different graphs. The results clearly show that AESP-based methods (AESP-LOCAPPR and AESP-LOCGD) achieve significantly faster error reduction during initial iterations, highlighting their superior convergence properties, while maintaining robust performance across all graph scales from small to extremely large graphs, which substantiates the algorithmic robustness.

Table 4 reveals that our algorithm exhibits suboptimal performance on certain graphs, which can be attributed to the computational overhead introduced by the iterative parameter initialization process (particularly for $\varphi_t$ and $\epsilon_t$ in inner-loops). While this initialization overhead marginally increases runtime in some cases, it crucially enables the superior convergence rates. What's more, this trade-

Table 3: Operations Needed for five local solvers on 19 graphs datasets.

| Graph | APPR | APPR Opt | LocGD | AESP-LocGD | AESP-LocAPPR |
|---|---|---|---|---|---|
| as-skitter | 3.06e+06 | 1.39e+06 | 1.94e+06 | 1.26e+06 | 1.00e+06 |
| cit-patent | 3.86e+06 | 1.81e+06 | 2.62e+06 | 1.45e+06 | 1.27e+06 |
| com-dblp | 6.07e+06 | 3.18e+06 | 4.66e+06 | 1.63e+06 | 1.43e+06 |
| com-friendster | 8.35e+05 | 3.20e+05 | 3.87e+05 | 6.65e+05 | 6.57e+05 |
| com-lj | 1.73e+06 | 7.14e+05 | 9.69e+05 | 8.18e+05 | 7.70e+05 |
| com-orkut | 1.32e+06 | 5.72e+05 | 7.06e+05 | 8.29e+05 | 8.19e+05 |
| com-youtube | 2.27e+06 | 1.33e+06 | 1.60e+06 | 1.27e+06 | 1.09e+06 |
| ogb-mag240m | 1.92e+06 | 8.46e+05 | 9.86e+05 | 7.54e+05 | 7.01e+05 |
| ogbl-ppa | 8.19e+05 | 4.36e+05 | 4.53e+05 | 7.45e+05 | 7.33e+05 |
| ogbn-arxiv | 1.20e+07 | 5.47e+06 | 8.99e+06 | 2.59e+06 | 2.25e+06 |
| ogbn-mag | 9.23e+05 | 3.89e+05 | 4.45e+05 | 6.65e+05 | 6.33e+05 |
| ogbn-papers100M | 1.18e+06 | 5.05e+05 | 5.86e+05 | 8.38e+05 | 7.98e+05 |
| ogbn-products | 2.00e+06 | 9.73e+05 | 1.30e+06 | 9.11e+05 | 8.89e+05 |
| ogbn-proteins | 7.55e+05 | 7.73e+05 | 7.60e+05 | 9.20e+05 | 9.20e+05 |
| soc-lj1 | 2.45e+06 | 1.09e+06 | 1.53e+06 | 1.03e+06 | 9.51e+05 |
| soc-pokec | 1.58e+06 | 7.13e+05 | 7.98e+05 | 9.38e+05 | 8.95e+05 |
| sx-stackoverflow | 9.08e+05 | 3.47e+05 | 4.39e+05 | 5.18e+05 | 4.88e+05 |
| wiki-en21 | 7.19e+05 | 2.18e+05 | 2.27e+05 | 5.47e+05 | 5.36e+05 |
| wiki-talk | 1.39e+06 | 7.76e+05 | 9.83e+05 | 6.79e+05 | 5.42e+05 |

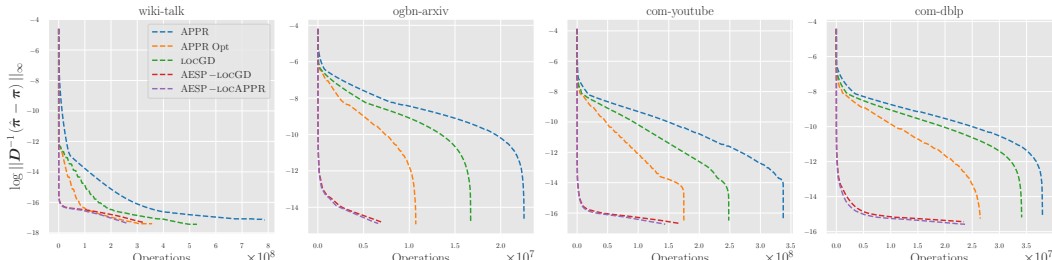

Figure 8: Performance comparison of five local solvers across four graphs: *wiki-talk*, *ogbn-arxiv*, *com-youtube*, and *com-dblp* (with parameters $\alpha = 0.01$ and $\epsilon = 0.1/n$).

off between initialization overhead and convergence acceleration becomes increasingly favorable as the graph size grows.

**Initialization of $z_t^{(0)}$.** Fig. 2 presents a comprehensive comparison of different initialization strategies for the inner-loop optimization in AESP-LocAPPR, where we identify $z_t^{(0)} = y^{(t-1)}$ as the recommended choice based on empirical evidence. This supplementary investigation further evaluates the performance of AESP-LOCAPPR and AESP-LOCGD under varying initialization approaches ($y^{(t-1)}$ versus momentum-free $x^{(t-1)}$ versus zero-initialization) with fixed parameters $\alpha = 0.01$ and $\epsilon = 0.1/n$. Fig. 10 demonstrating that the proposed $y^{(t-1)}$ initialization yields significantly superior convergence characteristics compared to both the $x^{(t-1)}$-based and cold-start alternatives. While all three initialization strategies ($y^{(t-1)}$, $x^{(t-1)}$ and zero-initialization) exhibit comparable performance during the initial iterations, the $y^{(t-1)}$-based approach establishes substantial superiority in later optimization stages.

Table 4: Running times

| Graph | APPR | APPR Opt | LocGD | AESP-LocGD | AESP-LocAPPR |
|---|---|---|---|---|---|
| as-skitter | 6.90e-01 | 3.18e-01 | 4.24e-01 | 5.62e+00 | 5.63e+00 |
| cit-patent | 5.05e-01 | 2.50e-01 | 9.97e-02 | 6.08e-02 | 1.87e-01 |
| com-dblp | 6.38e-01 | 3.28e-01 | 7.74e-02 | 2.32e-02 | 1.41e-01 |
| com-friendster | 1.60e+01 | 7.32e+00 | 9.93e+00 | 3.14e+02 | 3.15e+02 |
| com-lj | 1.18e-01 | 4.93e-02 | 2.70e-02 | 3.38e-02 | 6.87e-02 |
| com-orkut | 3.22e-02 | 1.47e-02 | 1.11e-02 | 1.74e-02 | 2.77e-02 |
| com-youtube | 2.90e-01 | 1.72e-01 | 3.55e-02 | 2.60e-02 | 1.33e-01 |
| ogb-mag240m | 6.17e-01 | 1.75e-01 | 3.56e-01 | 2.18e-01 | 2.40e-01 |
| ogbl-ppa | 2.00e-02 | 8.77e-03 | 5.30e-03 | 7.81e-03 | 1.78e-02 |
| ogbn-arxiv | 1.10e+00 | 5.04e-01 | 1.35e-01 | 3.08e-02 | 1.98e-01 |
| ogbn-mag | 4.46e-02 | 2.29e-02 | 1.09e-02 | 1.46e-02 | 4.47e-02 |
| ogbn-papers100M | 1.30e-01 | 6.09e-02 | 5.19e-02 | 1.81e-01 | 2.17e-01 |
| ogbn-products | 7.91e-02 | 3.69e-02 | 2.85e-02 | 2.48e-02 | 3.98e-02 |
| ogbn-proteins | 3.84e-03 | 3.71e-03 | 2.45e-03 | 2.50e-03 | 4.55e-03 |
| soc-lj1 | 1.73e-01 | 7.59e-02 | 4.09e-02 | 4.48e-02 | 9.75e-02 |
| soc-pokec | 9.18e-02 | 4.07e-02 | 2.04e-02 | 2.31e-02 | 5.93e-02 |
| sx-stackoverflow | 1.37e+00 | 5.44e-01 | 7.01e-01 | 2.90e+00 | 4.39e+00 |
| wiki-en21 | 2.71e-02 | 8.66e-03 | 6.85e-03 | 2.40e-02 | 3.82e-02 |
| wiki-talk | 2.40e-01 | 1.53e-01 | 3.08e-02 | 2.15e-02 | 1.06e-01 |

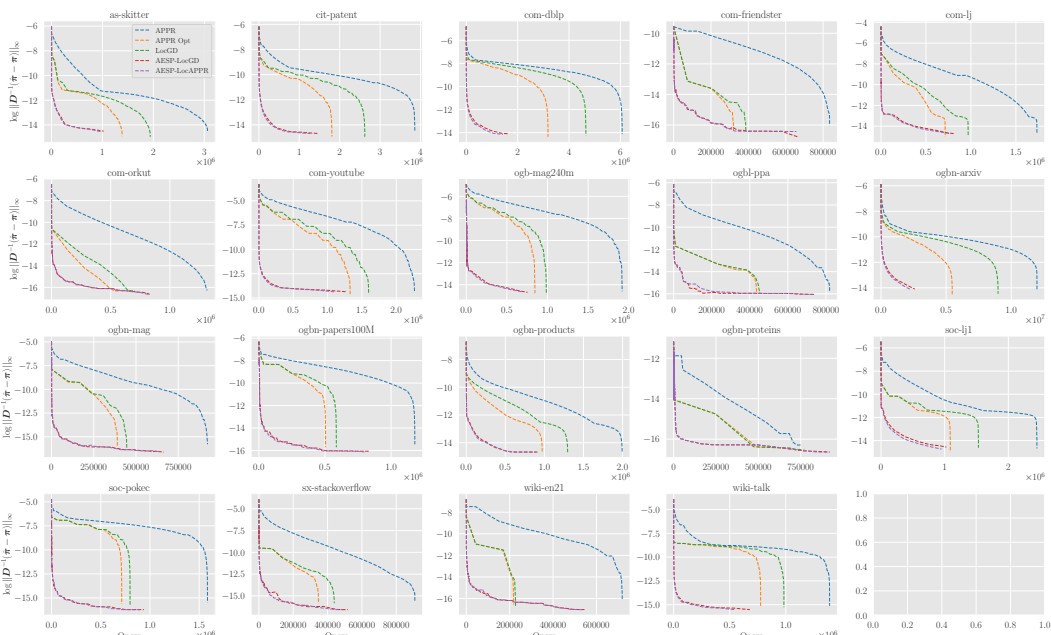

Figure 9: Comparison of five local solvers over 19 graphs

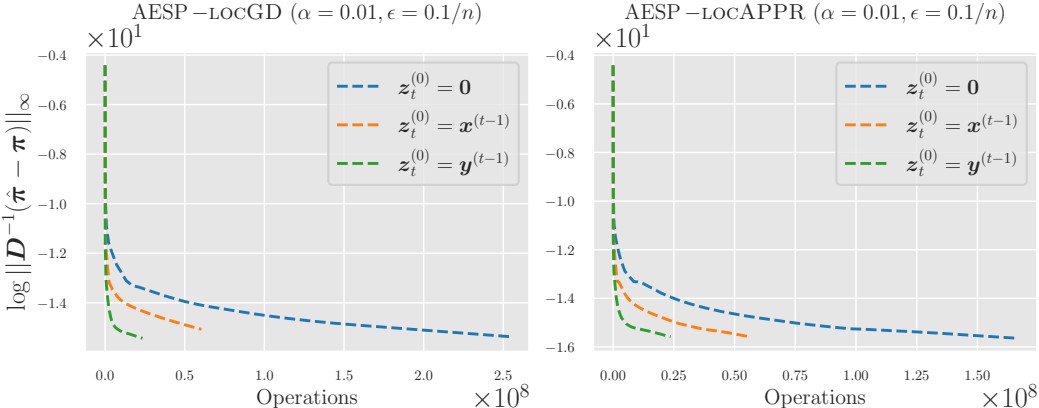

Figure 10: Comparison of AESP-LOCAPPR and AESP-LOCGD with three initializations on the graph *com-dblp* (with parameters $\alpha = 0.01$ and $\epsilon = 0.1/n$).

