# OpenReview forum: "Accelerated Evolving Set Processes for Local PageRank Computation"
_NeurIPS.cc/2025/Conference — NeurIPS 2025 poster_

### Official Review · Reviewer_qYeh · 2025-06-26

**Clarity:** 3
**Significance:** 3
**Originality:** 2
**Rating:** 4
**Confidence:** 3

**Summary:**

The paper proposes a novel framework called AESP (Accelerated Evolving Set Process) to approximate the Personalized PageRank (PPR) vectors in a faster way using local computations. AESP has a complexity of O(min{ R^2/(sqrt(α)ϵ^2) , m/sqrt(α) }), where R is a constant depending on the evolving set process and m, α, ϵ are the number of edges, the damping factor, the precision, respectively. Interestingly, when 1/ϵ^2 ≪ m the algorithm computes an ϵ-approximation of the PPR vector with an time  complexity independent of the underlying graph size. The authors show that AESP can accelerate any existing local solver, e.g. APPR, by (1) reformulating the PPR computation as a convex quadratic problem, (2) applying proximal point acceleration with an evolving set process to avoid accessing the whole graph and to solve easier subproblems, (3) using the local solver for each subproblem, (4) using momentum [39] to update the estimate with less iterations (from 1/α to 1/sqrt(α)).

**Questions:**

If one compares with O(1/(αϵ)) from APPR where are the wins and losses? How does the ϵ^2 in the proposed method here affect performance compared to APPR and other methods with just  ϵ involved?

How does the R in the complexity scale with dumping factor?

How does the method perform against other methods from the literature e.g. Ref.[37] ?

**Ethical Concerns:**

["NO or VERY MINOR ethics concerns only"]

**Final Justification:**

The authors have made a good effort in answering my main concerns, especially the scaling of R with respect to a, and the effect of epsilon^2. I will keep my score, for the following reasons.

1. Given the dependence of R on alpha<0.01 this should somehow be visible in the complexity expression. Strictly speaking R is not a global constant but a function of alpha, and this behavior is not sufficient to be shown empirically on a couple of datasets.

2. Aiming to improve the complexity over the alpha parameter, another aspect of the algorithm gets worse. Specifically, O(1/sqrt(alpha)) comes at the cost of a looser dependence on sparsity O(1/epsilon^2). The paper does not answer the question, can we get better without worsening another aspect or have we reached the limits?

I have appreciated the contribution of the paper, but I believe these two aspects should have been discussed and analyzed with more rigor, this is why I keep the Borderline accept score 4.

**Limitations:**

The limitations of the current algorithmic proposal have not been sufficiently discussed, although to be fair there are some interesting ablation studies. Furthermore, some cases in large graphs where the method underperforms compared to baselines have been identified, but I was not convinced from the explanations.

**Quality:**

3

**Strengths And Weaknesses:**

Pros: The paper answers an open conjecture [19, 52] about achievable time complexity for the PPR computation. The problem is well introduced and the presentation is rigorous. Theoretical analysis with mathematical proofs are provided. The evaluation on real-world large graphs has been conducted, with a comparison with baseline algorithms (APPR and LocGD from [52]). Experiments include ablations (different initialization methods, different α values) and are consistent with the theory. They show acceleration in error reduction during initial operations and overall speedup.

Cons: The constant R depends not only on graph parameters, and R is not explicitly bounded. The authors claim that given the empirical results, it still seems to be small and show in Fig.3 a value of at most 1.4. However these values are shown for alpha = 0.1 and 0.01. Even for these two values of damping factor, there is an increase in R for lower alpha. This means that since R(a) then this should be also incorporated in the claimed complexity. How does R grow with smaller alpha?
More generally, a study of the algorithm over the range of the damping factor is missing. For example, the authors assume that α < 0.5, but is the algorithm also robust to high α values?

Another issue that troubles me is that the complexity has a denominator in ϵ^2 instead of ϵ. Certainly there is a benefit to go from 1/α to 1/(sqrt(α) but there is penalty in having the squared precision in the denominator. This is not discussed sufficiently.

Furthermore, the comparison against APPR, LocGD and FISTA give interesting results, however, there are other state-of-the-art methods for approximating PageRank, e.g. it is missing comparison against Ref.[37] although extensively discussed in the Related Works section.

---

> ### Author Rebuttal · Authors · 2025-07-30
>
> **Q1.** If one compares with $O(1/(\alpha\epsilon))$ from APPR where are the wins and losses? How does the $\epsilon^2$ in the proposed method here affect performance compared to APPR and other methods with just $\epsilon$ involved?
>
> **A1.** Thank you for raising this important question regarding the trade-offs in our theoretical guarantees.
>
> 1) **Practical Impact of $\epsilon$**: In PPR approximation, the precision parameter $\epsilon$ is typically not set too small, as an excessively tight tolerance would cause local push-based methods (e.g., APPR) to degenerate into global algorithms due to an overgrown active set. Thus, in practice, optimizing the dependence on $\alpha$ is often more critical than the $\epsilon$-dependence.
>
> 2) **Trade-off Between $\alpha$ and $\epsilon$**: As discussed in Martínez-Rubio et al. [1], our bound reflects a fundamental trade-off: while we improve the condition number dependence to $O(1/\sqrt{\alpha})$, this comes at the cost of a looser $O(1/\epsilon^2)$ dependence on sparsity. This aligns with existing conjectures about such trade-offs in iterative methods. Whether a tighter $\epsilon$-dependence (e.g., $O(1/\epsilon)$) can be achieved without sacrificing the gains in $\alpha$ remains an open question, and we intend to explore this direction in future work.
>
>  [1] Martínez-Rubio, David, Elias Wirth, and Sebastian Pokutta. "Accelerated and sparse algorithms for approximate personalized PageRank and beyond." The Thirty Sixth Annual Conference on Learning Theory. PMLR, 2023.
>
> **Q2.** How does the $R$ in the complexity scale with dumping factor $\alpha$?
>
> **A2.**  We sincerely appreciate the reviewer's insightful comments regarding the scaling behavior of $R$ with respect to the damping factor $\alpha$. To address these concerns, we have conducted additional experiments (see table below) examining $R$ across both small $\alpha$ values and values approaching 0.5, using the com-dblp graph (with $\epsilon=1e-8$; results are similar for other graphs). Our findings demonstrate that: (1) for $0.01 \leq \alpha < 0.5$, **$R$ remains consistently at 1**, indicating our algorithm's robustness for larger $\alpha$ values; (2) **when $\alpha < 0.01$, $R$ exhibits a gradual increase with decreasing $\alpha$, but crucially this growth follows a logarithmic scaling of $\log(1/\alpha)$** - substantially milder than the $1/\alpha$ dependence. These empirical results validate the reasonableness of our treatment of $R$ in the complexity analysis while providing a more comprehensive characterization of its behavior across the full range of $\alpha$ values.
>
> **Q3.** How does the method perform against other methods from the literature e.g. Ref.[37] ?
>
> **A3.** Thank you for this important question regarding comparative performance. Regarding the comparison with Ref.[37]'s **ASPR** algorithm, we have included experimental comparisons in **Appendix C.3 (Figure 6)** due to space constraints in the main text. It should be noted that while ASPR maintains the desirable property of guaranteed monotonic decrease in the gradient's $\ell_1$-norm, this advantage comes at the cost of requiring progressively more iterative points, which ultimately leads to reduced computational efficiency compared to our proposed method. The detailed comparative results demonstrate this trade-off between theoretical guarantees and practical performance.
>
> | Method       | α      | 1/α      | log(1/α) | R    |
> |-------------|--------|----------|----------|------|
> | AESP-LocAPPR | 1e-5   | 1.00e+5  | 11.51    | 2.83 |
> | AESP-LocAPPR | 5e-5   | 2.00e+4  | 9.90     | 2.52 |
> | AESP-LocAPPR | 1e-4   | 1.00e+4  | 9.21     | 2.43 |
> | AESP-LocAPPR | 5e-4   | 2.00e+3  | 7.60     | 1.99 |
> | AESP-LocAPPR | 1e-3   | 1.00e+3  | 6.91     | 1.59 |
> | AESP-LocAPPR | 2e-3   | 5.00e+2  | 6.21     | 1.34 |
> | AESP-LocAPPR | 5e-3   | 2.00e+2  | 5.30     | 1.17 |
> | AESP-LocAPPR | 7e-3   | 1.43e+2  | 4.96     | 1.00 |
> | AESP-LocAPPR | 2e-2   | 5.00e+1  | 3.91     | 1.00 |
> | AESP-LocAPPR | 4e-2   | 2.50e+1  | 3.22     | 1.00 |
> | AESP-LocAPPR | 5e-2   | 2.00e+1  | 3.00     | 1.00 |
> | AESP-LocAPPR | 6e-2   | 1.67e+1  | 2.81     | 1.00 |
> | AESP-LocAPPR | 8e-2   | 1.25e+1  | 2.53     | 1.00 |
> | AESP-LocAPPR | 1e-1   | 1.00e+1  | 2.30     | 1.00 |
> | AESP-LocAPPR | 2e-1   | 5.00e+0  | 1.61     | 1.00 |
> | AESP-LocAPPR | 3e-1   | 3.33e+0  | 1.20     | 1.00 |
> | AESP-LocAPPR | 4e-1   | 2.50e+0  | 0.92     | 1.00 |
> | AESP-LocAPPR | 4.5e-1 | 2.22e+0  | 0.80     | 1.00 |
> | AESP-LocAPPR | 4.9e-1 | 2.04e+0  | 0.71     | 1.00 |
> | AESP-LocGD   | 1e-5   | 1.00e+5  | 11.51    | 2.93 |
> | AESP-LocGD   | 5e-5   | 2.00e+4  | 9.90     | 2.57 |
> | AESP-LocGD   | 1e-4   | 1.00e+4  | 9.21     | 2.45 |
> | AESP-LocGD   | 5e-4   | 2.00e+3  | 7.60     | 1.88 |
> | AESP-LocGD   | 1e-3   | 1.00e+3  | 6.91     | 1.60 |
> | AESP-LocGD   | 2e-3   | 5.00e+2  | 6.21     | 1.38 |
> | AESP-LocGD   | 5e-3   | 2.00e+2  | 5.30     | 1.16 |
> | AESP-LocGD   | 7e-3   | 1.43e+2  | 4.96     | 1.00 |
> | AESP-LocGD   | 2e-2   | 5.00e+1  | 3.91     | 1.00 |
> | AESP-LocGD   | 4e-2   | 2.50e+1  | 3.22     | 1.00 |
> | AESP-LocGD   | 5e-2   | 2.00e+1  | 3.00     | 1.00 |
> | AESP-LocGD   | 6e-2   | 1.67e+1  | 2.81     | 1.00 |
> | AESP-LocGD   | 8e-2   | 1.25e+1  | 2.53     | 1.00 |
> | AESP-LocGD   | 1e-1   | 1.00e+1  | 2.30     | 1.00 |
> | AESP-LocGD   | 2e-1   | 5.00e+0  | 1.61     | 1.00 |
> | AESP-LocGD   | 3e-1   | 3.33e+0  | 1.20     | 1.00 |
> | AESP-LocGD   | 4e-1   | 2.50e+0  | 0.92     | 1.00 |
> | AESP-LocGD   | 4.5e-1 | 2.22e+0  | 0.80     | 1.00 |
> | AESP-LocGD   | 4.9e-1 | 2.04e+0  | 0.71     | 1.00 |

---

> ### Comment · Reviewer_qYeh · 2025-08-04
> **Reply to authors' rebuttal**
>
> Thank you for the detailed response. The authors have made a good effort in answering my main concerns, especially the scaling of R with respect to a,  and the effect of epsilon^2. I will keep my score 4, for the following reasons.
>
> Given the dependence of R on alpha<0.01 this should somehow be visible in the complexity expression. Strictly speaking R is not a global constant but a function of alpha, and this behavior is not sufficient to be shown empirically on some datasets. How do we know how the R will behave in other datasets? Also, why is the change of monotonicity from 0.01 and lower? Where does this come from? Why not 0.012 or 0.005?
>
> Furthermore, the authors answer about the tradeoff in complexity that "while we improve the condition number dependence to O(1/sqrt(alpha)), this comes at the cost of a looser dependence on sparsity O(1/epsilon^2), following existing conjectures from the literature. However, this is exactly the point that leaves questions about the importance of the paper and the current method, because aiming to improve the complexity over the alpha parameter, another aspect of the algorithm gets worse. The paper does not answer the question, can we get better without worsening another aspect or have we reached the limits?
>
> Finally, regarding comparison with Ref.[37] I appreciate the inclusion of the experimental comparison in the Appendix, but this algorithm has already guaranteed the monotonic decrease in the gradient's norm, something that you have also claimed in your paper. I would have expected a more direct comparison against this algorithm in the main body of the paper, something which should also justify the importance of your own contribution in performance, i.e. what is actually new in the performance of the method you propose. These comments do not mean that I have not appreciated the contribution of the paper, this is why I keep the Borderline accept.

---

> > ### Author Response · Authors · 2025-08-09
> >
> > We thank the reviewer for their valuable feedback.
> >
> > 1. On $R(\alpha)$ dependence: While our empirical results during the rebuttal phase show monotonicity changes around $\alpha=0.01$, we acknowledge the need for further theoretical analysis of this threshold and additional validation across datasets.
> >
> > 2. Complexity trade-off: We agree the $O(1/\sqrt{\alpha})$ vs. $O(1/\epsilon^2)$ trade-off requires deeper investigation. We will clarify whether this represents a fundamental limit or if tighter bounds are possible under stronger assumptions.
> >
> > 3. Comparison with [37]: We will move the ASPR comparison to the main text, highlighting our method's superior early convergence. We will additionally include comparisons of active set evolution to demonstrate AESP's operational advantages

---

### Official Review · Reviewer_Ev2P · 2025-06-30

**Clarity:** 2
**Significance:** 3
**Originality:** 3
**Rating:** 5
**Confidence:** 4

**Summary:**

This paper introduces the Accelerated Evolving Set Process (AESP), a novel framework designed to accelerate the Approximate Personalized PageRank (APPR) algorithm. The core of AESP lies in its formulation based on inexact accelerated proximal point iterations. The authors provide a rigorous theoretical analysis, establishing sufficient conditions for convergence and detailing the method's time complexity. A key theoretical finding is that AESP achieves faster convergence rates than traditional APPR in the early stages. Extensive experiments on large-scale graphs, with node counts ranging from 6 million to 240 million, empirically validate this theoretical advantage, demonstrating that AESP-based methods significantly outperform baseline approaches in terms of early-stage convergence speed.

**Questions:**

Thank you for your work on this interesting problem. The paper is promising, particularly due to its strong theoretical contributions. However, I have a few key concerns regarding the clarity of the presentation and the experimental evaluation that prevent me from giving a higher score at this stage. My questions and suggestions below are intended to clarify these points. My evaluation score will be raised if you can address them convincingly in your rebuttal.
1.	On the clarity of proposed AESP Framework:
My primary concern is with the presentation of the AESP framework in Section 2.2. I found the exposition overly complex and the notation difficult to follow, which obscures the core intuition behind the method. To address this, I strongly encourage the authors to Consider adding a simple, illustrative figure or a running example that walks the reader through one or two iterations of the algorithm. This would significantly improve comprehensibility. My score would increase substantially if the authors can provide a clearer, more accessible explanation of their framework in the rebuttal.
2.	On the Robustness of the experimental evaluation:
The experimental section provides evidence for AESP's fast early-stage convergence. However, the choice of baseline methods seems somewhat outdated, which limits the persuasiveness of the results. To claim superiority, a comparison against the current state-of-the-art is essential. Could the authors please justify their choice of the current baselines? Are there specific reasons (e.g., implementation availability, direct comparability) why these were chosen over more recent methods from top conferences from the last 3-5 years?
3.	On the characterization of early-stage convergence:
The paper's key empirical results emphasize that AESP's primary advantage is its faster early-stage convergence. This raises a question about the method's performance envelope. Could the authors elaborate on the behavior of AESP in the later stages of convergence? Specifically, does the performance advantage over baselines diminish or disappear as the approximation error tolerance becomes very small? Is there a crossover point after which traditional methods might become more efficient?

**Ethical Concerns:**

["NO or VERY MINOR ethics concerns only"]

**Final Justification:**

I am fine with the response. I keep the decision and suggest to accept this work.

**Limitations:**

yes

**Quality:**

3

**Strengths And Weaknesses:**

Strengths:
Significant Theoretical Contributions: The paper provides a rigorous theoretical foundation for the proposed method. A key strength is the inclusion of not only a detailed time complexity analysis but also formal convergence guarantees. Generality and Extensibility: The authors demonstrate the versatility of the AESP framework by analyzing its application to several variational forms of the PPR problem. This analysis effectively shows how the proposed framework can be extended beyond the standard PPR setting, broadening its potential impact. Practical and Meaningful Problem: The work addresses the important and practical challenge of accelerating a general formulation of the Personalized PageRank problem. Developing faster algorithms for this fundamental task is a meaningful contribution to the field of large-scale graph analysis.

Weaknesses:
Clarity and Presentation: The presentation of the proposed framework is overly complex, which significantly hinders readability and makes the core ideas difficult to follow. Choice of Baselines in Experiments: The persuasiveness of the experimental results is somewhat limited by the choice of baseline methods. The selected baselines (e.g., APPR, APPR-opt) could be considered dated. A more robust evaluation against more recent PPR approximation algorithms would be necessary to convincingly demonstrate the superiority of the proposed AESP framework.

---

> ### Author Rebuttal · Authors · 2025-07-30
>
> **Q1.** **The concern about the clarity of proposed AESP Framework**
>
> **A1.** We appreciate this constructive suggestion and will add an illustrative figure to enhance the clarity of the AESP framework. The figure will demonstrate: (1) a simple graph with several nodes including the source node; (2) the first inner loop's local method (locGD or APPR) iterations, where the initial active set $\mathcal{S}_1^{(0)}$ contains only the source node, with subsequent iterations showing $\mathcal{S}_1^{(t)}$ first expanding then contracting; (3) the second inner loop recomputing the initial active set $\mathcal{S}_2^{(0)}$ based on the previous solution, where the shrinking tolerance $\epsilon_k$ leads to progressively larger active sets. Hope this will effectively demonstrate the dynamic evolution of active sets throughout the iterative process.
>
> **Q2.** **On the Robustness of the experimental evaluation**
>
> **A2.** We appreciate the reviewer's valuable feedback regarding baseline selection. Our choice of baseline methods was motivated by several key considerations: (1) Since AESP's inner loop employs either locGD or APPR, we directly compared against these fundamental methods to demonstrate the acceleration benefits; (2) We have indeed included comparisons with recent methods ASPR[1] (2023), LocCH[2] (2024) and maintained FISTA[3] (2009) as it remains a relevant benchmark, with these results presented in Appendix C.3 (Figures 6-7) due to space constraints; (3) The results clearly show AESP's superior early-stage convergence with rapid error reduction;
>
> [1] Martínez-Rubio, David, Elias Wirth, and Sebastian Pokutta. "Accelerated and sparse algorithms for approximate personalized PageRank and beyond." The Thirty Sixth Annual Conference on Learning Theory. PMLR, 2023.
>
> [2] Zhou, Baojian, et al. "Iterative methods via locally evolving set process." Advances in Neural Information Processing Systems 37 (2024): 141528-141586.
>
> [3] Beck, Amir, and Marc Teboulle. "A fast iterative shrinkage-thresholding algorithm for linear inverse problems." SIAM journal on imaging sciences 2.1 (2009): 183-202.
>
> **Q3.** **On the characterization of early-stage convergence**
>
> **A3.** We sincerely appreciate this insightful question regarding AESP's convergence behavior. Our analysis reveals that while AESP demonstrates significant early-stage convergence advantages, its time complexity does increase during updates as $\epsilon_t$ decreases. We conducted new experiments on the **ogbn-arxiv** dataset ($\alpha=0.1$, a random source node, different $\epsilon$ values range from $10^{-6}$ to $10^{-10}$). We found that, when AESP maintains a constant condition number for subproblems, its convergence rate in later stages becomes less pronounced compared to non-accelerated methods like APPR-Opt. This occurs because the subproblem precision $\epsilon_t$ eventually becomes smaller than the original problem's precision requirement. However, we emphasize that our outer loop incorporates an early-stopping mechanism (Algorithm 1, Line 7), which ensures comparable total operations between methods. Thus, while AESP's advantage becomes less pronounced at extremely small $\epsilon$ (the locality is not significant, so all local methods do not have any advantages) values, it maintains overall operational efficiency comparable to baseline methods throughout the convergence process. Whether one can optimize/accelerate the convergence of the later stage is an interesting problem to explore.

---

> > ### Comment · Reviewer_Ev2P · 2025-08-04
> > **reply to rebuttal**
> >
> > Thank you for the reply. I will keep my scores.

---

### Official Review · Reviewer_GvHx · 2025-07-02

**Clarity:** 3
**Significance:** 3
**Originality:** 3
**Rating:** 4
**Confidence:** 4

**Summary:**

The paper studies the time complexity of Personalized PageRank (PPR) computation. It introduces a nested evolving set process to improve the time complexity to $\tilde{O}(\min(R^2/\epsilon^2, m))$, where $R$ is a constant defined by the process and $\epsilon$ is the normalized additive error parameter. In particular, the time dependence on the damping factor $\alpha$ is improved to $1/\sqrt{\alpha}$. Theoretical analysis and experimental results are provided to demonstrate the effectiveness of the proposed methods.

**Questions:**

I’ve described my questions in the “Weaknesses” section above. Please refer to that for details.

**Ethical Concerns:**

["NO or VERY MINOR ethics concerns only"]

**Final Justification:**

After reading the paper and the authors' rebuttal, I decided to keep my original score.

**Limitations:**

Yes, there is not much concern about the potential negative societal impact of this paper.

**Paper Formatting Concerns:**

I think the paper follows the NeurIPS 2025 Paper Formatting Instructions.

**Quality:**

3

**Strengths And Weaknesses:**

Strengths:

S1. The problem of efficiently estimating Personalized PageRank (PPR) on large graphs has been widely studied and has numerous applications. Improving the time dependence on $1/\alpha$ is of particular interest in recent work, so the paper is well motivated.

S2. The paper is generally well written and easy to follow (though some issues should be addressed, as outlined below). The theoretical analysis appears sound. Experimental results and source code are also provided, which strengthens the paper’s contributions.

S3. The paper demonstrates that the time dependence on $1/\alpha$ can be improved using nested evolving sets. While the improvement is partial, the underlying idea and algorithmic structure could offer new insights to the community.


Weaknesses:

W1. The complexity results presented in the paper are not very interesting to me. The proposed algorithm has quadratic dependence on $1/\epsilon$, where $\epsilon$ is the degree-normalized additive error. In contrast, the basic Forward Push algorithm (FOCS 2006) has a time complexity of $O(1/(\epsilon \alpha))$ (please correct me during the rebuttal if there is any misunderstanding). In other words, while the paper improves the dependence on $\alpha$ from $1/\alpha$ to $1/\sqrt{\alpha}$, it introduces a worse dependence on $\epsilon$. This trade-off weakens the overall contribution.

W2. Although the writing is generally clear, some aspects of the presentation need refinement: First, it would be better to clearly define the problem (i.e., the approximation error requirement) at the beginning of the paper. Presenting the complexity bound before defining the problem makes the bound hard to interpret. Second, the use of the term ”$\epsilon$-approximation” is potentially confusing. In most literature, this refers to relative error, while in the paper, $\epsilon$ denotes degree-normalized additive error. Clarifying this distinction earlier would help avoid confusion.

---

> ### Author Rebuttal · Authors · 2025-07-30
>
> **W1.** Concern about the tradeoff between the sparsity (related with $\epsilon$) and the condition number (related with $\alpha$)
>
>  **A1.** We appreciate the reviewer’s insightful comment regarding the tradeoff between $\epsilon$ and the condition number $\alpha$. As discussed in Martínez-Rubio's work (see [1]), the final bound indeed represents a tradeoff between the dependence on the condition number, i.e., $1/\alpha$, and the sparsity, i.e., $1/\epsilon$. Specifically, while the condition number dependence scales as $1/\sqrt{\alpha}$, the sparsity dependence increases to $1/\epsilon^2$, which aligns with the conjecture and discussion in their work. Exploring whether a tighter dependence on $\epsilon$ can be achieved remains an interesting direction for future research, and we plan to investigate this further in our subsequent work.
>
> [1] Martínez-Rubio, David, Elias Wirth, and Sebastian Pokutta. "Accelerated and sparse algorithms for approximate personalized PageRank and beyond." The Thirty Sixth Annual Conference on Learning Theory. PMLR, 2023.
>
> **W2.** The concern of the presentation in our abstract and introduction.
>
> **A2.** We appreciate the reviewer's valuable suggestions for improving the clarity of our presentation. We acknowledge these points (e.g., approximation error definition, $\epsilon$-approximation, etc.) and will revise the paper to more intuitively and clearly present our results before introducing the problem formulation.

---

> > ### Comment · Reviewer_GvHx · 2025-08-01
> > **Reply to authors' rebuttal**
> >
> > Thanks for the response. I will keep my score.

---

### Official Review · Reviewer_P4e6 · 2025-07-03

**Clarity:** 3
**Significance:** 3
**Originality:** 3
**Rating:** 5
**Confidence:** 3

**Summary:**

This paper proposes the Accelerated Evolving Set Process (AESP) framework that leverages the accelerated inexact proximal operator to improve the computation efficiency of personalized PageRank. Particularly, the proposed algorithm improves the time complexity of existing methods for $\epsilon$-approximate personalized PageRank computation from $O(vol(\mathcal{S}_t)/(\alpha \gamma_t))$ to $O(vol(\mathcal{S}_t)/(\sqrt{\alpha} \gamma_t))$.

**Questions:**

1. How do the proposed methods compare to the sampling-based PPR estimation algorithms (e.g., FORA [KDD'2017]) in empirical efficiency?
2. What is the intuition underlying the proposed methods for the lower time complexity?

**Ethical Concerns:**

["NO or VERY MINOR ethics concerns only"]

**Final Justification:**

Thank you for the responses. I would maintain my rating given the theoretical contributions in this paper.

**Limitations:**

yes.

**Quality:**

3

**Strengths And Weaknesses:**

S1. The paper is theoretically grounded and technically grounded.

S2. The idea of utilizing the accelerated inexact proximal operator is novel.

S3. The methods are guaranteed to converge without any additional assumptions, and the time complexity achieves a provable improvement over existing works.

W1. The assumption on constant $R$ may not hold.

W2. The empirical studies that compare the proposed methods against the algorithms for personalized PageRank estimation on more datasets are needed. e.g., FORA [KDD'17], TopPPR [SIGMOD'19].

---

> ### Author Rebuttal · Authors · 2025-07-30
>
> **W1.** The concern of the assumption on a constant $R$  may not hold.
>
> **A1.** We appreciate the reviewer's insightful observation regarding the assumption of a constant $R$. As we discussed in line 248, we have provided explanations concerning this assumption. Overall, our experimental results demonstrate that: 1). $R$ does not exhibit any significant trend with respect to variations in graph size; 2). The increase of $R$ with decreasing $\alpha$ shows a relatively gradual trend. Based on our comprehensive experiments, we found that $R$ generally maintains a magnitude on the order of $\log(1/\alpha)$. These observations support the constant $R$ is a relatively small constant parameter in our analysis.
>
> **Q1.** How do the proposed methods compare to the sampling-based PPR estimation algorithms (e.g., FORA [KDD'2017]) in empirical efficiency?
>
> **A1.** Thank you for your question. We appreciate the opportunity to clarify the comparison with FORA [KDD'2017].  1) **Implementation Differences**: FORA is implemented in C++, while our method is implemented in Python with Numba acceleration. This difference in implementation languages and frameworks makes direct runtime comparisons less equitable.  2) **Algorithmic Frameworks**: FORA combines an approximate PPR (APPR, i.e., Forward Push) step with Monte Carlo random walks to refine the estimation. In contrast, our method (AESP) focuses on computing the full PPR vector for a given source node, rather than pairwise node-to-node probabilities. Thus, the two methods address distinct problem formulations, making operational comparisons less meaningful.  3) **Potential Synergy**: Empirically, our method demonstrates strong early-stage acceleration, which could potentially improve the Forward Push phase in FORA. This represents a promising direction for future work.
>
> **Q2.** What is the intuition underlying the proposed methods for the lower time complexity?
>
> **A2.** We sincerely thank the reviewer for raising this important question about the intuition behind our method's improved time complexity. Our proposed framework is based on the following fact: at each time, we use a faster algorithm for solving a simpler version of the original problem, which has a constant condition number. Combine these solutions together to obtain the approximation of the original problem. These techniques are in the same vein as those in [1]. In optimization, this is the Catalyist framework.
>
> [1] Peng, Richard, and Daniel A. Spielman. "An efficient parallel solver for SDD linear systems." Proceedings of the forty-sixth annual ACM Symposium on Theory of Computing (STOC). 2014.

---

> > ### Comment · Reviewer_P4e6 · 2025-08-04
> >
> > Thank you for the clarification. I am learning towards maintaining the scores.

---

### Note · Authors · 2025-08-15

We provide a summary of our key contributions and some interesting open questions as follows:

Our work proposes the **Accelerated Evolving Set Process (AESP)** framework, which leverages an accelerated inexact proximal operator to enhance the computational efficiency of PPR computation. The proposed algorithm achieves a theoretical improvement in time complexity for $\epsilon$-approximate personalized PageRank computation, reducing it from $O(\overline{\operatorname{vol}}(\mathcal{S}_t)/(\alpha \overline{\gamma}_t))$ to $O(\overline{\operatorname{vol}}(\mathcal{S}_t)/(\sqrt{\alpha} \overline{\gamma}_t))$, where $\alpha$ is the damping factor. The key strengths include:  1) **Novel framework**: The use of an accelerated inexact proximal operator introduces new local methods for a type of problem; 2) **Theoretical Contributions**: The paper establishes a solid theoretical foundation with provable guarantees without additional assumptions and demonstrates a provable improvement in time complexity over prior works; 3) **Empirical results**: validation on large-scale graphs further confirms the practical efficiency of AESP, particularly its early-stage acceleration. However, we acknowledge that certain aspects, such as the assumption of R and the trade-off between $\alpha$ (condition number) and $\epsilon$ (sparsity), which is an interesting open problem, need further study.

We sincerely thank all reviewers and ACs for their valuable feedback, which has greatly helped us improve the manuscript. We hope this response clarifies our contributions.

---

### Decision · Program_Chairs · 2025-09-17

**Decision:**

Accept (poster)

**Comment:**

The authors describe for computing personalized PageRank where the restart is supported on one node.  The AESP algorithm is based on an accelerated inexact proximal operator with impressive empirical performance (in terms of number of operations; the experiments do not appear to discuss wall clock time) and scaling in the damping parameter of $O(R^2 \alpha^{-1/2} \epsilon^{-2})$ as opposed to the usual $O(\alpha^{-1} \epsilon^{-1})$ time required for competing methods.

The reviewers were all at least somewhat positive, but almost all pointed to two common concerns:

- The "constant" $R$ is not very well characterized.  The authors show that empirically it remains bounded and modest for the examples given, but there is not any real characterization in terms of graph properties.
- The improved scaling in the damping factor $\alpha$ comes at the cost of worse scaling with the sparsity tolerance $\epsilon$.

The authors acknowledge that these need further study.  Discussion was otherwise fairly terse, though three of the reviewers also suggested looking at other baselines in the comparison.

The personal PageRank problem is an interesting and useful one, and the proposed method appears to have interesting theoretical and empirical advantages, even with the limitations noted by the reviewers (who were all at least borderline positive).